# TIC-GRPO: Provable and Efficient Optimization for Reinforcement Learning from Human Feedback

## Abstract

Group Relative Policy Optimization (GRPO), recently introduced by DeepSeek, is a critic-free reinforcement learning algorithm for fine-tuning large language models. GRPO replaces the value function in Proximal Policy Optimization (PPO) with group-normalized rewards while retaining PPO-style token-level importance sampling based on an old policy. We show that the GRPO update rule actually estimates the policy gradient at the old policy rather than the current one; however, because the old policy is refreshed every few steps, the gap remains small and the resulting bias is negligible in practice. To validate this, we perform an ablation study that removes importance sampling entirely and instead applies gradients estimated at a fixed old policy across multiple optimization steps. Remarkably, this simplified approach achieves performance comparable to standard GRPO.

Motivated by these findings, we propose a new algorithm: Trajectory level Importance Corrected GRPO (TIC-GRPO). TIC-GRPO replaces token level importance ratios with a single trajectory level probability ratio, yielding an unbiased estimate of the current policy gradient while preserving the critic free structure. Furthermore, we present the first theoretical convergence analysis for GRPO style methods, covering both the original GRPO and our proposed variant.

## 1 Introduction

Reinforcement learning from human feedback (RLHF) (Zhu et al., 2023; Bai et al., 2022; Greenblatt et al., 2024) has become a standard technique for aligning large language models (LLMs) with desired behaviors. Among RLHF approaches, Proximal Policy Optimization (PPO) (Schulman et al., 2017) is widely adopted but requires training an additional value network (critic), making it resource-intensive and difficult to scale. To address this, recent work proposed Group Relative Policy Optimization (GRPO) (Shao et al., 2024), a critic-free alternative that estimates advantages through group-wise reward normalization while retaining PPO-style importance sampling with respect to an old policy. Owing to its simplicity and effectiveness, GRPO has been integrated into several open-source RLHF pipelines.

Despite its empirical success, the theoretical properties of GRPO remain underexplored. In particular, GRPO employs token-level importance sampling against the old policy, yet its update rule is not a direct estimator of the current policy gradient. We show that the practical GRPO update in fact corresponds to the policy gradient evaluated at the old policy $\pi_{\text{old}}$, plus a bias term induced by the mismatch between $\pi$ and $\pi_{\text{old}}$. This bias is typically small in practice because $\pi_{\text{old}}$ is refreshed to the current policy every few optimization steps (e.g., every 4–10), limiting divergence. An ablation study confirms this intuition: when we entirely remove importance sampling and, within each inner loop, perform all updates using gradients estimated at $\pi_{\text{old}}$ before refreshing it, the resulting performance remains comparable to that of standard GRPO.

Motivated by this, we propose TIC-GRPO: replace token-level importance weights with trajectory-level ratios, and further introduce two lightweight modifications—length-corrected group normalization and upper-only clipping—which together yield a stable, unbiased, and memory-efficient update. Furthermore, we present the first theoretical convergence analysis for GRPO-style methods, covering both the original GRPO and our proposed variant. Finally, we validate TIC-GRPO

on two standard alignment benchmark AIME. Our experiments show that TIC-GRPO significantly outperforms standard GRPO in both accuracy and convergence rate.

**Contributions** This paper makes the following key contributions:

- We analyze the practical update rule of GRPO and show that it estimates the policy gradient at the old policy $\pi_{\text{old}}$, not the current one. We further explain why this approximation remains effective in practice due to limited policy drift.

- We propose a new algorithm, TIC-GRPO, which replaces token-level importance sampling with a single trajectory-level ratio. In addition, it incorporates two minor modifications: the Length-Corrected Group Normalization Regularizer and the Upward-Only Clipping Mechanism.

- We provide the first theoretical convergence analysis for GRPO-style methods, including both the original GRPO and our variant.

- We empirically validate TIC-GRPO on the AIME dataset, demonstrating consistent improvements in accuracy and convergence speed over the original GRPO. Ablation studies further show that our two minor modifications are effective even when applied individually on the token-level clipping mechanism.

**Related Work** A recent concurrent work by Zheng et al. (2025) proposes a similar idea of replacing token-level importance sampling in GRPO with a trajectory-level formulation, named Group Sequence Policy Optimization (GSPO). Importantly, their work was developed independently and concurrently with ours.

In comparison, our work provides a more detailed explanation of why the original GRPO update remains effective in practice despite its inherent bias, which we attribute to the limited policy drift arising from frequent updates to the old policy. Moreover, we present the first theoretical convergence analysis for GRPO-style methods. Our algorithm also differs in implementation details: we do not apply sequence-length square-root scaling to the importance sampling, and we adopt a modified clipping mechanism. In Section 6 and Appendix A, we include GSPO as a baseline for comparison and empirically observe that our method outperforms it.

Another important baseline considered in this work is the Decoupled Clip and Dynamic Sampling Policy Optimization (DAPO) algorithm (Yu et al., 2025).

## 2 PRELIMINARIES: REINFORCEMENT LEARNING FOR LLMS AND GRPO

We begin by formalizing the reinforcement learning (RL) setup used for aligning large language models (LLMs) and by reviewing the GRPO algorithm recently proposed by DeepSeek.

### 2.1 REINFORCEMENT LEARNING IN CoT REASONING

We model Chain-of-Thought (CoT) reasoning as a sequential decision-making process under an RL framework. Let $s_0$ denote the initial prompt. At each time step $t$, the language model generates a token $a_t \in \mathcal{V}$ from a vocabulary $\mathcal{V}$, forming an evolving reasoning chain

$$c_t = (s_0, a_1, a_2, \ldots, a_t),$$

which we refer to as the *partial chain* or intermediate reasoning state.

To ensure consistent representation across time, each intermediate chain $c_t$ is embedded into a fixed-dimensional space $\mathbb{R}^{T \times d}$ by zero-padding the remaining positions:

$$s_t = (s_0, a_1, \ldots, a_t, \underbrace{0, \ldots, 0}_{(T-t) \text{ tokens}})^\top.$$

Thus, all reasoning states share the same dimensionality, and the policy network at each step takes the zero-padded full chain $s_t$ (as an element of $\mathbb{R}^{T \times d}$) as its input. Different from conventional RL formulations—where the policy is typically parameterized on the current local state or a single

token—our framework conditions the policy on the *entire reasoning chain* up to time $t$. This design enables the model to exploit global contextual dependencies across all preceding reasoning steps when generating the next token. We denote the final state after the reasoning sequence by $s_T$.

Unlike conventional RL environments with dense or intermediate rewards, CoT reasoning provides a *single, sparse reward* observed only at the final step $T$, typically reflecting task correctness or logical validity. This structure introduces a long-horizon credit assignment challenge: intermediate reasoning steps receive no direct feedback, yet they critically determine the final outcome.

We parameterize the model policy $\pi_\theta(a \mid s)$ as the probability of generating token $a$ given the current padded chain $s \in \mathbb{R}^{T \times d}$, and define the expected return as

$$J(\theta) = \mathbb{E}_{s_T \sim \pi_\theta}[r(s_T)] - \beta \, \mathrm{KL}(\pi_\theta \,\|\, \pi_{\mathrm{ref}}),$$

where $r(s_T)$ denotes the final reward assigned to the complete chain, and the KL regularization term constrains the policy to remain close to a reference model $\pi_{\mathrm{ref}}$. The objective is optimized via gradient ascent. Notably, many popular algorithms for LLM alignment, such as PPO and the more recent GRPO family, share this gradient-ascent foundation; their primary distinctions lie in how they efficiently estimate and stabilize the underlying policy gradient.

Since the reward is observed only at the final timestep, the policy gradient takes the form

$$\nabla_\theta J(\theta) = \underbrace{\mathbb{E}_{s_T \sim \pi_\theta}[\nabla_\theta \log \pi_\theta(s_T)\, r(s_T)]}_{\text{Policy Gradient Term}} - \beta \nabla_\theta \mathrm{KL}(\pi_\theta \,\|\, \pi_{\mathrm{ref}}),$$

where $\pi_\theta(s_T) = \prod_{t=1}^{T} \pi_\theta(a_t \mid s_{t-1})$ denotes the trajectory probability.

This formulation captures the essence of *reasoning as trajectory optimization*: each CoT reasoning chain corresponds to a sequence of actions optimized for correctness under delayed reward feedback. It provides a principled framework for analyzing how RL fine-tuning enables LLMs to extend reasoning depth, stability, and coherence.

### 2.1.1 CONNECTION TO CONVENTIONAL REINFORCEMENT LEARNING

Although the reward in CoT reasoning is only provided at the final step, the framework can be naturally related to conventional RL formulations. By expanding the joint trajectory probability as

$$\pi_\theta(s_T) = \prod_{t=1}^{T} \pi_\theta(a_t \mid s_{t-1}),$$

the log-probability decomposes into a token-wise sum:

$$\log \pi_\theta(s_T) = \sum_{t=1}^{T} \log \pi_\theta(a_t \mid s_{t-1}).$$

Substituting this into the policy gradient yields

$$\text{Policy Gradient Term} = \mathbb{E}_{s_T \sim \pi_\theta}\left[\sum_{t=1}^{T} r(s_T)\, \nabla_\theta \log \pi_\theta(a_t \mid s_{t-1})\right]. \tag{1}$$

The final-step reward $r(s_T)$ can thus be interpreted through the lens of the classical *policy gradient theorem*, which states that

$$\text{Policy Gradient Term} = \mathbb{E}_{s_T \sim \pi_\theta}\left[\sum_{t=1}^{T} Q(s_{t-1}, a_t)\, \nabla_\theta \log \pi_\theta(a_t \mid s_{t-1})\right],$$

where $Q(s_{t-1}, a_t)$ denotes the state–action value function. By comparing Eq. 1 with the classical form, we observe that the broadcasted reward $r(s_T)$ serves the same functional role as $Q(s_{t-1}, a_t)$ in the traditional policy gradient. Indeed, $r(s_T)$ can be viewed as an *unbiased estimator* of the true state–action value, since

$$\mathbb{E}_{s_T \sim \pi_\theta}\big[r(s_T) \,\big|\, s_0, a_1, a_2, \ldots, a_t\big] = Q(s_{t-1}, a_t).$$

This observation reveals that CoT-RL can be regarded as a degenerate instance of standard RL, in which the return signal is available only at the terminal step and uniformly propagated to all preceding actions. Such equivalence provides a theoretical bridge connecting reasoning-oriented fine-tuning with classical policy-gradient theory.

**Proposed research: theoretical guarantees of PG with autoregressive policy and trajectory rewards.** The function approximation of an LLM policy takes a autoregressive form which has not been taken into account previously in standard RL. This task aims to first develop a convergence theory of PG methods by explicitly leveraging the autoregressive nature of the policy, and understand the impact of different forms of state space regularizations on the convergence, such as length regularization and format regularization. In addition, in the standard analysis of PG methods, it is often assumed the reward is provided at every step of the rollout trajectory, which facilitates the evaluation of the policy gradients and value functions. However, to avoid reward hacking, LLM reasoning typically only relies on the terminal reward at the end of the trajectory, which evaluates the correctness of the final answer. An intriguing question is how the credit assignment of the terminal reward such as in our preliminary work **?** impacts the policy gradient updates of an autoregressive policy, which we aim to investigate using the symbolic reasoning task in Thrust 1.

**Proposed research: emergence of test-time scaling.** One intriguing empirical behavior of RL is that the length of the CoT traces increases during training without explicit regularization **?**. The proposed task aims to provide theoretical understanding to this phenomenon, by using the LEGO task studied in Thrust 1 task 1a. Recall that our preliminary work **?** has established that a curriculum of self-labeled dataset with increasing lengths can bootstrap longer reasoning capabilities. We speculate that if we train RL directly over all problem lengths, the model will first obtain signals from the easiest task in the dataset (which requires shorter CoT), and learn through an implicit curriculum via gradually being able to complete the increasingly difficult task in the series of tasks (requiring longer CoT). The proposed research task will formalize this intuition and provide a rigorous analysis, which will lead to better understanding of the emergence of test-time scaling via the lens of training dynamics.

## 2.2 SETUP

Let $s_0$ denote the initial prompt. At each time step $t$, the large language model generates a token $a_t \in \mathcal{V}$, where $\mathcal{V}$ denotes the vocabulary. Each token in $\mathcal{V}$ is represented as a vector in $\mathbb{R}^d$. Then we define the state at time $t$ as $s'_t := (s_0, a_1, \ldots, a_t)^\top \in \mathbb{R}^{t \times d}$. To ensure consistent dimensionality across time steps, we embed each state into a fixed-dimensional space $\mathbb{R}^{T \times d}$ via zero-padding:

$$s_t := (s_0, a_1, \ldots, a_t, \underbrace{0, \ldots, 0}_{(T-t) \text{ tokens}})^\top,$$

where the final $T - t$ entries are zero vectors in $\mathbb{R}^d$. We also let $\mathcal{S}_t$ denote the set of all possible states $s_t$. We readily observe the following inclusion relation:

$$\mathcal{S}_1 \subset \mathcal{S}_2 \subset \ldots \subset \mathcal{S}_T.$$

In the CoT reasoning setting, we assume a predefined reward function

$$r(s) : \mathbb{R}^{T \times d} \to \mathbb{R},$$

which evaluates the quality of a complete generated state $s$. The rewards are sparse and provided only at the final step, i.e., when $t = T$.

The core of an CoT reasoning is the parameterized policy. We write

$$\pi_\theta(a \mid s) : \mathbb{R}^l \times \mathbb{R}^d \times \mathbb{R}^{T \times d} \to [0, 1]$$

to denote the probability of generating a token $a \in \mathbb{R}^d$ given the current state $s \in \mathbb{R}^{T \times d}$ under model parameters $\theta \in \mathbb{R}^l$. Since the token $a_t$ output by the model at time step $t$ together with the previous state $s_{t-1}$ uniquely determines the state $s_t$, we have the identity $\mathbb{P}_\theta(s_t \mid s_{t-1}) = \pi_\theta(a_t \mid s_{t-1})$. Here, $\mathbb{P}_\theta(s_t \mid s_{t-1}) : \mathbb{R}^l \times \mathbb{R}^{T \times d} \times \mathbb{R}^{T \times d} \to [0, 1]$ denotes the conditional probability of the current state $s_t$ given the previous state $s_{t-1}$, under parameters $\theta \in \mathbb{R}^l$.

We now define the trajectory probability and value function. The joint probability of generating a full trajectory under policy $\pi_\theta$ is given by:

$$\mathbb{P}_\theta(s_T \mid s_0) = \prod_{t=1}^{T} \mathbb{P}_\theta(s_t \mid s_{t-1}).$$

The goal is to maximize the expected return:

$$J(\theta) = \mathbb{E}_{s_T \sim \pi_\theta}\left[r(s_T)\right] - \mathbf{KL}(\pi_\theta \| \pi_{\theta_{\text{ref}}}) = \sum_{s_T \in \mathcal{S}_T} \mathbb{P}_\theta(s_T | s_0) r(s_T) - \mathbf{KL}(\pi_\theta \| \pi_{\theta_{\text{ref}}}). \tag{2}$$

Here $|s_T|$ denotes the length of the response $s_T$. The length $|s_T|$ is determined by the stop token: if the stop token appears before $T$, the generation terminates at that token. Moreover, for any $\theta$, we stipulate that the parameterized policy $\pi_\theta$ maps every state containing a stop token consistently to the stop token.

Because the reward of any meaningful reinforcement learning problem is necessarily bounded, the value function $J(\theta)$, $(\theta \in \mathbb{R}^d)$ admits a theoretical maximum, which we denote by $J^*$.

The optimization of $J(\theta)$ typically follows a gradient ascent (GA) scheme (Yuan et al., 2022; Zhang et al., 2020)[1]:

$$\theta_{n+1} = \theta_n + \eta \nabla_{\theta_n} J(\theta_n),$$

with learning rate $\eta$. Algorithms like PPO and GRPO build on this principle with various modifications to improve performance.

In CoT reasoning setting, since the reward $r(s_T)$ is assigned only at the final timestep and does not depend on $\theta$, the policy gradient simplifies as:

$$\nabla J(\theta) = \sum_{s_T \in \mathcal{S}_T} \left(\nabla \mathbb{P}_\theta(s_T | s_0)\right) r(s_T) = \mathbb{E}_{s_T \sim \pi_\theta}\left[\left(\nabla \log \mathbb{P}_\theta(s_T | s_0)\right) r(s_T)\right]. \tag{3}$$

**Notation.** *Throughout the remainder of the paper, $\nabla$ denotes gradients with respect to $\theta$ (or $\theta_s$), unless explicitly stated otherwise.*

## 2.3 REVIEW OF GROUP RELATIVE POLICY OPTIMIZATION (GRPO)

Group Relative Policy Optimization (GRPO), recently proposed by DeepSeek, is a reinforcement learning algorithm for aligning LLMs without a value-function critic. Instead of computing global advantage estimates, GRPO uses relative rewards within a group of candidate responses to estimate local advantage. Like PPO, GRPO employs a decoupled optimization structure: the old policy $\pi_{\theta_{\text{old}}}$ is held fixed while the current policy $\pi_\theta$ is updated over multiple gradient steps using the same batch of trajectories, improving sample efficiency.

For the convenience of the subsequent analysis, we define the $\sigma$-algebra generated by $\theta_{\text{old}}$ as $\mathscr{F}_{\text{old}} := \sigma(\theta_{\text{old}})$. Given a prompt $s_0$, the old policy $\pi_{\theta_{\text{old}}}$ generates a group $G = \{s_T^{(1)}, \ldots, s_T^{(|G|)}\}$ of full responses. For the convenience of the subsequent analysis, we define a random variable $\xi_G(\cdot) : \mathcal{S}_T \to [0, 1]$ that uniquely determines the group sampling. Specifically, if a state $s_T$ appears $w$ times in the sample $G$, then

$$\xi_G(s_T) := \frac{w}{|G|}.$$

With this definition, the summation over the group can be written in the following form:

$$\frac{1}{|G|} \sum_{i=1}^{|G|} f(s_T^{(i)}) = \sum_{s_T \in \mathcal{S}_T} \xi_G(s_T) f(s_T),$$

where $f : \mathcal{S}_T \to \mathbb{R}$ is any test function. Moreover, we shall denote this family of vectors uniformly by

$$\boldsymbol{\xi}_{\theta_{\text{old}}, G} := \left(\xi_G(s_T)\right)_{s_T \in \mathcal{S}_T}. \tag{4}$$

---

[1]We use gradient ascent as the goal is to maximize $J(\theta)$. Gradient descent is equivalent up to a sign change.

GRPO then computes normalized advantages within the group as:

$$A_G(s_T) = \frac{r(s_T) - \mu_G}{\sigma_G + \delta}, \quad \mu_G = \sum_{s_T \in \mathcal{S}_T} \xi_G(s_T) r(s_T), \quad \sigma_G = \sqrt{\sum_{s_T \in \mathcal{S}_T} \xi_G(s_T)(r(s_T) - \mu_G)^2},$$

where $\delta$ is a smoothing factor to prevent the denominator from approaching zero. The group-normalized advantages $A_G(s_T)$ are then used to construct the objective function.

**Optimization objective** With $\pi_{\theta_{\text{old}}}$ held fixed, we optimize

$$\mathcal{L}_{\text{GRPO}}(\theta, \theta_{\text{old}}) = \sum_{s_T \in \mathcal{S}_T} \xi_G(s_T) \frac{1}{|s_T|} \sum_{t=1}^{T} \text{ClipMin}\left(s_t, \theta, \theta_{\text{old}}\right). \tag{5}$$

Here,

$$\text{ClipMin}\left(s_t, \theta, \theta_{\text{old}}\right)$$
$$:= \min\left\{ \frac{\mathbb{P}_\theta(s_t | s_{t-1})}{\mathbb{P}_{\theta_{\text{old}}}(s_t | s_{t-1})} A_G(s_T), \text{Clip}\left( \frac{\mathbb{P}_\theta(s_t | s_{t-1})}{\mathbb{P}_{\theta_{\text{old}}}(s_t | s_{t-1})}, \epsilon_{\text{low}}, \epsilon_{\text{high}} \right) A_G(s_T) \right\} \tag{6}$$

with the clipping function

$$\text{Clip}(x, \epsilon_{\text{low}}, \epsilon_{\text{high}}) := \begin{cases} 1 - \epsilon_{\text{low}}, & x < 1 - \epsilon_{\text{low}}, \\ x, & 1 - \epsilon_{\text{low}} \le x \le 1 + \epsilon_{\text{high}}, \\ 1 + \epsilon_{\text{high}}, & x > 1 + \epsilon_{\text{high}}. \end{cases}$$

In Eq 5, note that since the large language model degenerates to the identity mapping after the stop token, all terms between $|s_T|$ and $T$ vanish.

Therefore, the summations are equal, i.e.,

$$\sum_{t=1}^{|s_T|} \text{ClipMin}\left(s_t, \theta, \theta_{\text{old}}\right) = \sum_{t=1}^{T} \text{ClipMin}\left(s_t, \theta, \theta_{\text{old}}\right).$$

In original GRPO (Shao et al., 2024), the clipping thresholds in the surrogate objective are symmetric, i.e., $\epsilon_{\text{low}} = \epsilon_{\text{high}}$. A subsequent study showed that employing asymmetric clipping ($\epsilon_{\text{low}} \ne \epsilon_{\text{high}}$) can improve empirical performance, and accordingly renamed the modified algorithm Decouple Clip and Dynamic Sampling Policy Optimization (DAPO) (Yu et al., 2025). In addition, original GRPO includes a regularization term involving the pretrained model $\pi_{\text{ref}}$, namely the KL divergence $\text{KL}(\pi_\theta \| \pi_{\text{ref}})$. However, as noted in DAPO, a model fine-tuned with human feedback may deviate substantially from the pretrained model, and this KL-divergence regularization can hinder performance (Yu et al., 2025). Consequently, it was removed. In this work, we follow the DAPO setting, removing the KL-divergence. In the remainder of this paper, we do not distinguish between the names DAPO and GRPO, as the two algorithms share similar mechanisms.

Eq. 5 can be maximized with stochastic gradient ascent (SGA) or adaptive methods such as Adam (Kingma & Ba, 2014; Wang et al., 2023; Jin et al., 2024); in this paper we adopt vanilla SGA.

**Update rule** We now present the update rule under a fixed old policy $\pi_{\text{old}}$:

$$\theta_{s+1} = \theta_s + \eta \nabla \mathcal{L}_{\text{GRPO}}(\theta_s, \theta_{\text{old}}),$$

where $\eta$ is the learning rate and the gradient $\nabla \mathcal{L}_{\text{GRPO}}(\theta, \theta_{\text{old}})$ can be written as

$$\nabla \mathcal{L}_{\text{GRPO}}(\theta, \theta_{\text{old}}) = \sum_{s_T \in \mathcal{S}_T} \xi_G(s_T) \frac{1}{|s_T|} \sum_{t=1}^{T} \nabla\left( \text{ClipMin}\left(s_t, \theta, \theta_{\text{old}}\right) \right). \tag{7}$$

After performing $K$ gradient steps under a fixed old policy $\pi_{\theta_{\text{old}}}$, the reference is updated according to $\pi_{\theta_{\text{old}}} \leftarrow \pi_\theta$. The full algorithm is summarized in Eq. 7.

## 3  A DECOMPOSITION OF GRPO'S GRADIENT TERM

In this section, we analyze the Gradient Term in Eq. 7 and show that it can be interpreted as an asymptotically unbiased estimator of the policy gradient evaluated at $\pi_{\theta_{\text{old}}}$. To do this, we first define the following two events:

$$\mathcal{B}^+(s_t, \theta, \theta_{\text{old}}) := \left\{ \frac{\mathbb{P}_\theta(s_t|s_{t-1})}{\mathbb{P}_{\theta_{\text{old}}}(s_t|s_{t-1})} \leq 1 + \epsilon_{\text{high}}, \ A_G(s_T) \geq 0 \right\},$$

$$\mathcal{B}^-(s_t, \theta, \theta_{\text{old}}) := \left\{ \frac{\mathbb{P}_\theta(s_t|s_{t-1})}{\mathbb{P}_{\theta_{\text{old}}}(s_t|s_{t-1})} \geq 1 - \epsilon_{\text{low}}, \ A_G(s_T) < 0 \right\},$$

and the event $\mathcal{B}(s_t, \theta, \theta_{\text{old}}) = \mathcal{B}^+(s_t, \theta, \theta_{\text{old}}) \cup \mathcal{B}^-(s_t, \theta, \theta_{\text{old}})$. Then we can get:

$$\nabla \mathcal{L}_{\text{GRPO}}(\theta, \theta_{\text{old}}) \overset{(*)}{=} \sum_{s_T \in \mathcal{S}_T} \xi_G(s_T) \frac{1}{|s_T|} \sum_{t=1}^{T} \mathbf{1}_{\mathcal{B}(s_t,\theta,\theta_{\text{old}})} \frac{\mathbb{P}_\theta(s_t|s_{t-1})}{\mathbb{P}_{\theta_{\text{old}}}(s_t|s_{t-1})} \nabla \log \mathbb{P}_\theta(s_t|s_{t-1}) A_G(s_T)$$

$$= \underbrace{\frac{1}{\sigma_{\theta_{\text{old}}}} \sum_{s_T \in \mathcal{S}_T} \xi_G(s_T) \frac{1}{|s_T|} \sum_{t=1}^{T} \nabla \log \mathbb{P}_{\theta_{\text{old}}}(s_t|s_{t-1}) r(s_T)}_{\tilde{\nabla} J(\theta_{\text{old}})}$$

$$+ \underbrace{\frac{1}{\sigma_{\theta_{\text{old}}}} \sum_{s_T \in \mathcal{S}_T} \xi_G(s_T) \frac{1}{|s_T|} \sum_{t=1}^{T} \frac{\mathbf{1}_{\mathcal{B}(s_t,\theta,\theta_{\text{old}})}}{\mathbb{P}_{\theta_{\text{old}}}(s_t|s_{t-1})} \left( \nabla \mathbb{P}_\theta(s_t|s_{t-1}) - \nabla \mathbb{P}_{\theta_{\text{old}}}(s_t|s_{t-1}) \right) A_G(s_T)}_{\Xi_g(\theta, \theta_{\text{old}})}$$

$$+ \underbrace{\sum_{s_T \in \mathcal{S}_T} \xi_G(s_T) \sum_{t=1}^{T} \nabla \log \mathbb{P}_{\theta_{\text{old}}}(s_t|s_{t-1}) B_G(s_T)}_{\frac{1}{\sigma_{\theta_{\text{old}}}} \Xi_s(\theta, \theta_{\text{old}})}$$

$$+ \underbrace{\frac{1}{\sigma_{\theta_{\text{old}}}} \left( - \sum_{s_T \in \mathcal{S}_T} \xi_G(s_T) \frac{1}{|s_T|} \sum_{t=1}^{T} \mathbf{1}_{\mathcal{B}^c(s_t,\theta,\theta_{\text{old}})} \nabla \log \mathbb{P}_{\theta_{\text{old}}}(s_t|s_{t-1}) A_G(s_T) \right)}_{\Xi_c(\theta, \theta_{\text{old}})}. \tag{8}$$

In the above expression, we define

$$\sigma_{\theta_{\text{old}}} := \delta + \mathbb{E}_{G \sim \pi_{\theta_{\text{old}}}} \left[ \sum_{s_T \in \mathcal{S}_T} \xi_G(s_T) \sigma_G \,\Big|\, \mathscr{F}_{\theta_{\text{old}}} \right], \quad B_G(s_T) := \frac{A_G(s_T)}{|s_T|} - \frac{1}{\sigma_{\theta_{\text{old}}}} \frac{r(s_T)}{|s_T|}. \tag{9}$$

Based on the decomposition above, we observe that $\tilde{\nabla} J(\theta_{\text{old}})$ serves as an unbiased estimator of the true policy gradient $\nabla J(\theta_{\text{old}})$, since we clearly have

$$\mathbb{E}_{G \sim \pi_{\theta_{\text{old}}}} \left[ \tilde{\nabla} J(\theta_{\text{old}}) | \mathscr{F}_{\theta_{\text{old}}} \right] = \mathbb{E}_{G \sim \pi_{\theta_{\text{old}}}} \left[ \sum_{s_T \in \mathcal{S}_T} \xi_G(s_T) \frac{1}{|s_T|} \sum_{t=1}^{T} \nabla \log \mathbb{P}_{\theta_{\text{old}}}(s_t|s_{t-1}) r(s_T) \,\Big|\, \mathscr{F}_{\theta_{\text{old}}} \right]$$

$$= \mathbb{E}_{s_T \sim \pi_{\theta_{\text{old}}}} \left[ \nabla \log \mathbb{P}_{\theta_{\text{old}}}(s_T|s_0) \frac{r(s_T)}{|s_T|} \,\Big|\, \mathscr{F}_{\theta_{\text{old}}} \right]$$

$$= \nabla J(\theta_{\text{old}}).$$

The remaining three terms are error terms. These error terms can be controlled during the algorithmic iterations.

A natural question arises:

*why does GRPO remain effective in practice, given that it estimates the gradient at the stale policy $\theta_{old}$ rather than the current iterate $\theta$?*

The key insight is that the old policy $\pi_{\theta_{\text{old}}}$ is refreshed every $K$ steps, i.e., $\pi_{\theta_{\text{old}}} \leftarrow \pi_\theta$. As a result, the discrepancy between $\pi_\theta$ and $\pi_{\theta_{\text{old}}}$ remains small throughout training, allowing the algorithm to

perform reliably even with stale gradient estimates. We empirically validate our hypothesis through a controlled ablation experiment. Specifically, we remove the importance sampling mechanism entirely from DAPO and, within each inner optimization loop where the old policy $\pi_{\theta_{\text{old}}}$ is held fixed, directly perform updates using the policy gradient estimated at $\pi_{\theta_{\text{old}}}$. This setting isolates the effect of importance sampling and allows us to examine how well GRPO performs when relying solely on stale gradients.

We conduct this experiment using the qwen3_1.7b-base model (Team, 2024) on a hybrid dataset comprising the full DAPO-17K corpus and several hundred examples from the AIME dataset (Liu et al., 2024; Ji et al., 2025). The model is trained for a single epoch, with each prompt used exactly once. We use a total batch size of 128 and a mini-batch size of 32, resulting in each sample being reused for 4 updates before refreshing the old policy.

As shown in Figure 1 in Appendix A.1, removing importance sampling does not lead to a significant drop in performance. Especially in the latter stages of the algorithm, removing importance sampling even produced a slight performance gain. This result empirically supports our earlier claim that, due to the limited drift between $\pi_\theta$ and $\pi_{\theta_{\text{old}}}$ within each update cycle, the policy gradient at $\pi_{\theta_{\text{old}}}$ remains a reliable update direction in practice.

This observation naturally leads to the following idea: if we could modify the importance sampling mechanism in GRPO such that the resulting estimator becomes a consistent and asymptotically unbiased estimate of the current policy gradient $\nabla J(\theta)$, then the algorithm's performance could be further improved—both in theory and in practice.

A natural candidate for such a correction is to replace the token-level importance weights used in GRPO with a trajectory-level importance ratio. That is, instead of reweighting each token individually, we consider using the probability ratio over the entire trajectory, aligning the estimator more closely with the form of the true policy gradient. This simple yet principled modification forms the basis of our proposed algorithm, which we introduce in the next section.

## 4 TRAJECTORY-LEVEL IMPORTANCE-CORRECTED GRPO (TIC-GRPO)

In this section, we propose our TIC-GRPO, a principled variant of GRPO. Apart from replacing importance sampling with its trajectory-level version, this paper introduces two relatively minor modifications. First, the group regularization is replaced by a version with a length penalty. Both of these minor changes can be added independently to the original GRPO with token-level importance sampling; second, the clipping mechanism is replaced by an up-only variant.

### 4.1 MAJOR MODIFICATION

**Trajectory-level Importance Sampling.** We replace the token-level importance sampling mechanism in Eq. 5 with a trajectory-level probability ratio $\mathbb{P}_\theta(s_T \mid s_0)/\mathbb{P}_{\theta_{\text{old}}}(s_T \mid s_0)$.

### 4.2 MINOR MODIFICATIONS

**Length-Corrected Group Normalization Regularizer.** We replace the group regularization with the following form:

$$A'_G(s_T) = \frac{\frac{r(s_T)}{|s_T|} - \mu'_G}{\sigma'_G + \delta}, \ \mu'_G = \sum_{s_T \in \mathcal{S}_T} \xi_G(s_T) \frac{r(s_T)}{|s_T|}, \ \sigma'_G = \sqrt{\sum_{s_T \in \mathcal{S}_T} \xi_G(s_T) \left(\frac{r(s_T)}{|s_T|} - \mu'_G\right)^2}.$$

As noted below Eq. 2, the original GRPO algorithm already effectively treats $r(s_T)/|s_T|$ as a new reward. Therefore, when applying group regularization, it is more natural to regularize with respect to this new reward.

**Upper-Only Clipping Mechanism.** We employ a minor technical modification to the standard clipping mechanism used in importance sampling. The original clipping strategy, as defined in Eq. 5, i.e.,

$$\min\left\{\frac{\mathbb{P}_\theta(s_T \mid s_0)}{\mathbb{P}_{\theta_{\text{old}}}(s_T \mid s_0)} A'_G(s_T), \text{Clip}\left(\frac{\mathbb{P}_\theta(s_T \mid s_0)}{\mathbb{P}_{\theta_{\text{old}}}(s_T \mid s_0)}, \epsilon_{\text{low}}, \epsilon_{\text{high}}\right) A'_G(s_T)\right\},$$

treats the sign of the estimated advantage $A'_G(s_T)$ separately: when $A'_G(s_T) \geq 0$, the importance weight is clipped from above by $1 + \epsilon_{\text{high}}$, whereas when $A'_G(s_T) < 0$, it is clipped from below by $1 - \epsilon_{\text{low}}$. However, we observe that retaining only the lower bound $1 - \epsilon_{\text{low}}$ while leaving the upper bound unconstrained fails to reduce the variance of the policy gradient estimator effectively even when $A_G(s_T) < 0$. Motivated by this, we adopt a modified clipping scheme in which only the upper bound is enforced, as follows:

$$\min \left\{ \frac{\mathbb{P}_\theta(s_T \mid s_0)}{\mathbb{P}_{\theta_{\text{old}}}(s_T \mid s_0)}, \text{Clip} \left( \frac{\mathbb{P}_\theta(s_T \mid s_0)}{\mathbb{P}_{\theta_{\text{old}}}(s_T \mid s_0)}, \epsilon_{\text{low}}, \epsilon_{\text{high}} \right) \right\} A'_G(s_T).$$

Under our modified clipping scheme, the importance sampling ratio is truncated from above at $1 + \epsilon_{\text{high}}$, independent of the sign of the estimated advantage $A'_G(s_T)$. The lower bound $1 - \epsilon_{\text{low}}$ is omitted and thus has no effect. This upper-only clipping more effectively reduces the variance of the policy gradient estimator, leading to improved empirical performance. For notational convenience, we denote:

$$\overline{\text{ClipMin}}(s_T, \theta, \theta_{\text{old}}) := \min \left\{ \frac{\mathbb{P}_\theta(s_T \mid s_0)}{\mathbb{P}_{\theta_{\text{old}}}(s_T \mid s_0)}, \text{Clip} \left( \frac{\mathbb{P}_\theta(s_T \mid s_0)}{\mathbb{P}_{\theta_{\text{old}}}(s_T \mid s_0)}, \epsilon_{\text{low}}, \epsilon_{\text{high}} \right) \right\} A'_G(s_T).$$

The original clipping mechanism fails to control variance when the advantage is negative, as noted by Ye et al. (2020); Jin et al. (2023) in PPO. They used a dual-clip approach, while we only clip the upper bound, allowing more tokens to contribute and improving efficiency.

It is worth noting that these two minor modification mechanisms can be applied to the original token-level importance sampling without any other changes. Their isolated effectiveness is confirmed by our ablation experiments in Appendix A.3.

## 4.3 RULES FOR TIC-GRPO

Apart from the above modifications, all other components remain consistent with the original GRPO formulation. The corresponding optimization objective is given by:

$$\mathcal{L}_{\text{TIC-GRPO}}(\theta, \theta_{\text{old}}) = \sum_{s_T \in \mathcal{S}_T} \xi_G(s_T) \overline{\text{ClipMin}} (s_T, \theta, \theta_{\text{old}}). \tag{10}$$

Similarly, we present the update rule under a fixed old policy $\pi_{\text{old}}$ :

$$\theta_{s+1} = \theta_s + \eta \nabla \mathcal{L}_{\text{TIC-GRPO}}(\theta_s, \theta_{\text{old}}),$$

where $\eta$ is the learning rate and the gradient $\nabla \mathcal{L}_{\text{TIC-GRPO}}(\theta, \theta_{\text{old}}, \theta_{\text{ref}})$ can be written as

$$\nabla \mathcal{L}_{\text{TIC-GRPO}}(\theta, \theta_{\text{old}}) = \sum_{s_T \in \mathcal{S}_T} \xi_G(s_T) \nabla \left( \overline{\text{ClipMin}} (s_T, \theta, \theta_{\text{old}}) \right). \tag{11}$$

Here we claim that $\nabla \mathcal{L}_{\text{TIC-GRPO}}(\theta, \theta_{\text{old}})$ can serve as a estimation of policy gradient $\nabla J(\theta)$ at $\theta$, which contrasts with Eq. 7, where it serves only as an estimation at $\theta_{\text{old}}$. Note that this estimation is not as immediate as in Eq. 8; we place the detailed derivation as a separate section in Appendix B.

As in GRPO, the old policy $\pi_{\theta_{\text{old}}}$ is refreshed every $K$ steps by assigning $\pi_{\theta_{\text{old}}} \leftarrow \pi_\theta$. The complete algorithm is summarized in Eq. 13

Intuitively, TIC-GRPO should be more sample-efficient than the original GRPO. However, such intuition alone is insufficient to rigorously justify the algorithm's advantage. In the next section, we address this gap by providing formal convergence rate analyses for both GRPO and TIC-GRPO under mild assumptions—specifically, assuming the score function is Lipschitz continuous and the reward function is bounded. To the best of our knowledge, this constitutes the first theoretical convergence analysis for GRPO-style algorithms. We also provide experimental validation of these findings in Section 6 and Appendix A.

## 5 CONVERGENCE RESULTS

In this section we establish the stationary-point convergence sample complexity of both the original GRPO and TIC-GRPO under two mild and commonly used assumptions.

To facilitate convergence analysis, we begin by rewriting both algorithms in iterative update forms:

**GRPO**

$$\theta_{n,0} = \theta_{n-1,K},$$
$$\theta_{n,s+1} = \theta_{n,s} + \eta\hat{\nabla}\mathcal{L}_{\text{GRPO}}(\theta_{n,s}, \theta_{n,0}), \quad (s = 0, 1, \dots, K). \tag{12}$$

**TIC-GRPO**

$$\theta_{n,0} = \theta_{n-1,K},$$
$$\theta_{n,s+1} = \theta_{n,s} + \eta\hat{\nabla}\mathcal{L}_{\text{TIC-GRPO}}(\theta_{n,s}, \theta_{n,0}), \quad (s = 0, 1, \dots, K). \tag{13}$$

Since we now analyze the overall performance of the above stochastic algorithm, we need to construct a filtration $\{\mathscr{F}_n\}_{n \geq 1}$. Specifically, for each $n$, the $\mathscr{F}_n$ is given by $\mathscr{F}_n := \sigma\left(\boldsymbol{\xi}_{\theta_{1,0},G}, \boldsymbol{\xi}_{\theta_{2,0},G}, \dots, \boldsymbol{\xi}_{\theta_{n,0},G}\right)$, where $\{\boldsymbol{\xi}_{\theta_{n,0},G}\}_{n \geq 1}$ is defined in Eq. 4.

We now present two key assumptions that underlie our convergence analysis for both GRPO and TIC-GRPO.

**Assumption 5.1** (**Lipschitz Continuous Score Function**). *Let $L > 0$ be fixed constants. For all states $s_t \in \mathcal{S}_T$, the score function is Lipschitz continuous in the following sense:* $\|\nabla\log\mathbb{P}_\theta(s_t \mid s_{t-1}) - \nabla\log\mathbb{P}_{\theta'}(s_t \mid s_{t-1})\| \leq L\|\theta - \theta'\|.$

In addition, we require a bounded reward assumption, stated as follows:

**Assumption 5.2** (**Bounded Reward**). *There exists a constant $R > 0$ such that the absolute value of the terminal reward is uniformly bounded. Specifically, for all $s_T$, we have $|r(s_T)| \leq R$.*

This is a common and mild assumption in reinforcement learning, especially in the context of LLM-based applications.

## 5.1 RESULTS OF GRPO

We now present the convergence result for the original GRPO:

**Theorem 5.1.** *(Convergence of GRPO) Assume that the conditions stated in Assumptions 5.1 and 5.2 are satisfied. Let $\theta_{1,0} \in \mathbb{R}^d$ denote an arbitrary initialization of the algorithm, and we set $\eta = \frac{1}{\log|\mathcal{V}|\sqrt{N}}$. Then the sequence $\{\theta_{n,s}\}$ generated by GRPO as defined in Eq. 12 admits the following upper bound:*

$$\frac{1}{N}\sum_{n=1}^{N}\sum_{s=0}^{K-1}\mathbb{E}\left[\|\nabla J(\theta_{n,s})\|^2\right] = \mathcal{O}\left(\frac{\log|\mathcal{V}|\sqrt{\mathbb{E}\left[\mathcal{M}_N^2\right]}}{\sqrt{N}}\right) + \mathcal{O}\left(\frac{1}{\sqrt{|G|}}\right) + \mathcal{O}(\overline{\sigma}_{s_T,N}^2),$$

*where*

$$\mathcal{M}_N := \max_{1 \leq n \leq N}\left\{\max_{1 \leq i \leq |G|}\frac{1}{\pi_{\theta_{n,0}}(a_t^{(i)}|s_{t-1}^{(i)})}\right\},$$

$$\overline{\sigma}_{s_T,N}^2 := \frac{1}{N}\sum_{n=1}^{N}\mathbb{E}_{\{s_T\}\sim\pi_{\theta_{n,0}}}\left[\left|\|s_T\| - \mathbb{E}\left[\|s_T\|\,|\,\mathscr{F}_{n-1}\right]\right|^2\right]. \tag{14}$$

*The quantities hidden in the $\mathcal{O}$ notation are constants depending only on other parameters of the problem.*

Due to space constraints, the proof of this theorem is deferred to Section C.

This represents the first rigorous theoretical result for GRPO. It can be observed that the convergence rate of the original GRPO depends on two quantities, $\mathcal{M}_N$ and $\overline{\sigma}_{s_T,N}^2$, both of which are non-optimizable. The first term arises because the conventional clipping mechanism only truncates the lower bound of importance sampling when the advantage is negative, while leaving the upper bound uncontrolled; as a result, its variance can only be bounded by $\mathcal{M}_N$. The second term comes from the fact that trajectories sampled under the same prompt may have different lengths, whereas the standard GRPO applies group regularization without any length normalization, thereby introducing a fixed error. We argue that these two factors may be among the reasons why the original GRPO suffers from collapse on certain tasks (Li et al., 2025; Chen et al., 2025).

## 5.2 RESULTS OF TIC-GRPO

**Theorem 5.2** (Convergence of TIC-GRPO). *Assume that the conditions stated in Assumptions 5.1 and 5.2 are satisfied. Let $\theta_{1,0} \in \mathbb{R}^d$ denote an arbitrary initialization of the algorithm, and we set $\eta = \min\left\{\frac{1}{\log|\mathcal{V}|\sqrt{N}}, \frac{1}{4K(1+\epsilon_{high})RL}\right\}$. Then the sequence $\{\theta_{n,s}\}$ generated by TIC-GRPO as defined in Eq. 13 admits the following upper bound:*

$$\frac{1}{N}\sum_{n=1}^{N}\sum_{s=0}^{K-1}\mathbb{E}\left[\|\nabla J(\theta_{n,s})\|^2\right] = \mathcal{O}\left(\frac{\log|\mathcal{V}|}{\sqrt{N}}\right) + \mathcal{O}\left(\frac{1}{\sqrt{|G|}}\right).$$

*The quantities hidden in the $\mathcal{O}$ notation are constants depending only on other parameters of the problem.*

Due to space constraints, the proof of this theorem is deferred to Section C.

It can be observed that, compared with the original GRPO, our TIC-GRPO algorithm eliminates the dependence on $\mathcal{M}_N$ and $\overline{\sigma}_{s_T,N}$ in the convergence rate, leading to a tighter convergence bound. We note, however, that this improvement stems solely from the adoption of the Length-Corrected Group Normalization Regularizer and the Upward-Only Clipping Mechanism. In other words, our theoretical results do not yet capture the benefits of response-level importance sampling: adding these two mechanisms alone on top of token-level importance sampling would achieve the same convergence rate. We believe that the advantages of response-level importance sampling are hidden in the constants within $\mathcal{O}(\cdot)$, i.e., the constants associated with response-level importance sampling are more favorable. We leave a precise characterization of this effect as future work.

## 6 EXPERIMENTS

We evaluate TIC-GRPO on the AIME benchmark. Table **??** summarizes the results, including two baselines—GSPO and GRPO (implemented with the DAPO framework)—as well as two ablation variants: one applying only response-level importance sampling and the other applying only upper-bound clipping with length-corrected group normalization. TIC-GRPO consistently outperforms all baselines and ablations, confirming the effectiveness of combining trajectory-level ratios with the two lightweight refinements. Additional experimental details, including training and evaluation plots and further ablation experiments, are provided in Appendix A.2 and A.3.

Table 1: Combined evaluation results on AIME24, AIME25, and MATH500. Numbers in parentheses indicate improvement over the baseline GRPO.

| Model | AIME24 | AIME25 | MATH500 |
|---|---|---|---|
| Qwen3_1.7B_GRPO | 9.17 | 5.31 | 66.6 |
| Qwen3_1.7B_Minor_Modications_Only | 10.31 (+1.14) | 6.64 (+1.33) | 67.4 (+0.8) |
| Qwen3_1.7B_Sentence_Important_Sampling_Only | 10.62 (+1.45) | 6.77 (+1.46) | 68.0 (+1.4) |
| Qwen3_1.7B_GSPO | 10.31 (+1.14) | 6.24 (+0.93) | 69 (+2.4) |
| **Qwen3_1.7B_TIC_GRPO** | **11.77 (+2.60)** | **6.98 (+1.67)** | **69.8 (+3.2)** |
| Qwen3_8B_GRPO | 31.35 | 22.9 | 88.6 |
| Qwen3_8B_GSPO | 30.21 (-1.14) | 22.5 (-0.4) | 88.4 (-0.2) |
| **Qwen3_8B_TIC_GRPO** | **33.34 (+1.99)** | **24.12 (+1.22)** | **90 (+1.4)** |

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

CONTENTS

## A  EXPERIMENTS

### A.1  REMOVING IMPORTANCE SAMPLING

**Experimental Setup**

We conduct this experiment using the qwen3_1.7b-base model on a hybrid dataset comprising the full DAPO-17K corpus together with several hundred examples from the AIME benchmark. The model is trained for a single epoch, ensuring that each prompt is used exactly once. We employ a total batch size of 128 and a mini-batch size of 32, which means each sample is reused for four gradient updates before the old policy is refreshed. This configuration isolates the effect of removing importance sampling while maintaining stable optimization.

**Results and Discussion**

As illustrated in Figure 1, eliminating importance sampling causes no significant drop in performance. In fact, during the latter stages of training, we observe a slight performance gain. This outcome empirically supports our earlier claim that, because the divergence between the current policy $\pi_\theta$ and the old policy $\pi_{\theta_{\mathrm{old}}}$ remains limited within each update cycle, the policy gradient computed at $\pi_{\theta_{\mathrm{old}}}$ continues to provide a reliable update direction in practice.

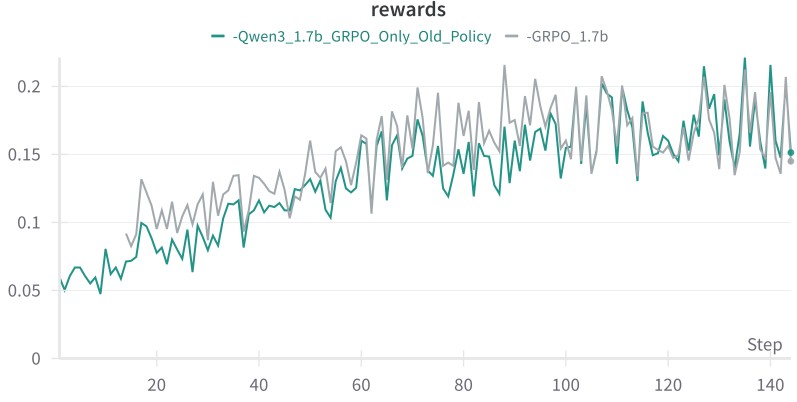

Figure 1: Experiment using policy gradient updates computed purely with the old policy $\pi_{\mathrm{old}}$.

### A.2  PRIMARY EMPIRICAL RESULTS

**Setup.**  Our training dataset combines the full DAPO-17K corpus with a subset of the AIME benchmark (1983–2022), resulting in a few hundred samples. We train the qwen3_1.7b-base model for a single epoch on an H200 GPU for over 24 hours. The total batch size is set to 128 and the mini-batch size to 32, so that each trajectory is reused for four gradient updates before refreshing the old policy. Similarly, the qwen3_8b model follows the same settings; although the physical batch differs, gradient accumulation ensures that the global batch size remains consistent. Training for qwen3_8b-base model is conducted on 2 H200 nodes over 48 hours. To eliminate confounding factors such as data-sampling randomness, we disable both Dynamic Sampling and the soft length penalty.

**Evaluation Figures.**  Figures 2 and 3 present the final evaluation performance of GRPO, GSPO, and our proposed TIC-GRPO on Qwen 1.7B and Qwen 8B models. To display the two evaluation plots side by side, we use a single figure environment:

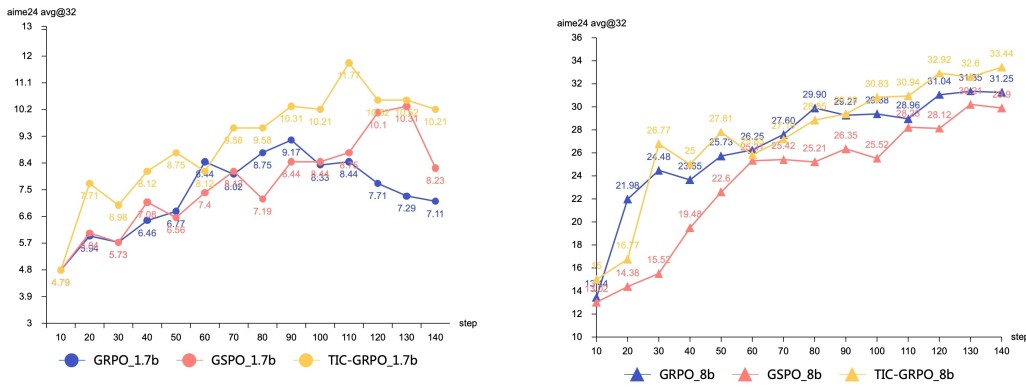

Figure 2: Evaluation performance on Qwen 1.7B

Figure 3: Evaluation performance on Qwen 8B

**Training Dynamics.** To further examine optimization behavior, Figures 4 and 5 show the training reward curve and critic score for Qwen 1.7B, while Figures 6 and 7 report the same metrics for Qwen 8B.

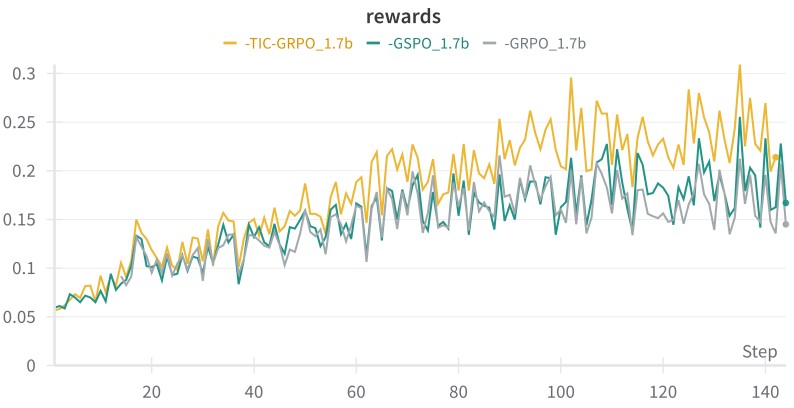

Figure 4: Training reward of GRPO, GSPO, and TIC-GRPO on Qwen 1.7B

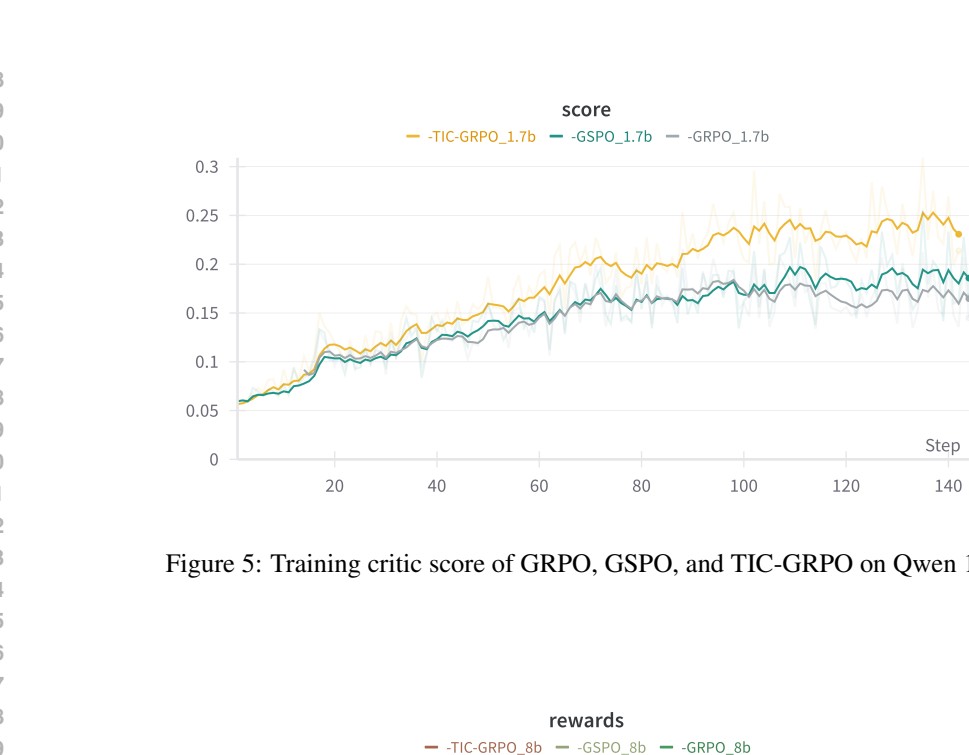

Figure 5: Training critic score of GRPO, GSPO, and TIC-GRPO on Qwen 1.7B

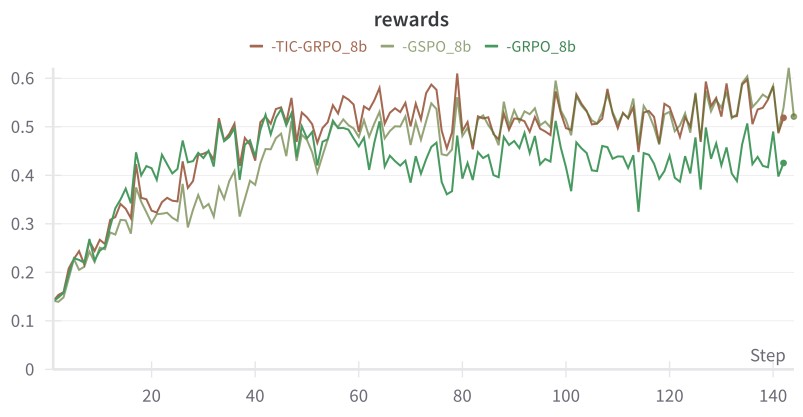

Figure 6: Training reward of GRPO, GSPO, and TIC-GRPO on Qwen 8B

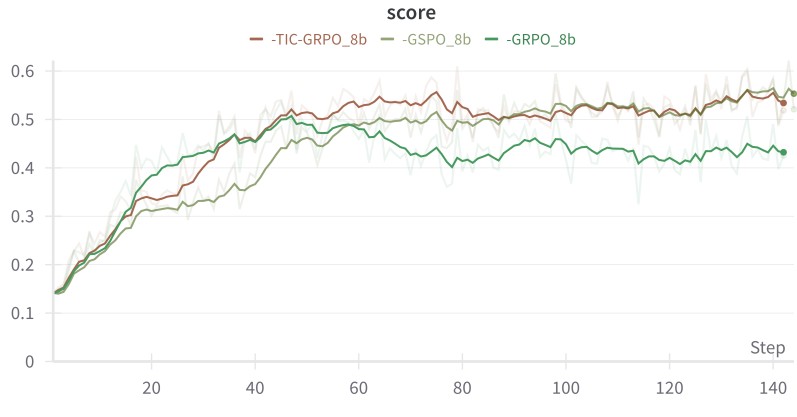

Figure 7: Training critic score of GRPO, GSPO, and TIC-GRPO on Qwen 8B

**Results.** Across both model sizes, TIC-GRPO not only converges faster but also delivers the highest final evaluation scores, consistently outperforming GRPO and GSPO. The side-by-side evaluation plots confirm that TIC-GRPO gains an early performance lead during training and maintains the top position through the entire evaluation phase, demonstrating stable superiority. This empirical evidence supports that our TIC-GRPO yields improved convergence and sustained best-in-class performance.

## A.3 ABLATION EXPERIMENTS

To further investigate the contribution of each modification in TIC-GRPO, we conduct an ablation study on the qwen3_1.7b-base model. Specifically, we evaluate two variants: (1) applying only response-level importance sampling, and (2) applying only upper-bound clipping with length-corrected group normalization. All other training settings remain identical to those used in the main experiment. As illustrated in the three plots in Figures 8–10, each modification individually improves convergence speed and final reward compared with the original GRPO baseline. These results demonstrate that both refinements are independently effective, while their combination in TIC-GRPO yields the strongest overall performance.

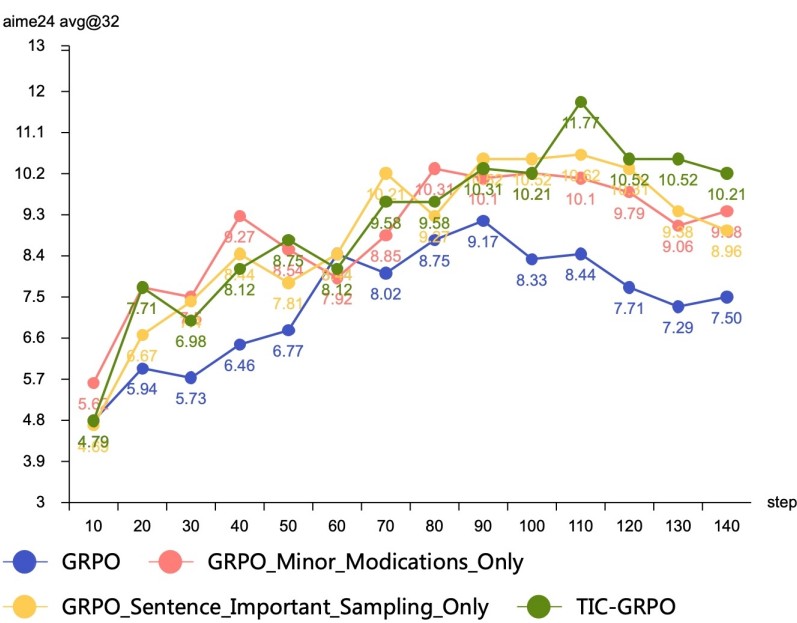

Figure 8: Evaluation performance of GRPO, GRPO_Sentence_Important_Sampling_Only, GRPO_Minor_Modications_Only and TIC-GRPO on Qwen 1.7B

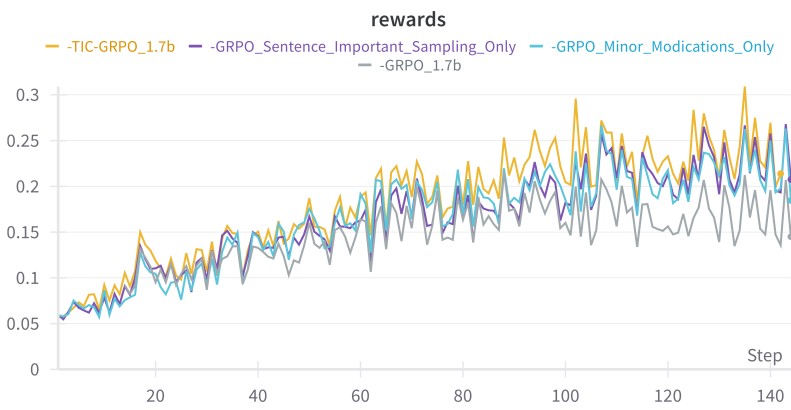

Figure 9: Training reward of GRPO, GRPO_Sentence_Important_Sampling_Only,GRPO_Minor_Modications_Only and TIC-GRPO on Qwen 1.7B

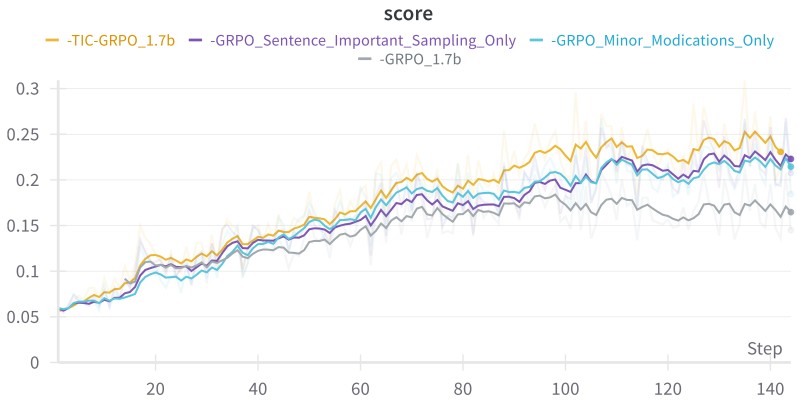

Figure 10: Training critic score of GRPO, GRPO_Sentence_Important_Sampling_Only,GRPO_Minor_Modications_Only and TIC-GRPO on Qwen 1.7B

## B  GRADIENT ESTIMATOR IN SECTION 4

Here the decomposition of $\nabla\mathcal{L}_{\text{TIC-GRPO}}(\theta, \theta_{\text{old}}, \theta_{\text{ref}})$ as an estimation of $\nabla J(\theta)$ is more involved than in the original GRPO. We now explain the reason. Analogous to the original GRPO, we may perform the following simple decomposition of $\tilde{\nabla} J(\theta)$:

$$\nabla\mathcal{L}_{\text{TIC-GRPO}}(\theta, \theta_{\text{old}}, \theta_{\text{ref}}) = \underbrace{\sum_{s_T \in \mathcal{S}_T} \xi_G(s_T) \frac{\mathbb{P}_\theta(s_T \mid s_0)}{\mathbb{P}_{\theta_{\text{old}}}(s_T \mid s_0)} \nabla \log \mathbb{P}_\theta(s_T|s_0) \frac{r(s_T)}{|s_T|}}_{\bar{\nabla} J(\theta)}$$

$$- \underbrace{\sum_{s_T \in \mathcal{S}_T} \xi_G(s_T) \mathbf{1}_{\mathcal{D}^c}(s_T, \theta, \theta_{\text{old}}) \frac{\mathbb{P}_\theta(s_T \mid s_0)}{\mathbb{P}_{\theta_{\text{old}}}(s_T \mid s_0)} \nabla \log \mathbb{P}_\theta(s_T|s_0) \left( \frac{r(s_T)}{|s_T|} - A_G(s_T) \right)}_{\Delta'(\theta, \theta_{\text{old}})}.$$

As in the original GRPO, we can prove that the error term $\Delta'(\theta, \theta_{\text{old}})$ is a controllable error term. However, it should be noted that $\bar{\nabla} J(\theta)$ is not a strictly unbiased estimator of $\nabla J(\theta)$. In particular,

we need to observe the following inequality:

$$\mathbb{E}\left[\bar{\nabla}J(\theta)\big|\mathscr{F}_{\theta_{\text{old}}}\right] = \mathbb{E}\left[\sum_{s_T \in \mathcal{S}_T} \xi_G(s_T)\frac{\mathbb{P}_\theta(s_T \mid s_0)}{\mathbb{P}_{\theta_{\text{old}}}(s_T \mid s_0)}\nabla\log\mathbb{P}_\theta(s_T|s_0)\frac{r(s_T)}{|s_T|}\bigg|\mathscr{F}_{\theta_{\text{old}}}\right]$$

$$\neq \sum_{s_T \in \mathcal{S}_T}\mathbb{E}\left[\xi_G(s_T)|\mathscr{F}_{\theta_{\text{old}}}\right]\left(\mathbb{E}\left[\frac{\mathbb{P}_\theta(s_T \mid s_0)}{\mathbb{P}_{\theta_{\text{old}}}(s_T \mid s_0)}\nabla\log\mathbb{P}_\theta(s_T|s_0)\frac{r(s_T)}{|s_T|}\bigg|\mathscr{F}_{\theta_{\text{old}}}\right]\right)$$

$$= \left(\sum_{s_T \in \mathcal{S}_T}\mathbb{P}_{\theta_{\text{old}}}(s_T \mid s_0)\right)\left(\mathbb{E}\left[\frac{\mathbb{P}_\theta(s_T \mid s_0)}{\mathbb{P}_{\theta_{\text{old}}}(s_T \mid s_0)}\nabla\log\mathbb{P}_\theta(s_T|s_0)\frac{r(s_T)}{|s_T|}\bigg|\mathscr{F}_{\theta_{\text{old}}}\right]\right)$$

$$= \mathbb{E}\left[\sum_{s_T \in \mathcal{S}_T}\mathbb{P}_{\theta_{\text{old}}}(s_T \mid s_0)\frac{\mathbb{P}_\theta(s_T \mid s_0)}{\mathbb{P}_{\theta_{\text{old}}}(s_T \mid s_0)}\nabla\log\mathbb{P}_\theta(s_T|s_0)\frac{r(s_T)}{|s_T|}\bigg|\mathscr{F}_{\theta_{\text{old}}}\right]$$

$$= \mathbb{E}\left[\nabla J(\theta)|\mathscr{F}_{\theta_{\text{old}}}\right].$$

The reason for the above inequality is that random variable $\boldsymbol{\xi}_G$ and the parameter $\theta$ are measurable with respect to the $\sigma$-algebra $\mathscr{F}_{\theta_{\text{old}},\boldsymbol{\xi}_G}$. Therefore, when taking the conditional expectation with respect to the sub-$\sigma$-algebra $\mathscr{F}_{\theta_{\text{old}}}$, we cannot simply move the random variable $\boldsymbol{\xi}_G$ outside of the conditional expectation.

Therefore, the asymptotic unbiased decomposition of $G(\theta, \theta_{\text{old}})$ is not immediate. We state the following:

$$\nabla\mathcal{L}_{\text{TIC-GRPO}}(\theta, \theta_{\text{old}}) = \frac{1}{\sigma'_{\theta_{\text{old}}}} \sum_{s_T \in \mathcal{S}_T} \xi_G(s_T) \mathbf{1}_{\mathcal{D}(s_T, \theta, \theta_{\text{old}})} \frac{\mathbb{P}_\theta(s_T \mid s_0)}{\mathbb{P}_{\theta_{\text{old}}}(s_T \mid s_0)} \nabla \log \mathbb{P}_\theta(s_T|s_0) \left( \frac{r(s_T)}{|s_T|} - \mu'_{\theta_{\text{old}}} \right)$$

$$+ \underbrace{\sum_{s_T \in \mathcal{S}_T} \xi_G(s_T) \mathbf{1}_{\mathcal{D}(s_t, \theta, \theta_{\text{old}})} \frac{\mathbb{P}_\theta(s_T \mid s_0)}{\mathbb{P}_{\theta_{\text{old}}}(s_T \mid s_0)} \nabla \log \mathbb{P}_\theta(s_T|s_0) B'_G(s_T)}_{\frac{1}{\sigma'_{\theta_{\text{old}}}} \Delta_s(\theta, \theta_{\text{old}})}$$

$$= \frac{1}{\sigma'_{\theta_{\text{old}}}} \sum_{s_T \in \mathcal{S}_T} \mathbb{P}_{\theta_{\text{old}}}(s_T|s_0) \mathbf{1}_{\mathcal{D}(s_T, \theta, \theta_{\text{old}})} \frac{\mathbb{P}_\theta(s_T \mid s_0)}{\mathbb{P}_{\theta_{\text{old}}}(s_T \mid s_0)} \nabla \log \mathbb{P}_\theta(s_T|s_0) \left( \frac{r(s_T)}{|s_T|} - \mu'_{\theta_{\text{old}}} \right)$$

$$+ \frac{1}{\sigma'_{\theta_{\text{old}}}} \sum_{s_T \in \mathcal{S}_T} \left( \xi_G(s_T) - \mathbb{P}_{\theta_{\text{old}}}(s_T|s_0) \right) \mathbf{1}_{\mathcal{D}(s_T, \theta, \theta_{\text{old}})} \frac{\mathbb{P}_\theta(s_T \mid s_0)}{\mathbb{P}_{\theta_{\text{old}}}(s_T \mid s_0)} \nabla \log \mathbb{P}_\theta(s_T|s_0) \left( \frac{r(s_T)}{|s_T|} - \mu'_{\theta_{\text{old}}} \right)$$

$$+ \frac{1}{\sigma'_{\theta_{\text{old}}}} \Delta_s(\theta, \theta_{\text{old}})$$

$$= \frac{1}{\sigma'_{\theta_{\text{old}}}} \underbrace{\sum_{s_T \in \mathcal{S}_T} \mathbb{P}_\theta(s_T \mid s_0) \nabla \log \mathbb{P}_\theta(s_T|s_0) \left( \frac{r(s_T)}{|s_T|} - \mu'_{\theta_{\text{old}}} \right)}_{\nabla J(\theta)}$$

$$+ \frac{1}{\sigma'_{\theta_{\text{old}}}} \underbrace{\left( - \sum_{s_T \in \mathcal{S}_T} \mathbf{1}_{\mathcal{D}^c(s_T, \theta, \theta_{\text{old}})} \mathbb{P}_\theta(s_T \mid s_0) \nabla \log \mathbb{P}_\theta(s_T|s_0) \left( \frac{r(s_T)}{|s_T|} - \mu'_{\theta_{\text{old}}} \right) \right)}_{\Delta_c(\theta, \theta_{\text{old}})}$$

$$+ \frac{1}{\sigma'_{\theta_{\text{old}}}} \underbrace{\sum_{s_T \in \mathcal{S}_T} \left( \xi_G(s_T) - \mathbb{P}_{\theta_{\text{old}}}(s_T|s_0) \right) \nabla \log \mathbb{P}_{\theta_{\text{old}}}(s_T \mid s_0) \left( \frac{r(s_T)}{|s_T|} - \mu'_{\theta_{\text{old}}} \right)}_{M_{\theta_{\text{old}}, 1}}$$

$$+ \frac{1}{\sigma'_{\theta_{\text{old}}}} \underbrace{\sum_{s_T \in \mathcal{S}_T} \left( \xi_G(s_T) - \mathbb{P}_{\theta_{\text{old}}}(s_T|s_0) \right) \mathbf{1}_{\mathcal{D}(s_T, \theta, \theta_{\text{old}})} \frac{\nabla \mathbb{P}_\theta(s_T \mid s_0) - \nabla \mathbb{P}_{\theta_{\text{old}}}(s_T \mid s_0)}{\mathbb{P}_{\theta_{\text{old}}}(s_T \mid s_0)} \left( \frac{r(s_T)}{|s_T|} - \mu'_{\theta_{\text{old}}} \right)}_{\Delta_{g,1}(\theta, \theta_{\text{old}})}$$

$$+ \frac{1}{\sigma'_{\theta_{\text{old}}}} \underbrace{\left( - \sum_{s_T \in \mathcal{S}_T} \left( \xi_G(s_T) - \mathbb{P}_{\theta_{\text{old}}}(s_T|s_0) \right) \mathbf{1}_{\mathcal{D}^c(s_T, \theta, \theta_{\text{old}})} \nabla \log \mathbb{P}_{\theta_{\text{old}}}(s_T \mid s_0) \left( \frac{r(s_T)}{|s_T|} - \mu'_{\theta_{\text{old}}} \right) \right)}_{\Delta_{g,2}(\theta, \theta_{\text{old}})}$$

$$+ \frac{1}{\sigma'_{\theta_{\text{old}}}} \Delta_s(\theta, \theta_{\text{old}}). \tag{15}$$

In the above expression, we define the following quantities:

$$\mu'_{\theta_{\text{old}}} := \mathbb{E}_{G \sim \pi_{\theta_{\text{old}}}} \left[ \mu'_G | \mathscr{F}_{\theta_{\text{old}}} \right]$$

$$\sigma'_{\theta_{\text{old}}} := \delta + \mathbb{E}_{G \sim \pi_{\theta_{\text{old}}}} \left[ \sigma'_G | \mathscr{F}_{\theta_{\text{old}}} \right]$$

$$B'_G(s_T) := A'_G(s_T) - \frac{1}{\sigma'_{\theta_{\text{old}}}} \left( \frac{r(s_T)}{|s_T|} - \mu'_{\theta_{\text{old}}} \right). \tag{16}$$

## C THEORETICAL ANALYSIS AND PROOFS

In this section, we present the complete proof of the theorem, including all necessary lemmas and their proofs, and provide a proof sketch for ease of reading. Because the proofs for GRPO and TIC-GRPO differ substantially, we organize them into two separate subsections.

### C.1 PROOF SKETCH OF GRPO

The core of the proof lies in constructing a descent lemma, namely evaluating $J^* - J(\theta_{n+1,0}) - \left(J^* - J(\theta_{n,0})\right)$, which follows the same principle as in classical gradient-descent methods such as SGD, MSGD, and Adam. Here $J^*$ denotes the theoretical maximum of the value function $J(\theta)$, as defined in Subsection 2.2. To expand $J^* - J(\theta_{n+1,0}) - \left(J^* - J(\theta_{n,0})\right)$, We first establish that the second derivative of $J(\theta_n)$ exists almost everywhere and is bounded. To this end, we state the following lemma, whose proof is provided in Subsection C.4.5.

**Lemma C.1.** *Assume that the conditions in Assumptions 5.1 and 5.2 hold. Then, for any $\theta, \theta' \in \mathbb{R}^d$, the following inequalities are satisfied:*

$$\left(J^* - J(\theta)\right) - \left(J^* - J(\theta')\right) \leq -\nabla J(\theta')^\top (\theta - \theta') + \frac{RL}{2}\left(2T\log|\mathcal{V}| + 1\right)\|\theta - \theta'\|^2,$$

$$\left\|\nabla J(\theta) - \nabla J(\theta')\right\| \leq RL\left(2T\log|\mathcal{V}| + 1\right)\|\theta - \theta'\|.$$

With the above expansion in hand, we substitute $\theta_{n+1,0}$ and $\theta_{n,0}$ for $\theta$ and $\theta'$ in Lemma C.1, which yields (informal version)

$$\left(J^* - J(\theta_{n+1,0})\right) - \left(J^* - J(\theta_{n,0})\right)$$

$$\stackrel{\text{Lemma C.1}}{\leq} -\frac{\eta}{2R}\sum_{s=0}^{K-1}\|\nabla J(\theta_{n,s})\|^2$$

$$+ \mathcal{O}\left(\|\nabla J(\theta_{n,s})\|\right)\mathcal{O}\left(\sum_{s=0}^{K-1}\left(\|\Xi_g(\theta_{n,s},\theta_{n,0})\| + \|\Xi_s(\theta_{n,s},\theta_{n,0})\| + \|\Xi_c(\theta_{n,s},\theta_{n,0})\|\right)\right)$$

$$+ \mathcal{O}\left(\|\theta_{n+1,0} - \theta_{n,0}\|^2\right) + \mathcal{O}\left(\|\theta_{n+1,0} - \theta_{n,0}\|\right).$$

In the above expression, the terms $\|\Xi_g(\theta_{n,s},\theta_{n,0})\|$, $\|\Xi_s(\theta_{n,s},\theta_{n,0})\|$, and $\|\Xi_c(\theta_{n,s},\theta_{n,0})\|$ are defined in Eq. 8.

We observe that the first term on the right-hand side of the inequality provides the desired descent term, while $\{M_n', \mathscr{F}_n\}$ forms a martingale difference sequence and therefore has zero contribution after taking expectations. Next, we bound $\|\nabla J(\theta_{n,s})\|$, $\|\Xi_g(\theta_{n,s},\theta_{n,0})\|$, $\|\Xi_s(\theta_{n,s},\theta_{n,0})\|$, $\|\Xi_c(\theta_{n,s},\theta_{n,0})\|$ and $\|\theta_{n+1,0} - \theta_{n,0}\|^p$, $(p = 1, 2)$ separately. To this end, we establish the following five lemmas, whose proofs are given in Subsections C.4.4, Subsection C.4.6, Subsection C.4.7, Subsection C.4.8 and Subsection C.4.9.

**Lemma C.2.** *Assume that the conditions in Assumptions 5.1 and 5.2 hold. Then, for any $\theta \in \mathbb{R}^d$, the following inequalities are satisfied:*

$$\|\nabla J(\theta)\| \leq \sqrt{2L}R\sqrt{\log|\mathcal{V}|}.$$

**Lemma C.3.** *Assume that the conditions in Assumptions 5.1 and 5.2 hold. Let $\theta_{1,0} \in \mathbb{R}^d$ be an arbitrary initialization of the algorithm, and let the sequence $\{\theta_{n,s}\}$ be generated by GRPO as defined in Eq. 12. Then the term $\|\Xi_g(\theta_{n,s},\theta_{n,0})\|$ satisfies the following upper bound:*

$$\mathbb{E}\left[\|\Xi_g(\theta_{n,s},\theta_{n,0})\| \,\big|\, \mathscr{F}_{n-1}\right] \leq 16RL\sqrt{\mathbb{E}\left[\mathcal{M}_N^2 \,\big|\, \mathscr{F}_{n-1}\right]} \cdot \sqrt{\mathbb{E}\left[\|\theta_{n,s} - \theta_{n,0}\|^2 \,\big|\, \mathscr{F}_{n-1}\right]},$$

*where $\mathcal{M}_N$ is defined in Eq. 14.*

**Lemma C.4.** *Assume that the conditions stated in Assumptions 5.1 and 5.2 are satisfied. Let $\theta_{1,0} \in \mathbb{R}^d$ denote an arbitrary initialization of the algorithm. Then the term $\|\Xi_g(\theta_{n,s},\theta_{n,0})\|$ admits the following upper bound:*

$$\mathbb{E}\left[\Xi_s(\theta_{n,s},\theta_{n,0})|\mathscr{F}_{n-1}\right] = \mathcal{O}\left(\frac{1}{\sqrt{|G|}}\right) + \mathcal{O}(\sigma_{s,\theta_{n,0}}^2),$$

*where $\mathcal{M}_n$ is defines in Eq. 14, and $\mathbb{E}_{s_T \sim \pi_{\theta_{n,0}}}\left[||s_T| - |s|_{\theta_{n,0}}^2|\mathscr{F}_{n-1}\right] := \sigma_{s,\theta_{n,0}}^2$.*

**Lemma C.5.** *Assume that the conditions stated in Assumptions 5.1 and 5.2 are satisfied. Let $\theta_{1,0} \in \mathbb{R}^d$ denote an arbitrary initialization of the algorithm. Then the term $\|\Xi_c(\theta_{n,s}, \theta_{n,0})\|$ admits the following upper bound:*

$$\mathbb{E}\left[\|\Xi_c(\theta_{n,s}, \theta_{n,0})\|\big|\mathscr{F}_{n-1}\right] \leq \frac{2RTL\sqrt{\log|\mathcal{V}|}}{\delta \min\{\epsilon_{low}, \epsilon_{high}\}} \mathbb{E}^{1/4}\left[\|\theta_{n,s} - \theta_{n,0}\|^4 \big|\mathscr{F}_{n-1}\right]\sqrt{\mathbb{E}\left[\mathcal{M}_N^2|\mathscr{F}_{n-1}\right]},$$

*where $\mathcal{M}_n$ is defines in Eq. 14.*

**Lemma C.6.** *Assume that the conditions stated in Assumptions 5.1 and 5.2 are satisfied. Let $\theta_{1,0} \in \mathbb{R}^d$ denote an arbitrary initialization of the algorithm. Then the term $\|\theta_{n,s+1} - \theta_{n,s}\|^p$, $p \in \{1, 2, 4\}$ admits the following upper bound:*

$$\mathbb{E}\left[\|\theta_{n,s+1} - \theta_{n,s}\|^p\big|\mathscr{F}_{n-1}\right] \leq \eta^p (2R)^p (2L)^{p/2}\sqrt{\mathbb{E}\left[\mathcal{M}_N^{2p}|\mathscr{F}_{n-1}\right]}\log^{p/2}|\mathcal{V}|.$$

*where $\mathcal{M}_n$ is defines in Eq. 14.*

By incorporating the bounds from the five lemmas into the preceding computations, we obtain the following descent inequality.

$$\mathbb{E}\left[J^* - J(\theta_{n+1,0})\right] - \mathbb{E}\left[J^* - J(\theta_{n,0})\right]$$

$$\leq -\frac{\eta}{2R}\sum_{s=0}^{K-1}\mathbb{E}\left[\|\nabla J(\theta_{n,s})\|^2\right] + \mathcal{O}\left(\log^2|\mathcal{V}|\sqrt{\mathbb{E}\left[\mathcal{M}_N^2\right]}\right)\eta^2 + \mathcal{O}\left(\frac{1}{\sqrt{|G|}}\right)\eta + \mathcal{O}(\bar{\sigma}_{s,\theta_{n,0}}^2)\eta.$$

Finally, summing the descent inequality over $n = 1$ to $N$ yields the result stated in Theorem 5.1.

## C.2 Proof sketch of TIC-GRPO

As in the GRPO analysis, the core is a descent argument on $J^* - J(\theta_{n+1,0}) - \left(J^* - J(\theta_{n,0})\right)$. We follow the same approach used in the GRPO analysis and apply Lemma C.1 to expand $J^* - J(\theta_{n+1,0}) - \left(J^* - J(\theta_{n,0})\right)$, obtaining the following expression:

$$\left(J^* - J(\theta_{n+1,0})\right) - \left(J^* - J(\theta_{n,0})\right) \leq -\frac{\eta}{2R}\sum_{s=0}^{K-1}\|\nabla J(\theta_{n,s})\|^2$$

$$+ \mathcal{O}\left(\|\theta_{n+1,0} - \theta_{n,0}\|^2\right) + \mathcal{O}\left(\sum_{s=0}^{K-1}\|\theta_{n,s} - \theta_{n,0}\|\right) + M_n$$

$$+ \mathcal{O}\left(\|\nabla J(\theta_{n,s})\|\right)\mathcal{O}\left(\sum_{s=0}^{K-1}\left(\|\Delta_{g,1}(\theta_{n,s}, \theta_{n,0})\| + \|\Delta_{g,2}(\theta_{n,s}, \theta_{n,0})\| + \|\Delta_c(\theta_{n,s}, \theta_{n,0})\|\right.\right.$$

$$\left.\left.+ \|\Delta_s(\theta_{n,s}, \theta_{n,0})\|\right)\right).$$

In the above expression, the terms $\|\Delta_{g,1}(\theta_{n,s}, \theta_{n,0})\|$, $\|\Delta_{g,2}(\theta_{n,s}, \theta_{n,0})\|$, $\|\Delta_c(\theta_{n,s}, \theta_{n,0})\|$, and $\|\Delta_s(\theta_{n,s}, \theta_{n,0})\|$ are defined in Eq. 15. We can reuse Lemma C.2 to estimate $\|\nabla J(\theta_{n,s})\|$; however, because the algorithmic update conditions have changed, we must re-estimate $\sum_{s=0}^{K-1}\|\theta_{n,s} - \theta_{n,0}\|^p$, ($p \in \{1, 2, 4\}$). At the same time, we also need to estimate $\|\Delta_{g,1}(\theta_{n,s}, \theta_{n,0})\|$, $\|\Delta_{g,2}(\theta_{n,s}, \theta_{n,0})\|$, $\|\Delta_c(\theta_{n,s}, \theta_{n,0})\|$ and $\|\Delta_s(\theta_{n,s}, \theta_{n,0})\|$. Similarly, the sequence $\{M_n, \mathscr{F}_n\}$ is a martingale difference sequence, and therefore its expectation is zero and does not affect the overall result. We complete the above estimates through five lemmas, whose full proofs can be found in Subsection C.4.11, Subsection C.4.12, Subsection C.4.13, Subsection C.4.14 and Subsection C.4.15.

**Lemma C.7.** *Assume that the conditions stated in Assumptions 5.1 and 5.2 are satisfied. Then the term $\|\Delta_{g,1}(\theta, \theta_{old})\|$ satisfies following inequality:*

$$\mathbb{E}\left[\|\Delta_{g,1}(\theta, \theta_{old})\|\big|\mathscr{F}_{\theta_{old}}\right] \leq 4RLT\left(1 + \frac{\epsilon_{high}}{\log(1 + \epsilon_{high})}\log|\mathcal{V}|\right)\mathbb{E}\left[\|\theta - \theta_{old}\|\big|\mathscr{F}_{\theta_{old}}\right]$$

$$+ 4RT^{3/2}L^{3/2}\frac{\epsilon_{high}}{\log(1 + \epsilon_{high})}\sqrt{\log|\mathcal{V}|}\,\mathbb{E}\left[\|\theta - \theta_{old}\|^2|\mathscr{F}_{\theta_{old}}\right].$$

**Lemma C.8.** *Assume that the conditions stated in Assumptions 5.1 and 5.2 are satisfied. Then the term $\|\Delta_{g,2}(\theta, \theta_{old})\|$ satisfies following inequality:*

$$\mathbb{E}\left[\|\Delta_{g,2}(\theta, \theta_{old})\| \,|\, \mathscr{F}_{\theta_{old}}\right] \leq \frac{4R}{\log(1 + \epsilon_{high})} \left[4T^2 L \log|\mathcal{V}| \sqrt{\mathbb{E}\left[\|\theta - \theta_{old}\|^2 |\mathscr{F}_{\theta_{old}}\right]} \right.$$
$$\left. + 2\sqrt{2} L^{3/2} T \sqrt{\log|\mathcal{V}|} \sqrt{\mathbb{E}\left[\|\theta - \theta_{old}\|^4 |\mathscr{F}_{\theta_{old}}\right]}\right].$$

**Lemma C.9.** *Assume that the conditions stated in Assumptions 5.1 and 5.2 are satisfied. Then the term $\|\Delta_c(\theta, \theta_{old})\|$ satisfies following inequality:*

$$\mathbb{E}\left[\|\Delta_c(\theta, \theta_{old})\||\mathscr{F}_{\theta_{old}}\right]$$
$$\leq \frac{4\sqrt{2}RLT^2}{\log(1 + \epsilon_{high})} \left(\sqrt{2} \log|\mathcal{V}| \sqrt{\mathbb{E}\left[\|\theta - \theta_{old}\|^2 |\mathscr{F}_{\theta_{old}}\right]} + \sqrt{L} \log|\mathcal{V}| \sqrt{\mathbb{E}\left[\|\theta - \theta_{old}\|^4 |\mathscr{F}_{\theta_{old}}\right]}\right).$$

**Lemma C.10.** *Assume that the conditions stated in Assumptions 5.1 and 5.2 are satisfied. Then the term $\|\Delta_s(\theta, \theta_{old})\|$ satisfies following inequality:*

$$\mathbb{E}\left[\|\Delta_{s,1}(\theta, \theta_{old})\||\mathscr{F}_{\theta_{old}}\right] \leq \mathcal{O}\left(\frac{1}{\sqrt{|G|}}\right).$$

**Lemma C.11.** *Assume that the conditions stated in Assumptions 5.1 and 5.2 are satisfied. Let $\theta_{1,0} \in \mathbb{R}^d$ denote an arbitrary initialization of the algorithm, and we set $\eta \leq \frac{1}{4K(1+\epsilon_{high})RL}$. The sequence $\{\theta_{n,s}\}$ generated by TIC-GRPO as defined in Eq. 13. Then the term $\sum_{s=1}^{K} \mathbb{E}\left[\|\theta_{n,s} - \theta_{n,0}\|^p |\mathscr{F}_{n-1}\right]$ $(p \in \{1, 2, 4\})$ admits the following upper bound:*

$$\sum_{s=1}^{K} \mathbb{E}\left[\|\theta_{n,s} - \theta_{n,0}\|^p |\mathscr{F}_{n-1}\right] \leq K^{p+1} \eta^p (2R)^p (1 + \epsilon_{high})^p 2^p (2L)^{p/2} \log^{p/2} |\mathcal{V}|.$$

By incorporating the bounds from the five lemmas into the preceding computations, we obtain the following descent inequality.

$$\mathbb{E}\left[J^* - J(\theta_{n+1,0})\right] - \mathbb{E}\left[J^* - J(\theta_{n,0})\right]$$
$$\leq -\frac{\eta}{2R} \sum_{s=0}^{K-1} \mathbb{E}\left[\|\nabla J(\theta_{n,s})\|^2\right] + \mathcal{O}\left(\log^2 |\mathcal{V}|\eta^2\right) + \mathcal{O}\left(\frac{1}{\sqrt{|G|}}\right) \eta.$$

Finally, summing the descent inequality over $n = 1$ to $N$ yields the result stated in Theorem 5.2.

## C.3 AUXILIARY LEMMAS

In this subsection, we provide the technical lemmas required for the proof.

**Lemma C.12** (Upper bound for $\sum_{i=1}^{n} x_i \log^2 x_i$)**.** *Let $x_1, \ldots, x_n \in (0, 1)$ satisfy $\sum_{i=1}^{n} x_i = 1$. Then*

$$\sum_{i=1}^{n} x_i \log^2 x_i \leq \begin{cases} \frac{28}{e^2} < 4, & \text{if } n \leq 7, \\ \log^2 n, & \text{if } n \geq 8. \end{cases}$$

**Lemma C.13** (Lemma C.2 of Jin et al. (2024))**.** *Suppose that $f(x)$ is differentiable and lower bounded, i.e. $f^* = \inf_{x \in \mathbb{R}^d} f(x) > -\infty$, and $\nabla f(x)$ is Lipschitz continuous with parameter $\mathcal{L} > 0$, then $\forall\, x \in \mathbb{R}^d$, we have*

$$\left\|\nabla f(x)\right\|^2 \leq 2\mathcal{L}\left(f(x) - f^*\right).$$

**Lemma C.14.** *Assume that the conditions in Assumptions 5.1 and 5.2 hold. Then, for any $\theta, \theta' \in \mathbb{R}^d$, the following inequalities are satisfied:*

$$\left|\|\nabla \log \mathbb{P}_\theta(s_T|s_0)\|^p - \|\nabla \log \mathbb{P}_{\theta'}(s_T|s_0)\|^p\right|$$
$$\leq \sum_{q=1}^{p} \binom{p}{q} 2^{\frac{p-q}{2}} (|s_T|L)^{\frac{3(p-q)}{2}} (-\log \mathbb{P}_{\theta'}(s_T|s_0))^{\frac{p-q}{2}} \|\theta - \theta'\|^q,$$

*and*

$$|\log \mathbb{P}_\theta(s_T|s_0) - \log \mathbb{P}_{\theta'}(s_T|s_0)| \leq \|\nabla \log \mathbb{P}_{\theta'}(s_T|s_0)\|\|\theta - \theta'\| + |s_T|L\|\theta - \theta'\|^2.$$

**Lemma C.15.** *Assume that the conditions in Assumptions 5.1 and 5.2 hold. Then, for any $\theta \in \mathbb{R}^d$, the following inequalities are satisfied:*

$$\|\nabla \mathbb{P}_\theta(s_T|s_0)\| \leq \sqrt{2|s_T|L}e.$$

### C.4  COMPLETE PROOFS

In this subsection, we provide the complete proofs of all lemmas and the main theorem.

#### C.4.1  THE PROOF OF LEMMA C.12

*Proof.* We consider the univariate function $f(x) := x \log^2 x$ for $x \in (0,1)$. Then our objective can clearly be written as

$$\sum_{i=1}^n f(x_i), \qquad \text{subject to} \quad \sum_{i=1}^n x_i = 1.$$

It is easy to verify the limit:

$$\lim_{x \to 0^+} x \log^2 x = 0.$$

We now analyze the monotonicity of the function $f(x)$. To this end, we compute its derivative as follows:

$$f'(x) = \log^2 x + 2 \log x.$$

Based on this, we can easily prove that $f(x)$ can be upper bounded by a piecewise function, i.e.,

$$f(x) \leq g(x) := \begin{cases} f(x), & x \in (0, e^{-2}], \\ \frac{4}{e^2}, & x \in (e^{-2}, 1). \end{cases} \tag{17}$$

It is easy to verify that $g(x)$ is continuously differentiable of first order, and its derivative is given by:

$$g'(x) = \begin{cases} \log^2 x + 2 \log x, & x \in (0, e^{-2}], \\ \frac{4}{e^2}, & x \in (e^{-2}, 1). \end{cases}$$

Furthermore, it is straightforward to show that $g''(x) \leq 0$ for all $x \in (0, e^{-2})$, i.e.,

$$g''(x) = \frac{2(\log x + 1)}{x} \leq 0, \qquad \forall\, x \in (0, e^{-2}).$$

In addition, since $g'(x) = 0$ for all $x \in [e^{-2}, 1]$, and $g''(x) \leq 0$ on $(0,1)$, we conclude that $g'(x)$ is monotonically decreasing over $(0,1)$. This implies that $g(x)$ is concave on the interval $(0,1)$. We can therefore use this property to estimate our objective as follows:

$$\sum_{i=1}^n f(x_i) \overset{\text{Eq. 17}}{\leq} \sum_{i=1}^n g(x_i) \overset{\text{Jensen's inequality}}{\leq} ng\left(\frac{\sum_{i=1}^n x_i}{n}\right) = ng\left(\frac{1}{n}\right).$$

According to the definition of $g$ in Eq. 17, it is easy to verify that when $n \leq 7$, we have

$$ng\left(\tfrac{1}{n}\right) \leq \frac{28}{e^2} < 4.$$

On the other hand, for $n \geq 8$, we observe that

$$ng\left(\tfrac{1}{n}\right) = nf\left(\tfrac{1}{n}\right) = \log^2 n.$$

With this, we complete the proof. $\qquad\square$

### C.4.2 THE PROOF OF LEMMA C.14

*Proof.* For any arbitrary trajectory $\{s_t\}_{t=0}^T$, we have:

$$\left| \|\nabla \log \mathbb{P}_\theta(s_T|s_0)\| - \|\nabla \log \mathbb{P}_{\theta'}(s_T|s_0)\| \right| \leq \|\nabla \log \mathbb{P}_\theta(s_T|s_0) - \nabla \log \mathbb{P}_{\theta'}(s_T|s_0)\|$$

$$= \left\| \sum_{t=1}^T \left( \nabla \log \mathbb{P}_\theta(s_t|s_{t-1}) - \nabla \log \mathbb{P}_{\theta'}(s_t|s_{t-1}) \right) \right\|$$

$$\leq \sum_{t=1}^T \|\nabla \log \mathbb{P}_\theta(s_t|s_{t-1}) - \nabla \log \mathbb{P}_{\theta'}(s_t|s_{t-1})\|$$

$$= \sum_{t=1}^T \|\nabla \log \mathbb{P}_\theta(s_t|s_{t-1}) - \nabla \log \mathbb{P}_{\theta'}(s_t|s_{t-1})\|$$

$$\leq |s_T| L \|\theta - \theta'\|. \tag{18}$$

Then for any $p \in \mathbb{Z}_+$, it is clear that:

$$\left| \|\nabla \log \mathbb{P}_\theta(s_T|s_0)\|^p - \|\nabla \log \mathbb{P}_{\theta'}(s_T|s_0)\|^p \right|$$

$$= \left| \|\nabla \log \mathbb{P}_{\theta'}(s_T|s_0) + (\nabla \log \mathbb{P}_\theta(s_T|s_0) - \nabla \log \mathbb{P}_{\theta'}(s_T|s_0))\|^p - \|\nabla \log \mathbb{P}_{\theta'}(s_T|s_0)\|^p \right|$$

$$\overset{\text{The } \textit{Binomial} \text{ theorem}}{=} \sum_{q=0}^p \binom{p}{q} \|\nabla \log \mathbb{P}_{\theta'}(s_T|s_0)\|^{p-q} \|\nabla \log \mathbb{P}_\theta(s_T|s_0) - \nabla \log \mathbb{P}_{\theta'}(s_T|s_0)\|^q$$

$$- \|\nabla \log \mathbb{P}_{\theta'}(s_T|s_0)\|^p$$

$$= \sum_{q=1}^p \binom{p}{q} \|\nabla \log \mathbb{P}_{\theta'}(s_T|s_0)\|^{p-q} \|\nabla \log \mathbb{P}_\theta(s_T|s_0) - \nabla \log \mathbb{P}_{\theta'}(s_T|s_0)\|^q$$

$$\overset{\text{Lemma C.13}}{\leq} \sum_{q=1}^p \binom{p}{q} (2|s_T|L)^{\frac{p-q}{2}} (-\log \mathbb{P}_{\theta'}(s_T|s_0))^{\frac{p-q}{2}} \|\nabla \log \mathbb{P}_\theta(s_T|s_0) - \nabla \log \mathbb{P}_{\theta'}(s_T|s_0)\|^q$$

$$\overset{\text{Eq. 18}}{\leq} \sum_{q=1}^p \binom{p}{q} 2^{\frac{p-q}{2}} (|s_T|L)^{\frac{3(p-q)}{2}} (-\log \mathbb{P}_{\theta'}(s_T|s_0))^{\frac{p-q}{2}} \|\theta - \theta'\|^q.$$

Next, we focus on the second inequality, for which we have

$$|\log \mathbb{P}_\theta(s_T|s_0) - \log \mathbb{P}_{\theta'}(s_T|s_0)|$$

$$\overset{(*)}{=} \left| \nabla \log \mathbb{P}_{\theta''}(s_T|s_0)^\top (\theta - \theta') \right|$$

$$= \left| \nabla \log \mathbb{P}_{\theta'}(s_T|s_0)^\top (\theta - \theta') + (\nabla \log \mathbb{P}_{\theta'}(s_T|s_0) - \nabla \log \mathbb{P}_{\theta''}(s_T|s_0))^\top (\theta - \theta') \right|$$

$$\leq \|\nabla \log \mathbb{P}_{\theta'}(s_T|s_0)\| \|\theta - \theta'\| + |s_T| L \|\theta - \theta'\|^2.$$

In the above derivation, step $(*)$ follows from the *Lagrange's mean value* theorem, where $\theta''$ denotes some point lying between $\theta$ and $\theta'$.

With this, we complete the proof. $\square$

### C.4.3 THE PROOF OF LEMMA C.15

*Proof.* For an arbitrary trajectory $\{s_t\}_{t=0}^T$, the following differentiation holds:

$$\|\nabla \mathbb{P}_\theta(s_T|s_0)\| = \mathbb{P}_\theta(s_T|s_0) \|\nabla \log \mathbb{P}_\theta(s_T|s_0)\|$$

$$\overset{\text{Lemma C.13, C.14}}{\leq} -\sqrt{2|s_T|L} \, \mathbb{P}_\theta(s_T|s_0) \log \mathbb{P}_\theta(s_T|s_0)$$

$$\leq \sqrt{2|s_T|L} \, e.$$

With this, we complete the proof.

$\square$

### C.4.4 THE PROOF OF LEMMA C.2

*Proof.* From Eq. 3, we know that

$$\nabla J(\theta) = \sum_{s_T \in \mathcal{S}_T} \mathbb{P}_\theta(s_T|s_0) \nabla \log \mathbb{P}_\theta(s_T|s_0) \frac{r(s_T)}{|s_T|},$$

which means following inequality:

$$\|\nabla J(\theta)\| \le R \sum_{s_T \in \mathcal{S}_T} \mathbb{P}_\theta(s_T|s_0) \frac{1}{|s_T|} \|\nabla \log \mathbb{P}_\theta(s_T|s_0)\|$$

$$\overset{(*)}{\le} \sqrt{2L}R \sum_{s_T \in \mathcal{S}_T} \frac{1}{\sqrt{|s_T|}} \mathbb{P}_\theta(s_T|s_0) \sqrt{-\log \mathbb{P}_\theta(s_T|s_0)}$$

$$\overset{\text{Jensen's inequality}}{\le} \sqrt{2L}R\sqrt{\log |\mathcal{V}|}.$$

Here, step $(*)$ follows from Lemma C.14, which guarantees that $\nabla \log \mathbb{P}_\theta(s_T|s_0)$ is Lipschitz continuous with constant $|s_T|L$. Applying Lemma C.13, we then obtain

$$\|\nabla \log \mathbb{P}_\theta(s_T|s_0)\| \le \sqrt{2|s_T|L}\sqrt{-\log \mathbb{P}_\theta(s_T|s_0)}.$$

With this, we complete the proof.

$\square$

### C.4.5 THE PROOF OF LEMMA C.1

*Proof.* First, for any trajectory $\{s_t\}_{t=0}^T$, by Lemma C.14, the mapping

$$\theta \mapsto \nabla_\theta \log \mathbb{P}_\theta(s_t \mid s_t)$$

is $|s_T|L$-Lipschitz continuous on $\mathbb{R}^d$, where $|s_T|L > 0$ is the Lipschitz constant. By *Rademacher's theorem*, any Lipschitz mapping from $\mathbb{R}^d$ to $\mathbb{R}^d$ is differentiable almost everywhere (a.e.) in $\mathbb{R}^d$. Therefore, $\nabla \log \mathbb{P}_\theta(s_T \mid s_0)$ is differentiable a.e., and $\nabla^2 \log \mathbb{P}_\theta(s_T \mid s_0)$ exists for almost every $\theta$.

Moreover, the Lipschitz continuity implies the uniform bound

$$\left\|\nabla^2 \log \mathbb{P}_\theta(s_t \mid s_{t-1})\right\| \le L \quad \text{a.e.} \tag{19}$$

Then, we construct the following univariate function:

$$f_{s_T}(\tau) := \mathbb{P}_{\theta(\tau)}(s_t|s_{t-1}),$$
$$\theta(t) := \theta' + (\theta - \theta')\tau, \ (\tau \in [0,1]).$$

Intuitively, this can be interpreted as the value at a point along the line segment connecting $\mathbb{P}_\theta(s_T \mid s_0)$ and $\mathbb{P}_{\theta'}(s_T \mid s_0)$, parameterized by the ratio between its distance to $\theta'$ and the total distance $\|\theta - \theta'\|$.

It is clear that $f_{s_T}$ is differentiable on $(0,1)$. In particular, its derivative can be computed as

$$f'_{s_T}(t) = (\theta - \theta')^\top \nabla \mathbb{P}_{\theta(t)}(s_T|s_0)$$
$$= (\theta - \theta')^\top \left(\mathbb{P}_{\theta(t)}(s_T|s_0)\nabla \log \mathbb{P}_{\theta(t)}(s_T|s_0)\right)$$
$$= f_{s_T}(t)(\theta - \theta')^\top \nabla \log \mathbb{P}_{\theta(t)}(s_T|s_0).$$

According to Lemma C.15, we have

$$|f'_{s_T}(t)| \le \sqrt{2|s_T|L}e,$$

which implies that $f_{s_T}(t)$ is absolutely continuous on $[0,1]$. On the other hand, by Lemma C.14, we know that

$$\nabla \log_{\theta(t)}\left(s_T \mid s_0\right)$$

is Lipschitz continuous on $[0,1]$ with Lipschitz constant $|s_T|L$, which implies that $(\theta - \theta')^\top \nabla \log_{\theta(t)}(s_T \mid s_0)$ is also absolutely continuous on $[0,1]$. Therefore, as the product of $f_{s_T}(t)$ and $\nabla \log_{\theta(t)}(s_T \mid s_0)$,

$$f'_{s_T}(t) := f_{s_T}(t) (\theta - \theta')^\top \nabla \log \mathbb{P}_{\theta(t)}(s_T|s_0)$$

is absolutely continuous on $[0,1]$. By the *Second Fundamental Theorem of Calculus for Absolutely Continuous Functions*, $f''_{s_T}(t)$ exists in $[0,1]$ almost everywhere, i.e.,

$$\begin{aligned} f''_{s_T}(t) = & f_{s_T}(t) \left( (\theta - \theta')^\top \nabla \log \mathbb{P}_{\theta(t)}(s_T|s_0) \right)^2 \\ & + f_{s_T}(t) (\theta - \theta')^\top \nabla^2 \log \mathbb{P}_{\theta(t)}(s_T|s_0) (\theta - \theta') \quad \text{a.e.,} \end{aligned} \tag{20}$$

and the following *Newton–Leibniz* formula holds in the sense of the Lebesgue integral

$$f'_{s_T}(1) - f'_{s_T}(0) = \int_0^1 f''_{s_T}(t) \mathrm{d}t. \tag{21}$$

We then compute $f_{s_T}(1) - f_{s_T}(0)$. Specifically, we have the following:

$$\begin{aligned} f_{s_T}(1) - f_{s_T}(0) &= \int_0^1 f'_{s_T}(t) \mathrm{d}t \\ &= \int_0^1 f'_{s_T}(0) \mathrm{d}t + \int_0^1 (f'_{s_T}(t) - f'_{s_T}(0)) \mathrm{d}t \\ &\overset{\text{Eq. 21}}{=} f'_{s_T}(0) + \int_0^1 \mathrm{d}t \int_0^t f''_{s_T}(s) \mathrm{d}s \\ &\overset{\text{Eq. 20}}{=} f'_{s_T}(0) + \int_0^1 \mathrm{d}t \int_0^t \left( f_{s_T}(s) \left( (\theta - \theta')^\top \nabla \log \mathbb{P}_{\theta(s)}(s_T|s_0) \right)^2 \right) \mathrm{d}s \\ &\quad + \int_0^1 \mathrm{d}t \int_0^t \left( f_{s_T}(s) (\theta - \theta')^\top \nabla^2 \log \mathbb{P}_{\theta(s)}(s_T|s_0) (\theta - \theta') \right) \mathrm{d}s \\ &\geq \mathbb{P}_{\theta'}(s_T|s_0) (\theta - \theta')^\top \nabla \log \mathbb{P}_{\theta'}(s_T|s_0) \\ &\quad - \int_0^1 \mathrm{d}t \int_0^t \mathbb{P}_{\theta(s)}(s_T|s_0) \|\theta - \theta'\|^2 \|\nabla \log \mathbb{P}_{\theta(s)}(s_T|s_0)\|^2 \mathrm{d}s \\ &\quad - \int_0^1 \mathrm{d}t \int_0^t \mathbb{P}_{\theta(s)}(s_T|s_0) \|\theta - \theta'\|^2 \|\nabla^2 \log \mathbb{P}_{\theta(t)}(s_T|s_0)\| \mathrm{d}s \\ &\overset{\text{Eq. 19}}{\geq} \mathbb{P}_{\theta'}(s_T|s_0) (\theta - \theta')^\top \nabla \log \mathbb{P}_{\theta'}(s_T|s_0) \\ &\quad - \int_0^1 \mathrm{d}t \int_0^t \mathbb{P}_{\theta(s)}(s_T|s_0) \|\theta - \theta'\|^2 \|\nabla \log \mathbb{P}_{\theta(s)}(s_T|s_0)\|^2 \mathrm{d}s \\ &\quad - |s_T|L \int_0^1 \mathrm{d}t \int_0^t \mathbb{P}_{\theta(s)}(s_T|s_0) \|\theta - \theta'\|^2 \mathrm{d}s. \end{aligned} \tag{22}$$

According to Eq. 2, we know the following expression for the value function:

$$J(\theta) = \mathbb{E}_{s_T \sim \pi_\theta} \left[ \frac{r(s_T)}{|s_T|} \right] = \sum_{s_T \in \mathcal{S}_T} \mathbb{P}_\theta(s_T|s_0) \frac{r(s_T)}{|s_T|}.$$

Then we calculate $(J^* - J(\theta)) - (J^* - J(\theta'))$, acquiring:

$$(J^* - J(\theta)) - (J^* - J(\theta')) = - \sum_{s_T \in \mathcal{S}_T} \left( \mathbb{P}_\theta(s_T|s_0) - \mathbb{P}_{\theta'}(s_T|s_0) \right) \frac{r(s_T)}{|s_T|}$$

$$= - \sum_{s_T \in \mathcal{S}_T} \left( f_{s_T}(1) - f_{s_T}(0) \right) \frac{r(s_T)}{|s_T|}$$

$$\overset{\text{Eq. 22}}{\leq} - \sum_{s_T \in \mathcal{S}_T} \mathbb{P}_{\theta'}(s_T|s_0) \left( \theta - \theta' \right)^\top \nabla \log \mathbb{P}_{\theta'}(s_T|s_0) \frac{r(s_T)}{|s_T|}$$

$$+ R \sum_{s_T \in \mathcal{S}_T} \frac{1}{|s_T|} \left( \int_0^1 \mathrm{d}t \int_0^t \mathbb{P}_{\theta(s)}(s_T|s_0) \|\theta - \theta'\|^2 \|\nabla \log \mathbb{P}_{\theta(s)}(s_T|s_0)\|^2 \mathrm{d}s \right)$$

$$+ RL \sum_{s_T \in \mathcal{S}_T} \left( \int_0^1 \mathrm{d}t \int_0^t \mathbb{P}_{\theta(s)}(s_T|s_0) \|\theta - \theta'\|^2 \, \mathrm{d}s \right)$$

$$= -\nabla J(\theta')^\top (\theta - \theta')$$

$$+ R \int_0^1 \mathrm{d}t \int_0^t \sum_{s_T \in \mathcal{S}_T} \frac{1}{|s_T|} \left( \mathbb{P}_{\theta(s)}(s_T|s_0) \|\theta - \theta'\|^2 \|\nabla \log \mathbb{P}_{\theta(s)}(s_T|s_0)\|^2 \mathrm{d}s \right)$$

$$+ \frac{RL}{2} \|\theta - \theta'\|^2$$

$$\overset{\text{Lemma C.13}}{\leq} -\nabla J(\theta')^\top (\theta - \theta')$$

$$+ 2RL \|\theta - \theta'\|^2 \int_0^1 \mathrm{d}t \int_0^t \sum_{s_T \in \mathcal{S}_T} - \left( \mathbb{P}_{\theta(s)}(s_T|s_0) \log \mathbb{P}_{\theta(s)}(s_T|s_0) \mathrm{d}s \right)$$

$$+ \frac{RL}{2} \|\theta - \theta'\|^2$$

$$\overset{\textit{Jensen's} \text{ inequality}}{\leq} -\nabla J(\theta')^\top (\theta - \theta') + 2TL \|\theta - \theta'\|^2 \int_0^1 \mathrm{d}t \int_0^t \log |\mathcal{V}| \mathrm{d}s$$

$$+ \frac{RL}{2} \|\theta - \theta'\|^2$$

$$= -\nabla J(\theta')^\top (\theta - \theta') + \frac{RL}{2} \left( 2T \log |\mathcal{V}| + 1 \right) \|\theta - \theta'\|^2. \tag{23}$$

At this point, the proof of the first inequality is complete. We now proceed to establish the second inequality.

For the gradient $\nabla J(\theta)$, we denote its $i$-th component by $(\nabla J(\theta))_i$. Next, for an arbitrary index $i$ and points $\theta$, $\theta'$, we construct the following univariate function:

$$g_i(t) := (\nabla J(\theta(t)))_i,$$
$$\theta(t) := \theta' + (\theta - \theta')t, \quad t \in [0, 1].$$

We define $[\nabla^2 J(\theta)]_i$ as the $i$-th row of the Hessian matrix $\nabla^2 J(\theta)$. Then it is straightforward to see that

$$g_i'(t) = [\nabla^2 J(\theta(t))]_i (\theta - \theta') \text{ a.e.}$$

Analogous to the derivation from Eq. 20 to Eq. 21, we can also establish the following *Newton–Leibniz* formula:

$$g_i(1) - g_i(0) = \int_0^1 g_i'(t)\mathrm{d}t = \int_0^1 [\nabla^2 J(\theta(t))]_i (\theta - \theta')\mathrm{d}t$$

By combining all the components, we obtain

$$\nabla J(\theta) - \nabla J(\theta') = \int_0^1 \nabla^2 J(\theta(t))(\theta - \theta')\mathrm{d}t.$$

Taking norms on both sides and applying the derivation in Eq. 23, we obtain

$$\|\nabla J(\theta) - \nabla J(\theta')\| \leq RL \left( 2T \log |\mathcal{V}| + 1 \right) \|\theta - \theta'\|.$$

With this, we complete the proof. $\qquad\square$

#### C.4.6 THE PROOF OF LEMMA C.3

*Proof.* First, by the definition of $\Xi_g(\theta_{n,s}, \theta_{n,0})$, we obtain

$$\|\Xi_g(\theta_{n,s}, \theta_{n,0})\| \leq 2R\mathcal{M}_N \sum_{s_T \in \mathcal{S}_T} \xi_G(s_T) \sum_{t=1}^{T} \|\nabla \mathbb{P}_{\theta_{n,s}}(s_t|s_{t-1}) - \nabla \mathbb{P}_{\theta_{n,0}}(s_t|s_{t-1})\|$$

$$\overset{(*)}{\leq} 16RL\mathcal{M}_N \|\theta_{n,s} - \theta_{n,0}\|. \tag{24}$$

In step $(*)$ we use the following derivation:

$$\|\nabla \mathbb{P}_{\theta_{n,s}}(s_t|s_{t-1}) - \nabla \mathbb{P}_{\theta_{n,0}}(s_t|s_{t-1})\|$$

$$\overset{(**)}{\leq} \left(\mathbb{P}_{\theta_\zeta}(s_t|s_{t-1})\|\nabla \mathbb{P}_{\theta(\zeta)}(s_t|s_{t-1})\|^2 + L\mathbb{P}_{\theta_\zeta}(s_t|s_{t-1})\right)\|\theta_{n,s} - \theta_{n,0}\|$$

$$\leq \left(2L\left(\mathbb{P}_{\theta_\zeta}(s_T|s_{t-1})(-\log \mathbb{P}_{\theta_\zeta}(s_t|s_{t-1})) + L\right)\right)\|\theta_{n,s} - \theta_{n,0}\|$$

$$\leq 8L\|\theta_{n,s} - \theta_{n,0}\|$$

In step $(**)$, $\theta_\zeta$ denotes a point lying between $\theta_{n,s}$ and $\theta_{n,0}$. Here we apply the mean value theorem for integrals in the sense of Lebesgue integration. The reader may refer to Lemma C.1 for the treatment of $\mathbb{P}_{\theta_{n,s}}(s_T \mid s_0)$, which we do not repeat here.

Taking the conditional expectation with respect to $\mathscr{F}_{n-1}$ on both sides of Eq. 24, we obtain

$$\mathbb{E}\left[\|\Xi_g(\theta_{n,s}, \theta_{n,0})\| \big| \mathscr{F}_{n-1}\right] \leq 16RL\,\mathbb{E}\left[\mathcal{M}_N\|\theta_{n,s} - \theta_{n,0}\| \big| \mathscr{F}_{n-1}\right]$$

$$\leq 16RL\sqrt{\mathbb{E}\left[\mathcal{M}_N^2|\mathscr{F}_{n-1}\right]} \cdot \sqrt{\mathbb{E}\left[\|\theta_{n,s} - \theta_{n,0}\|^2|\mathscr{F}_{n-1}\right]}.$$

With this, we complete the proof. $\qquad\square$

#### C.4.7 THE PROOF OF LEMMA C.4

*Proof.* From Eq. 9, we obtain

$$B_G(s_T) = \frac{A_G(s_T)}{|s_T|} - \frac{1}{\sigma_{\theta_{\text{old}}}} \frac{r(s_T)}{|s_T|}$$

$$= \frac{1}{|s_T|}\left(\frac{r(s_T) - \mu_G}{\sigma_G + \delta}\right) - \frac{1}{\sigma_{\theta_{\text{old}}}} \frac{r(s_T)}{|s_T|}$$

$$= \frac{r(s_T)}{|s_T|}\underbrace{\left(\frac{1}{\sigma_G + \delta} - \frac{1}{\sigma_{\theta_{\text{old}}}}\right)}_{B_1} - \frac{1}{|s_T|}\frac{\mu_G}{\sigma_G + \delta}$$

$$= \frac{r(s_T)}{|s_T|}B_1 - \frac{1}{|s_T|}\underbrace{\left(\frac{\mu_G}{\sigma_G + \delta} - \frac{\mu_{\theta_{\text{old}}}}{\sigma_{\theta_{\text{old}}}}\right)}_{B_2} - \frac{\mu_{\theta_{\text{old}}}}{|s_T|\sigma_{\theta_{\text{old}}}}$$

$$= \frac{r(s_T)}{|s_T|}B_1 - \frac{1}{|s_T|}B_2 - \underbrace{\left(\frac{1}{|s_T|} - |s|_{\theta_{\text{old}}}^{-1}\right)}_{B_3}\frac{\mu_{\theta_{\text{old}}}}{\sigma_{\theta_{\text{old}}}} - |s|_{\theta_{\text{old}}}^{-1}\frac{\mu_{\theta_{\text{old}}}}{\sigma_{\theta_{\text{old}}}}.$$

In the above expression, we set

$$|s|_{\theta_{\text{old}}} := \sum_{s_T \in \mathcal{S}_T} \mathbb{P}_{\theta_{\text{old}}}(s_T|s_0)|s_T|$$

Next we treat $B_1$, $B_2$, and $B_3$ separately. First, for $B_1$, we have:

$$\mathbb{E}\left[B_1^2|\mathscr{F}_{\theta_{\text{old}}}\right] \leq \frac{1}{\delta^4}\,\mathbb{E}\left[\left(\sigma_G - \mathbb{E}\left[\sigma_G|\mathscr{F}_{\theta_{\text{old}}}\right]\right)^2 \big| \mathscr{F}_{\theta_{\text{old}}}\right] \leq \frac{16R^2}{\delta^4\sqrt{|G|}}$$

Similarly, we readily obtain

$$\mathbb{E}\left[B_2^2|\mathscr{F}_{\theta_{\text{old}}}\right] \leq \frac{4R^2}{\delta^2|G|} + \frac{32R^3}{\delta^4\sqrt{|G|}}.$$

For $B_3$, we have

$$\mathbb{E}\left[B_3^2|\mathscr{F}_{\theta_{\text{old}}}\right] \leq \mathbb{E}_{s_T \sim \pi_{\theta_{\text{old}}}}\left[||s_T| - |s|^2_{\theta_{\text{old}}}|\mathscr{F}_{\theta_{\text{old}}}\right] := \sigma^2_{s,\theta_{\text{old}}}.$$

This quantity represents the variance of the sentence length under the policy $\pi_{\theta_{\text{old}}}$.

We now compute $\mathbb{E}\left[\Xi_s(\theta_{n,s}, \theta_{n,0})|\mathscr{F}_{n-1}\right]$ directly, and we have

$$\mathbb{E}\left[\Xi_s(\theta_{n,s}, \theta_{n,0})|\mathscr{F}_{n-1}\right] = \sum_{s_T \in \mathcal{S}_T} \mathbb{E}\left[\xi_G(s_T)\sum_{t=1}^{T}\nabla\log\mathbb{P}_{\theta_{n,0}}(s_t|s_{t-1})B_G(s_T)\Big|\mathscr{F}_{n-1}\right]$$

$$= \sum_{s_T \in \mathcal{S}_T} \mathbb{E}\left[\xi_G(s_T)\sum_{t=1}^{T}\nabla\log\mathbb{P}_{\theta_{n,0}}(s_t|s_{t-1})\frac{r(s_T)}{|s_T|}B_1\Big|\mathscr{F}_{n-1}\right]$$

$$- \sum_{s_T \in \mathcal{S}_T} \mathbb{E}\left[\xi_G(s_T)\sum_{t=1}^{T}\nabla\log\mathbb{P}_{\theta_{n,0}}(s_t|s_{t-1})\frac{B_2}{|s_T|}\Big|\mathscr{F}_{n-1}\right]$$

$$+ \sum_{s_T \in \mathcal{S}_T} \mathbb{E}\left[\xi_G(s_T)\sum_{t=1}^{T}\nabla\log\mathbb{P}_{\theta_{n,0}}(s_t|s_{t-1})\frac{\mu_{\theta_{n,0}}}{\sigma_{\theta_{n,0}}}B_3\Big|\mathscr{F}_{n-1}\right]$$

$$+ \sum_{s_T \in \mathcal{S}_T} \mathbb{E}\left[\xi_G(s_T)\sum_{t=1}^{T}\nabla\log\mathbb{P}_{\theta_{n,0}}(s_t|s_{t-1})|s|^{(-1)}_{\theta_{n,0}}\frac{\mu_{\theta_{n,0}}}{\sigma_{\theta_{n,0}}}\Big|\mathscr{F}_{n-1}\right]$$

$$= \mathcal{O}\left(\frac{1}{\sqrt{|G|}}\right) + \mathcal{O}(\sigma^2_{s,\theta_{n,0}}),$$

The constant hidden in the $\mathcal{O}$ notation depends only on the constants specified in the assumptions and is independent of $n$.

$\square$

### C.4.8 THE PROOF OF LEMMA C.5

*Proof.* First, for the event $\mathcal{B}^c(s_t, \theta_{n,s}, \theta_{n,0})$, we have:

$$\mathcal{B}^c(s_t, \theta_{n,s}, \theta_{n,0}) = \left\{\frac{\mathbb{P}_{\theta_{n,s}}(s_t|s_{t-1})}{\mathbb{P}_{\theta_{n,0}}(s_t|s_{t-1})} \geq 1 + \epsilon_{\text{high}},\ A_G(s_T) \geq 0\right\} \cup \left\{\frac{\mathbb{P}_{\theta_{n,s}}(s_t|s_{t-1})}{\mathbb{P}_{\theta_{n,0}}(s_t|s_{t-1})} \leq 1 - \epsilon_{\text{low}},\ A_G(s_T) < 0\right\}$$

$$\subset \left\{\left\|\frac{\mathbb{P}_{\theta_{n,s}}(s_t|s_{t-1})}{\mathbb{P}_{\theta_{n,0}}(s_t|s_{t-1})} - 1\right\| \geq \min\{\epsilon_{\text{low}}, \epsilon_{\text{high}}\}\right\}.$$

Next, based on the above derivation, we apply *Markov's* inequality to handle $\Xi_c(\theta_{n,s}, \theta_{n,0})$:

$$\|\Xi_c(\theta_{n,s}, \theta_{n,0})\| \tag{25}$$

$$\overset{\textit{Markov's inequality}}{\leq} \frac{2R}{\delta\min\{\epsilon_{\text{low}}, \epsilon_{\text{high}}\}} \sum_{s_T \in \mathcal{S}_T} \xi_G(s_T)\sum_{t=1}^{T}\left\|\frac{\mathbb{P}_{\theta_{n,s}}(s_t|s_{t-1})}{\mathbb{P}_{\theta_{n,0}}(s_t|s_{t-1})} - 1\right\|\|\nabla\log\mathbb{P}_{\theta_{n,0}}(s_t|s_{t-1})\|$$

$$\leq \frac{2R\mathcal{M}_N\sqrt{2L}}{\delta\min\{\epsilon_{\text{low}}, \epsilon_{\text{high}}\}} \sum_{s_T \in \mathcal{S}_T} \xi_G(s_T)\sum_{t=1}^{T}\|\mathbb{P}_{\theta_{n,0}}(s_t|s_{t-1}) - \mathbb{P}_{\theta_{n,s}}(s_t|s_{t-1})\|\sqrt{-\log\mathbb{P}_{\theta_{n,0}}(s_t|s_{t-1})}$$

$$\overset{\textit{Lagrange mean value theorem}}{\leq} \frac{2R\mathcal{M}_N\sqrt{2L}}{\delta\min\{\epsilon_{\text{low}}, \epsilon_{\text{high}}\}} \sum_{s_T \in \mathcal{S}_T} \xi_G(s_T)\sum_{t=1}^{T}\|\nabla\mathbb{P}_{\theta_{\zeta,s_t}}(s_t|s_{t-1})\|\|\theta_{n,s} - \theta_{n,0}\|\sqrt{-\log\mathbb{P}_{\theta_{n,0}}(s_t|s_{t-1})}$$

$$\overset{(*)}{\leq} \frac{2R\mathcal{M}_N L}{\delta\min\{\epsilon_{\text{low}}, \epsilon_{\text{high}}\}} \sum_{s_T \in \mathcal{S}_T} \xi_G(s_T)\sum_{t=1}^{T}\|\theta_{n,s} - \theta_{n,0}\|\sqrt{-\log\mathbb{P}_{\theta_{n,0}}(s_t|s_{t-1})}. \tag{26}$$

In the above derivation, at step $(*)$ we handle $\|\nabla \mathbb{P}_{\theta_{\zeta,s_t}}(s_t|s_{t-1})\|$ using the following method:

$$\|\nabla \mathbb{P}_{\theta_{\zeta,s_t}}(s_t|s_{t-1})\| = \mathbb{P}_{\theta_{\zeta,s_t}}(s_t|s_{t-1})\|\nabla \log \mathbb{P}_{\theta_{\zeta,s_t}}(s_t|s_{t-1})\|$$

$$\overset{\text{Lemma C.13}}{\leq} \sqrt{2L}\,\mathbb{P}_{\theta_{\zeta,s_t}}(s_t|s_{t-1})\sqrt{-\log \mathbb{P}_{\theta_{\zeta,s_t}}(s_t|s_{t-1})}$$

$$\leq \frac{\sqrt{2L}}{2}$$

Taking the conditional expectation with respect to $\mathscr{F}_{n-1}$ on both sides of Eq. 25, we obtain

$$\mathbb{E}\left[\|\Xi_c(\theta_{n,s},\theta_{n,0})\|\big|\mathscr{F}_{n-1}\right]$$

$$\leq \frac{2RL}{\delta \min\{\epsilon_{\text{low}}, \epsilon_{\text{high}}\}} \mathbb{E}\left[\mathcal{M}_N\|\theta_{n,s}-\theta_{n,0}\| \sum_{s_T \in \mathcal{S}_T} \xi_G(s_T) \sum_{t=1}^{T} \sqrt{-\log \mathbb{P}_{\theta_{n,0}}(s_t|s_{t-1})}\Big|\mathscr{F}_{n-1}\right]$$

$$\leq \frac{2RTL\sqrt{\log|\mathcal{V}|}}{\delta \min\{\epsilon_{\text{low}}, \epsilon_{\text{high}}\}} \mathbb{E}^{1/4}\left[\|\theta_{n,s}-\theta_{n,0}\|^4\big|\mathscr{F}_{n-1}\right] \sqrt{\mathbb{E}\left[\mathcal{M}_N^2\big|\mathscr{F}_{n-1}\right]}.$$

With this, we complete the proof. $\qquad\square$

### C.4.9 THE PROOF OF LEMMA C.6

*Proof.* For any $p \in \{1, 2, 4\}$, we can compute the following expression:

$$\|\theta_{n,s+1}-\theta_{n,s}\|^p$$

$$\leq \eta^p \left\|\sum_{s_T \in \mathcal{S}_T} \xi_G(s_T)\frac{1}{|s_T|}\sum_{t=1}^{T} \mathbf{1}_{\mathcal{B}(s_t,\theta_{n,s},\theta_{n,0})}\left(\nabla\left(\text{ClipMin}(s_T,\theta_{n,s},\theta_{n,0})\right)\right)A_G(s_T)\right\|^p$$

$$\overset{\text{\textit{AM-GM} inequality}}{\leq} \eta^p(2R)^p \underbrace{\left\|\sum_{s_T \in \mathcal{S}_T} \xi_G(s_T)\frac{1}{|s_T|}\sum_{t=1}^{T} \mathbf{1}_{\mathcal{B}(s_t,\theta_{n,s},\theta_{n,0})}\left(\nabla\left(\text{ClipMin}(s_T,\theta_{n,s},\theta_{n,0})\right)\right)\right\|^p}_{\Psi_{n,s}}.$$

$$(27)$$

Then for $\Psi_s$, we have:

$$\Psi_{n,s}$$

$$\overset{\text{\textit{AM-GM} inequality}}{\leq} \sum_{s_T \in \mathcal{S}_T} \xi_G(s_T)\frac{1}{|s_T|}\sum_{t=1}^{T} \mathbf{1}_{\mathcal{B}(s_t,\theta_{n,s},\theta_{n,0})}\|\nabla\left(\text{ClipMin}(s_T,\theta_{n,s},\theta_{n,0})\right)\|^p$$

$$\leq \sum_{s_T \in \mathcal{S}_T} \xi_G(s_T)\frac{1}{|s_T|}\sum_{t=1}^{T} \mathbf{1}_{\mathcal{B}(s_t,\theta_{n,s},\theta_{n,0})}\left(\frac{\mathbb{P}_{\theta_{n,s}}(s_t|s_{t-1})}{\mathbb{P}_{\theta_{n,0}}(s_t|s_{t-1})}\right)^p \|\nabla \log \mathbb{P}_{\theta_{n,s}}(s_t|s_{t-1})\|^p$$

$$\overset{\text{Lemma C.13}}{\leq} (2L)^{p/2}\sum_{s_T \in \mathcal{S}_T} \xi_G(s_T)\frac{1}{|s_T|}\sum_{t=1}^{T} \mathbf{1}_{\mathcal{B}(s_t,\theta_{n,s},\theta_{n,0})}\left(\frac{\mathbb{P}_{\theta_{n,s}}(s_t|s_{t-1})}{\mathbb{P}_{\theta_{n,0}}(s_t|s_{t-1})}\right)^p \left|-\log \mathbb{P}_{\theta_{n,s}}(s_t|s_{t-1})\right|^{p/2}$$

$$\leq (2L)^{p/2}\sum_{s_T \in \mathcal{S}_T} \xi_G(s_T)\mathcal{M}_N^p \frac{1}{|s_T|}\sum_{t=1}^{T} \mathbb{P}_{\theta_{n,s}}^p(s_t|s_{t-1})\left|-\log \mathbb{P}_{\theta_{n,s}}(s_t|s_{t-1})\right|^{p/2},$$

where $\mathcal{M}_N$ is defined in Eq. 14. In step $(*)$, we use the following inequality to obtain an upper bound:

$$\mathbf{1}_{\mathcal{B}(s_t,\theta_{n,s},\theta_{n,0})}\text{Clip}\left(\frac{\mathbb{P}_{\theta_{n,s}}(s_t|s_{t-1})}{\mathbb{P}_{\theta_{n,0}}(s_t|s_{t-1})}, \epsilon_{\text{low}}, \epsilon_{\text{high}}\right) \leq \mathbf{1}_{\mathcal{B}(s_t,\theta_{n,s},\theta_{n,0})}\frac{\mathbb{P}_{\theta_{n,s}}(s_t|s_{t-1})}{\mathbb{P}_{\theta_{n,0}}(s_t|s_{t-1})}.$$

Note that in this case we **cannot** apply the following relaxation:

$$\mathbf{1}_{\mathcal{B}(s_t,\theta_{n,s},\theta_{n,0})}\text{Clip}\left(\frac{\mathbb{P}_{\theta_{n,s}}(s_t|s_{t-1})}{\mathbb{P}_{\theta_{n,0}}(s_t|s_{t-1})}, \epsilon_{\text{low}}, \epsilon_{\text{high}}\right) \leq \mathbf{1}_{\mathcal{B}(s_t,\theta_{n,s},\theta_{n,0})}\epsilon_{\text{low}} \text{ or } \mathbf{1}_{\mathcal{B}(s_t,\theta_{n,s},\theta_{n,0})}\epsilon_{\text{high}}.$$

This is because, when $A_G(s_T) \leq 0$, the original Clip mechanism imposes no upper bound on the ratio $\mathbb{P}_{\theta_{n,s}}(s_t|s_{t-1}) / \mathbb{P}_{\theta_{n,0}}(s_t|s_{t-1})$. Then we take the conditional expectation with respect to $\mathscr{F}_{n-1}$ on both sides of the above inequality, and we obtain

$$
\mathbb{E}\left[\Psi_{n,s}|\mathscr{F}_{n-1}\right]
$$

$$
\leq (2L)^{p/2}\,\mathbb{E}\left[\sum_{s_T \in \mathcal{S}_T} \xi_G(s_T)\mathcal{M}_N^p \frac{1}{|s_T|}\sum_{t=1}^{T}\mathbb{P}_{\theta_{n,s}}^p(s_t|s_{t-1})\left|-\log \mathbb{P}_{\theta_{n,s}}(s_t|s_{t-1})\right|^{p/2}\,\middle|\,\mathscr{F}_{n-1}\right]
$$

$$
\overset{\textit{Cauchy–Schwarz}\ \text{inequality}}{\leq} (2L)^{p/2}\sqrt{\mathbb{E}\left[\sum_{s_T \in \mathcal{S}_T}\xi_G(s_T)\mathcal{M}_N^{2p}\,\middle|\,\mathscr{F}_{n-1}\right]}
$$

$$
\times \sqrt{\mathbb{E}\left[\sum_{s_T \in \mathcal{S}_T}\frac{1}{|s_T|}\sum_{t=1}^{T}\mathbb{P}_{\theta_{n,s}}(s_t|s_{t-1})\left|-\log \mathbb{P}_{\theta_{n,s}}(s_t|s_{t-1})\right|^p\,\middle|\,\mathscr{F}_{n-1}\right]}
$$

$$
\overset{\text{Lemma C.12}}{\leq}(2L)^{p/2}\sqrt{\mathbb{E}\left[\mathcal{M}_N^{2p}\,\middle|\,\mathscr{F}_{n-1}\right]}\,\log^{p/2}|\mathcal{V}|.
$$

Substituting the above estimate for $\Psi_s$ into Eq. 27, we obtain

$$
\mathbb{E}\left[\|\theta_{n,s+1}-\theta_{n,s}\|^p\big|\mathscr{F}_{n-1}\right]\leq \eta^p(2R)^p(2L)^{p/2}\sqrt{\mathbb{E}\left[\mathcal{M}_N^{2p}\big|\mathscr{F}_{n-1}\right]}\,\log^{p/2}|\mathcal{V}|.
$$

With this, we complete the proof. $\qquad\square$

### C.4.10 The Proof of Theorem 5.1

*Proof.* Then we focus on

$$
(J^* - J(\theta_{n+1,0})) - (J^* - J(\theta_{n,0})).
$$

To handle this, we invoke Lemma C.1. In particular, we have

$$
(J^* - J(\theta_{n+1,0})) - (J^* - J(\theta_{n,0}))
$$

$$
\overset{\text{Lemma C.1}}{\leq} -\nabla J(\theta_{n,0})^\top (\theta_{n+1,0}-\theta_{n,0}) + \frac{RL}{2}\left(2T\log|\mathcal{V}|+1\right)\|\theta_{n+1,0}-\theta_{n,0}\|^2
$$

$$
= \underbrace{-\eta\sum_{s=0}^{K-1}\nabla J(\theta_{n,0})^\top\nabla\mathcal{L}_{\text{GRPO}}(\theta_{n,s},\theta_{n,0})}_{Y_n} + \frac{RL}{2}\left(2T\log|\mathcal{V}|+1\right)\|\theta_{n+1,0}-\theta_{n,0}\|^2. \qquad (28)
$$

According to Eq. 8, we can further process the term $Y_n$, which yields:

$$Y_n = -\frac{\eta}{\sigma_{\theta_{n,0}}} \sum_{s=0}^{K-1} \|\nabla J(\theta_{n,0})\|^2 \underbrace{-\frac{\eta}{\sigma_{\theta_{n,0}}} \sum_{s=0}^{K-1} \nabla J(\theta_{n,0})^\top \left(\tilde{\nabla} J(\theta_{n,0}) - \nabla J(\theta_{n,0})\right)}_{M_n'}$$

$$-\frac{\eta}{\sigma_{\theta_{n,0}}} \sum_{s=0}^{K-1} \nabla J(\theta_{n,0})^\top \left(\Xi_g(\theta_{n,s}, \theta_{n,0}) + \Xi_s(\theta_{n,s}, \theta_{n,0}) + \Xi_c(\theta_{n,s}, \theta_{n,0})\right)$$

$$\leq -\frac{\eta}{2R} \sum_{s=0}^{K-1} \|\nabla J(\theta_{n,s})\|^2 + \frac{\eta}{\sigma_{\theta_{n,0}}} \underbrace{\sum_{s=0}^{K-1} \left|\|\nabla J(\theta_{n,0})\|^2 - \|\nabla J(\theta_{n,s})\|^2\right|}_{Y_{n,1}}$$

$$+\frac{\eta}{\sigma_{\theta_{n,0}}} \sum_{s=0}^{K-1} \|\nabla J(\theta_{n,0})\| \left(\|\Xi_g(\theta_{n,s}, \theta_{n,0})\| + \|\Xi_s(\theta_{n,s}, \theta_{n,0})\| + \|\Xi_c(\theta_{n,s}, \theta_{n,0})\|\right) + M_n'$$

$$\overset{\text{Lemma C.2}}{\leq} -\frac{\eta}{2R} \sum_{s=0}^{K-1} \|\nabla J(\theta_{n,s})\|^2 + \frac{\eta}{\delta} Y_{n,1}$$

$$+\frac{\eta}{\delta} \sqrt{2L} T R \sqrt{\log |\mathcal{V}|} \sum_{s=0}^{K-1} \left(\|\Xi_g(\theta_{n,s}, \theta_{n,0})\| + \|\Xi_s(\theta_{n,s}, \theta_{n,0})\| + \|\Xi_c(\theta_{n,s}, \theta_{n,0})\|\right) + M_n'$$

$$\overset{(*)}{\leq} -\frac{\eta}{2R} \sum_{s=0}^{K-1} \|\nabla J(\theta_{n,s})\|^2 + \frac{6\sqrt{2}}{\delta} \eta L^{3/2} T^2 R^2 \log^{3/2} |\mathcal{V}| \sum_{s=0}^{K-1} \|\theta_{n,s} - \theta_{n,0}\|$$

$$+\frac{9}{\delta} \eta R^2 T^2 L^2 \log^2 |\mathcal{V}| \sum_{s=0}^{K-1} \|\theta_{n,s} - \theta_{n,0}\|^2$$

$$+\frac{\eta}{\delta} \sqrt{2L} T R \sqrt{\log |\mathcal{V}|} \sum_{s=0}^{K-1} \left(\|\Xi_g(\theta_{n,s}, \theta_{n,0})\| + \|\Xi_s(\theta_{n,s}, \theta_{n,0})\| + \|\Xi_c(\theta_{n,s}, \theta_{n,0})\|\right) + M_n'.$$

$$(29)$$

It can be observed that the sequence $\{M_n', \mathscr{F}_n\}_{n \geq 1}$ constitutes a martingale difference sequence. In step $(*)$ we specifically apply the following relaxation to $Y_{n,1}$:

$$Y_{n,1} = \sum_{s=0}^{K-1} \left|\|\nabla J(\theta_{n,0})\|^2 - \|\nabla J(\theta_{n,s})\|^2\right|$$

$$= \sum_{s=0}^{K-1} \left|\|\nabla J(\theta_{n,0}) + \nabla J(\theta_{n,s}) - \nabla J(\theta_{n,0})\|^2 - \|\nabla J(\theta_{n,0})\|^2\right|$$

$$= \sum_{s=0}^{K-1} \left|\|\nabla J(\theta_{n,0})\|^2 + 2\|\nabla J(\theta_{n,0})\| \|\nabla J(\theta_{n,s}) - \nabla J(\theta_{n,0})\| + \|\nabla J(\theta_{n,s}) - \nabla J(\theta_{n,0})\|^2 - \|\nabla J(\theta_{n,0})\|^2\right|$$

$$\leq 2\sum_{s=0}^{K-1} \|\nabla J(\theta_{n,0})\| \|\nabla J(\theta_{n,s}) - \nabla J(\theta_{n,0})\| + \sum_{s=0}^{K-1} \|\nabla J(\theta_{n,s}) - \nabla J(\theta_{n,0})\|^2$$

$$\overset{\text{Lemma C.2 and C.1}}{\leq} 6\sqrt{2} L^{3/2} T^2 R^2 \log^{3/2} |\mathcal{V}| \sum_{s=0}^{K-1} \|\theta_{n,s} - \theta_{n,0}\| + 9R^2 T^2 L^2 \log^2 |\mathcal{V}| \sum_{s=0}^{K-1} \|\theta_{n,s} - \theta_{n,0}\|^2.$$

Taking the conditional expectation with respect to $\mathscr{F}_{n-1}$ on both sides of Eq. 29, we obtain

$$
\mathbb{E}\left[Y_n | \mathscr{F}_{n-1}\right]
$$

$$
\leq -\frac{\eta}{2R} \sum_{s=0}^{K-1} \mathbb{E}\left[\|\nabla J(\theta_{n,s})\|^2 | \mathscr{F}_{n-1}\right] + \frac{6\sqrt{2}}{\delta} \eta L^{3/2} T^2 R^2 \log^{3/2} |\mathcal{V}| \sum_{s=0}^{K-1} \mathbb{E}\left[\|\theta_{n,s} - \theta_{n,0}\| | \mathscr{F}_{n-1}\right]
$$

$$
+ \frac{9}{\delta} \eta R^2 T^2 L^2 \log^2 |\mathcal{V}| \sum_{s=0}^{K-1} \mathbb{E}\left[\|\theta_{n,s} - \theta_{n,0}\|^2 | \mathscr{F}_{n-1}\right]
$$

$$
+ \frac{\eta}{\delta} \sqrt{2L} T R \sqrt{\log |\mathcal{V}|} \sum_{s=0}^{K-1} \left(\mathbb{E}\left[\|\Xi_g(\theta_{n,s}, \theta_{n,0})\| | \mathscr{F}_{n-1}\right] + \mathbb{E}\left[\|\Xi_s(\theta_{n,s}, \theta_{n,0})\| | \mathscr{F}_{n-1}\right]\right.
$$

$$
\left. + \mathbb{E}\left[\|\Xi_c(\theta_{n,s}, \theta_{n,0})\| | \mathscr{F}_{n-1}\right]\right).
$$

Substituting the above result into Eq. 28, we obtain

$$
\mathbb{E}\left[J^* - J(\theta_{n+1,0}) - (J^* - J(\theta_{n,0})) | \mathscr{F}_{n-1}\right]
$$

$$
\leq -\frac{\eta}{2R} \sum_{s=0}^{K-1} \mathbb{E}\left[\|\nabla J(\theta_{n,s})\|^2 | \mathscr{F}_{n-1}\right] + \frac{6\sqrt{2}}{\delta} \eta L^{3/2} T^2 R^2 \log^{3/2} |\mathcal{V}| \sum_{s=0}^{K-1} \mathbb{E}\left[\|\theta_{n,s} - \theta_{n,0}\| | \mathscr{F}_{n-1}\right]
$$

$$
+ \left(\frac{9}{\delta} \eta R^2 T^2 L^2 \log^2 |\mathcal{V}| + \frac{1}{2}(2L \log |\mathcal{V}| + TL)\right) \sum_{s=0}^{K-1} \mathbb{E}\left[\|\theta_{n,s} - \theta_{n,0}\|^2 | \mathscr{F}_{n-1}\right]
$$

$$
+ \frac{\eta}{\delta} \sqrt{2L} T R \sqrt{\log |\mathcal{V}|} \sum_{s=0}^{K-1} \left(\mathbb{E}\left[\|\Xi_g(\theta_{n,s}, \theta_{n,0})\| | \mathscr{F}_{n-1}\right] + \mathbb{E}\left[\|\Xi_s(\theta_{n,s}, \theta_{n,0})\| | \mathscr{F}_{n-1}\right]\right.
$$

$$
\left. + \mathbb{E}\left[\|\Xi_c(\theta_{n,s}, \theta_{n,0})\| | \mathscr{F}_{n-1}\right]\right).
$$

Substituting into the above inequality the results on $\mathbb{E}\left[\|\Xi_g(\theta_{n,s}, \theta_{n,0})\| | \mathscr{F}_{n-1}\right]$, $\mathbb{E}\left[\|\Xi_s(\theta_{n,s}, \theta_{n,0})\| | \mathscr{F}_{n-1}\right]$, $\mathbb{E}\left[\|\Xi_c(\theta, \theta_{\mathrm{old}})\| | \mathscr{F}_{n-1}\right]$, from Lemmas C.3, C.4, C.5, respectively, we obtain

$$
\mathbb{E}\left[J^* - J(\theta_{n+1,0}) - (J^* - J(\theta_{n,0})) | \mathscr{F}_{n-1}\right]
$$

$$
\leq -\frac{\eta}{2R} \sum_{s=0}^{K-1} \mathbb{E}\left[\|\nabla J(\theta_{n,s})\|^2 | \mathscr{F}_{n-1}\right] + \mathcal{O}\left(\log^{3/2} |\mathcal{V}|\right) \eta \sum_{s=0}^{K-1} \mathbb{E}\left[\|\theta_{n,s} - \theta_{n,0}\| | \mathscr{F}_{n-1}\right]
$$

$$
+ \mathcal{O}\left(\log^2 |\mathcal{V}| \eta + \log |\mathcal{V}|\right) \sum_{s=0}^{K-1} \mathbb{E}\left[\|\theta_{n,s} - \theta_{n,0}\|^2 | \mathscr{F}_{n-1}\right]
$$

$$
+ \mathcal{O}\left(\sqrt{\log |\mathcal{V}|} \sqrt{\mathbb{E}\left[\mathcal{M}_N^2 | \mathscr{F}_{\theta_{n,0}}\right]}\right) \eta \sqrt{\sum_{s=0}^{K-1} \mathbb{E}\left[\|\theta_{n,s} - \theta_{n,0}\|^2 | \mathscr{F}_{n-1}\right]}
$$

$$
+ \mathcal{O}\left(\sqrt{\log |\mathcal{V}|}\right) \eta \left(\mathcal{O}\left(\frac{1}{\sqrt{|G|}}\right) + \mathcal{O}(\sigma_{s,\theta_{n,0}}^2)\right)
$$

$$
+ \mathcal{O}\left(\log^{3/2} |\mathcal{V}| \sqrt{\mathbb{E}\left[\mathcal{M}_N^2 | \mathscr{F}_{n,0}\right]}\right) \eta \left(\sum_{s=0}^{K-1} \mathbb{E}\left[\|\theta_{n,s} - \theta_{n,0}\|^4 | \mathscr{F}_{n-1}\right]\right)^{1/4}.
$$

Note that the quantities hidden in the $\mathcal{O}$ notation are constants depending only on other parameters of the problem and are independent of the iteration number $n$.

Then, substituting the estimate for

$$
\sum_{s=1}^{K} \mathbb{E}\left[\|\theta_{n,s} - \theta_{n,0}\|^p | \mathscr{F}_{n-1}\right], \quad p \in \{1, 2, 4\}
$$

from Lemma C.6 into the above expression, we finally obtain

$$\mathbb{E}\left[J^* - J(\theta_{n+1,0})\right] - \mathbb{E}\left[J^* - J(\theta_{n,0})\right]$$

$$\leq -\frac{\eta}{2R}\sum_{s=0}^{K-1}\mathbb{E}\left[\|\nabla J(\theta_{n,s})\|^2\right] + \mathcal{O}\left(\log^2|\mathcal{V}|\sqrt{\mathbb{E}\left[\mathcal{M}_N^2\right]}\right)\eta^2 + \mathcal{O}\left(\frac{1}{\sqrt{|G|}}\right)\eta + \mathcal{O}(\sigma_{s,\theta_{n,0}}^2)\eta.$$

Summing the above inequality over the index $n$ from $1$ to $N$, we finally obtain

$$\frac{1}{N}\sum_{n=1}^{N}\sum_{s=0}^{K-1}\mathbb{E}\left[\|\nabla J(\theta_{n,s})\|^2\right] = \mathcal{O}\left(\frac{1}{N\eta}\right) + \mathcal{O}\left(\log^2|\mathcal{V}|\sqrt{\mathbb{E}\left[\mathcal{M}_N^2\right]}\eta\right) + \mathcal{O}\left(\frac{1}{\sqrt{|G|}}\right) + \mathcal{O}(\bar{\sigma}_{s_T,N}^2).$$

Therefore, we conclude that when $\eta = \frac{1}{\log|\mathcal{V}|\sqrt{N}}$, we achieve the optimal convergence rate:

$$\frac{1}{N}\sum_{n=1}^{N}\sum_{s=0}^{K-1}\mathbb{E}\left[\|\nabla J(\theta_{n,s})\|^2\right] = \mathcal{O}\left(\frac{\log|\mathcal{V}|\sqrt{\mathbb{E}\left[\mathcal{M}_N^2\right]}}{\sqrt{N}}\right) + \mathcal{O}\left(\frac{1}{\sqrt{|G|}}\right) + \mathcal{O}(\bar{\sigma}_{s_T,N}^2).$$

With this, we complete the proof. □

### C.4.11 THE PROOF OF LEMMA C.7

*Proof.* By calculation, we obtain

$$\left\|\mathbf{1}_{\mathcal{D}(s_T,\theta,\theta_{\text{old}})}\frac{\nabla\mathbb{P}_\theta(s_T|s_0) - \nabla\mathbb{P}_{\theta_{\text{old}}}(s_T|s_0)}{\mathbb{P}_{\theta_{\text{old}}}(s_T|s_0)}\right\|$$

$$\leq \left\|\mathbf{1}_{\mathcal{D}(s_T,\theta,\theta_{\text{old}})}\frac{\mathbb{P}_\theta(s_T|s_0)}{\mathbb{P}_{\theta_{\text{old}}}(s_T|s_0)}\right\| \cdot \left\|\mathbf{1}_{\mathcal{D}(s_T,\theta,\theta_{\text{old}})}\frac{\nabla\mathbb{P}_\theta(s_T|s_0) - \nabla\mathbb{P}_{\theta_{\text{old}}}(s_T|s_0)}{\mathbb{P}_\theta(s_T|s_0)}\right\|$$

$$\leq \left\|\mathbf{1}_{\mathcal{D}(s_T,\theta,\theta_{\text{old}})}\frac{\mathbb{P}_\theta(s_T|s_0)}{\mathbb{P}_{\theta_{\text{old}}}(s_T|s_0)}\right\|\left\|\mathbf{1}_{\mathcal{D}(s_T,\theta,\theta_{\text{old}})}\nabla\log\mathbb{P}_{\theta_{\text{old}}}(s_T|s_0)\left(1 - \frac{\mathbb{P}_{\theta_{\text{old}}}(s_T|s_0)}{\mathbb{P}_\theta(s_T|s_0)}\right)\right\|$$

$$+ \left\|\mathbf{1}_{\mathcal{D}(s_T,\theta,\theta_{\text{old}})}\left(\nabla\log\mathbb{P}_\theta(s_T|s_0) - \nabla\log\mathbb{P}_{\theta_{\text{old}}}(s_T|s_0)\right)\right\|$$

$$\leq \left\|\mathbf{1}_{\mathcal{D}(s_T,\theta,\theta_{\text{old}})}\nabla\log\mathbb{P}_{\theta_{\text{old}}}(s_T|s_0)\left(\frac{\mathbb{P}_\theta(s_T|s_0)}{\mathbb{P}_{\theta_{\text{old}}}(s_T|s_0)} - 1\right)\right\| + |s_T|L\|\theta - \theta_{\text{old}}\|.$$

Next, we employ the following elementary inequality to handle the term $\frac{\mathbb{P}_\theta(s_T|s_0)}{\mathbb{P}_{\theta_{\text{old}}}(s_T|s_0)} - 1$:

$$|x - 1| \leq \frac{\epsilon_{\text{high}}}{\log(1 + \epsilon_{\text{high}})}\left|\log x\right|, \quad \forall x \in (0,\, 1 + \epsilon_{\text{high}}).$$

This is a trivial result, and hence we omit the proof here.

By applying the above inequality, we further obtain

$$\left\|\mathbf{1}_{\mathcal{D}(s_T,\theta,\theta_{\text{old}})}\frac{\nabla\mathbb{P}_\theta(s_T|s_0) - \nabla\mathbb{P}_{\theta_{\text{old}}}(s_T|s_0)}{\mathbb{P}_{\theta_{\text{old}}}(s_T|s_0)}\right\|$$

$$\leq \frac{\epsilon_{\text{high}}}{\log(1 + \epsilon_{\text{high}})}\left|\log\mathbb{P}_\theta(s_T|s_0) - \log\mathbb{P}_{\theta_{\text{old}}}(s_T|s_0)\right|\|\nabla\log\mathbb{P}_{\theta_{\text{old}}}(s_T|s_0)\| + |s_T|L\|\theta - \theta_{\text{old}}\|$$

$$\overset{\text{Lemma C.14}}{\leq} \frac{\epsilon_{\text{high}}}{\log(1 + \epsilon_{\text{high}})}\left(\|\nabla\log\mathbb{P}_{\theta_{\text{old}}}(s_T|s_0)\|^2\|\theta - \theta_{\text{old}}\| + |s_T|L\|\nabla\log\mathbb{P}_{\theta_{\text{old}}}(s_T|s_0)\|\|\theta - \theta_{\text{old}}\|^2\right)$$

$$+ |s_T|L\|\theta - \theta_{\text{old}}\|$$

$$\overset{\text{Lemma C.13}}{\leq} |s_T|L\left(-\frac{2\epsilon_{\text{high}}}{\log(1 + \epsilon_{\text{high}})}\log\mathbb{P}_{\theta_{\text{old}}}(s_T|s_0) + 1\right)\|\theta - \theta_{\text{old}}\|$$

$$+ \frac{\epsilon_{\text{high}}}{\log(1 + \epsilon_{\text{high}})}(|s_T|L)^{3/2}\sqrt{-\log\mathbb{P}_{\theta_{\text{old}}}(s_T|s_0)}\|\theta - \theta_{\text{old}}\|^2.$$

Next, summing over the relevant terms and applying *Jensen's* inequality, we obtain

$$
\mathbb{E}\left[\|\Delta_{g,1}(\theta, \theta_{\text{old}})\| \,\middle|\, \mathscr{F}_{\theta_{\text{old}}}\right]
$$

$$
\leq 2R \sum_{s_T \in \mathcal{S}_T} \mathbb{E}\left[(\xi_G(s_T) + \mathbb{P}_{\theta_{\text{old}}}(s_T|s_0)) \left\|\mathbf{1}_{\mathcal{D}(s_T,\theta,\theta_{\text{old}})} \frac{\nabla \mathbb{P}_\theta(s_T|s_0) - \nabla \mathbb{P}_{\theta_{\text{old}}}(s_T|s_0)}{\mathbb{P}_{\theta_{\text{old}}}(s_T|s_0)}\right\| \,\middle|\, \mathscr{F}_{\theta_{\text{old}}}\right]
$$

$$
\leq 4R \sum_{s_T \in \mathcal{S}_T} |s_T|\, \mathbb{E}\left[\mathbb{P}_{\theta_{\text{old}}}(s_T|s_0) L\left(-\frac{2\epsilon_{\text{high}}}{\log(1+\epsilon_{\text{high}})}\log \mathbb{P}_{\theta_{\text{old}}}(s_T|s_0) + 1\right)\|\theta - \theta_{\text{old}}\| \,\middle|\, \mathscr{F}_{\theta_{\text{old}}}\right]
$$

$$
+ 4R \sum_{s_T \in \mathcal{S}_T} |s_T|\, \mathbb{E}\left[\mathbb{P}_{\theta_{\text{old}}}(s_T|s_0)\frac{\epsilon_{\text{high}}}{\log(1+\epsilon_{\text{high}})} T^{3/2} L^{3/2}\sqrt{-\log \mathbb{P}_{\theta_{\text{old}}}(s_T|s_0)}\|\theta - \theta_{\text{old}}\|^2 \,\middle|\, \mathscr{F}_{\theta_{\text{old}}}\right]
$$

$$
\overset{\text{\textit{Jensen's} inequality}}{\leq} 4RLT\left(1 + \frac{\epsilon_{\text{high}}}{\log(1+\epsilon_{\text{high}})}\log|\mathcal{V}|\right)\mathbb{E}\left[\|\theta - \theta_{\text{old}}\| \,\middle|\, \mathscr{F}_{\theta_{\text{old}}}\right]
$$

$$
+ 4RT^{3/2}L^{3/2}\frac{\epsilon_{\text{high}}}{\log(1+\epsilon_{\text{high}})}\sqrt{\log|\mathcal{V}|}\,\mathbb{E}\left[\|\theta - \theta_{\text{old}}\|^2 \,\middle|\, \mathscr{F}_{\theta_{\text{old}}}\right].
$$

With this, we complete the proof. $\qquad\square$

### C.4.12 THE PROOF OF LEMMA C.8

*Proof.* We first focus on the event $\mathcal{D}^c(s_T, \theta, \theta_{\text{old}})$, for which we have:

$$
\mathcal{D}^c(s_T, \theta, \theta_{\text{old}}) = \left\{\frac{\mathbb{P}_\theta(s_T|s_0)}{\mathbb{P}_{\theta_{\text{old}}}(s_T|s_0)} \geq 1 + \epsilon_{\text{high}}\right\}
$$

$$
\subset \left\{|\log \mathbb{P}_\theta(s_T|s_0) - \log \mathbb{P}_{\theta_{\text{old}}}(s_T|s_0)| \geq \log(1+\epsilon_{\text{high}})\right\}
$$

$$
\overset{\text{Lemma C.14}}{\subset} \left\{\|\nabla\log \mathbb{P}_{\theta_{\text{old}}}(s_T|s_0)\|\|\theta - \theta_{\text{old}}\| + |s_T|L\|\theta - \theta_{\text{old}}\|^2 \geq \log(1+\epsilon_{\text{high}})\right\}. \tag{30}
$$

Then we clearly have

$$
\mathbb{E}\left[\|\Delta_{g,2}(\theta, \theta_{\text{old}})\| \,\middle|\, \mathscr{F}_{\theta_{\text{old}}}\right]
$$

$$
\leq 4R\,\mathbb{E}\left[\sum_{s_T \in \mathcal{S}_T}(\xi_G(s_T) + \mathbb{P}_{\theta_{\text{old}}}(s_T|s_0))\mathbf{1}_{\mathcal{D}^c(s_T,\theta,\theta_{\text{old}})}\|\nabla\log \mathbb{P}_{\theta_{\text{old}}}(s_T \mid s_0)\| \,\middle|\, \mathscr{F}_{\theta_{\text{old}}}\right]
$$

$$
\overset{\text{\textit{Markov's} inequality}}{\leq} \frac{4R}{\log(1+\epsilon_{\text{high}})}\,\underbrace{\mathbb{E}\left[\sum_{s_T \in \mathcal{S}_T}(\xi_G(s_T) + \mathbb{P}_{\theta_{\text{old}}}(s_T|s_0))\|\nabla\log \mathbb{P}_{\theta_{\text{old}}}(s_T|s_0)\|^2\|\theta - \theta_{\text{old}}\| \,\middle|\, \mathscr{F}_{\theta_{\text{old}}}\right]}_{O_1}
$$

$$
+ \frac{4LR}{\log(1+\epsilon_{\text{high}})}\,\underbrace{\mathbb{E}\left[\sum_{s_T \in \mathcal{S}_T}(\xi_G(s_T) + \mathbb{P}_{\theta_{\text{old}}}(s_T|s_0))\|\nabla\log \mathbb{P}_{\theta_{\text{old}}}(s_T|s_0)\|\|\theta - \theta_{\text{old}}\|^2 \,\middle|\, \mathscr{F}_{\theta_{\text{old}}}\right]}_{O_2}.
$$

$$
\tag{31}
$$

We split the analysis into two parts. We first treat $O_1$, we have

$$O_1 \overset{\text{Cauchy–Schwarz inequality}}{\le} \mathbb{E}\left[\sqrt{\sum_{s_T\in\mathcal{S}_T}(\xi_G(s_T)+\mathbb{P}_{\theta_{\text{old}}}(s_T|s_0))\|\theta-\theta_{\text{old}}\|^2}\right.$$

$$\left.\times\sqrt{\sum_{s_T\in\mathcal{S}_T}(\xi_G(s_T)+\mathbb{P}_{\theta_{\text{old}}}(s_T|s_0))\|\nabla\log\mathbb{P}_{\theta_{\text{old}}}(s_T|s_0)\|^4}\,\middle|\,\mathscr{F}_{\theta_{\text{old}}}\right]$$

$$\overset{\text{Cauchy–Schwarz inequality}}{\le} \sqrt{\mathbb{E}\left[\sum_{s_T\in\mathcal{S}_T}(\xi_G(s_T)+\mathbb{P}_{\theta_{\text{old}}}(s_T|s_0))\|\theta-\theta_{\text{old}}\|^2\,\middle|\,\mathscr{F}_{\text{old}}\right]}$$

$$\times\sqrt{\mathbb{E}\left[\sum_{s_T\in\mathcal{S}_T}(\xi_G(s_T)+\mathbb{P}_{\theta_{\text{old}}}(s_T|s_0))\|\nabla\log\mathbb{P}_{\theta_{\text{old}}}(s_T|s_0)\|^4\,\middle|\,\mathscr{F}_{\theta_{\text{old}}}\right]}$$

$$=\sqrt{2\,\mathbb{E}[\|\theta-\theta_{\text{old}}\|^2|\mathscr{F}_{\theta_{\text{old}}}]}\sqrt{\sum_{s_T\in\mathcal{S}_T}\|\nabla\log\mathbb{P}_{\theta_{\text{old}}}(s_T|s_0)\|^4\,\mathbb{E}\left[(\xi_G(s_T)+\mathbb{P}_{\theta_{\text{old}}}(s_T|s_0))\,\middle|\,\mathscr{F}_{\theta_{\text{old}}}\right]}$$

$$=2\sqrt{\mathbb{E}[\|\theta-\theta_{\text{old}}\|^2|\mathscr{F}_{\theta_{\text{old}}}]}\sqrt{\sum_{s_T\in\mathcal{S}_T}\mathbb{P}_{\theta_{\text{old}}}(s_T|s_0)\|\nabla\log\mathbb{P}_{\theta_{\text{old}}}(s_T|s_0)\|^4}$$

$$\overset{\text{Lemma C.13}}{\le} 4TL\sqrt{\mathbb{E}[\|\theta-\theta_{\text{old}}\|^2|\mathscr{F}_{\theta_{\text{old}}}]}\sqrt{\sum_{s_T\in\mathcal{S}_T}\mathbb{P}_{\theta_{\text{old}}}(s_T|s_0)|\log\mathbb{P}_{\theta_{\text{old}}}(s_T|s_0)|^2}$$

$$\overset{\text{Lemma C.12}}{\le} 4T^2 L\log|\mathcal{V}|\sqrt{\mathbb{E}[\|\theta-\theta_{\text{old}}\|^2|\mathscr{F}_{\theta_{\text{old}}}]}.$$

For $O_2$, by a similar argument we obtain

$$O_2\le 2\sqrt{2}LT\sqrt{\log|\mathcal{V}|}\sqrt{\mathbb{E}[\|\theta-\theta_{\text{old}}\|^4|\mathscr{F}_{\theta_{\text{old}}}]}.$$

Substituting the above estimates for $O_1$ and $O_2$ into Eq. 31, we finally obtain

$$\mathbb{E}[\|\Delta_{g,2}(\theta,\theta_{\text{old}})\|\,|\mathscr{F}_{\theta_{\text{old}}}]\le \frac{4R}{\log(1+\epsilon_{\text{high}})}\left[4T^2 L\log|\mathcal{V}|\sqrt{\mathbb{E}[\|\theta-\theta_{\text{old}}\|^2|\mathscr{F}_{\theta_{\text{old}}}]}\right.$$

$$\left.+2\sqrt{2}L^{3/2}T\sqrt{\log|\mathcal{V}|}\sqrt{\mathbb{E}[\|\theta-\theta_{\text{old}}\|^4|\mathscr{F}_{\theta_{\text{old}}}]}\right].$$

With this, we complete the proof. $\qquad\square$

### C.4.13 THE PROOF OF LEMMA C.9

*Proof.* First, for the event $\mathcal{D}^c(s_T,\theta,\theta_{\text{old}})$, we have:

$$\mathcal{D}^c(s_T,\theta,\theta_{\text{old}})=\left\{\frac{\mathbb{P}_\theta(s_T|s_0)}{\mathbb{P}_{\theta_{\text{old}}}(s_T|s_0)}\ge 1+\epsilon_{\text{high}}\right\}$$

$$\subset\{|\log\mathbb{P}_\theta(s_T|s_0)-\log\mathbb{P}_{\theta_{\text{old}}}(s_T|s_0)|\ge\log(1+\epsilon_{\text{high}})\}$$

$$\overset{\text{Lemma C.14}}{\subset}\left\{\|\nabla\log\mathbb{P}_\theta(s_T|s_0)\|\|\theta-\theta_{\text{old}}\|+TL\|\theta-\theta_{\text{old}}\|^2\ge\log(1+\epsilon_{\text{high}})\right\}. \tag{32}$$

Then by direct computation, we obtain

$$\mathbb{E}[\|\Delta_c(\theta,\theta_{\text{old}})\|\,|\mathscr{F}_{\theta_{\text{old}}}]\le 4R\sum_{s_T\in\mathcal{S}_T}\mathbb{E}\left[\mathbf{1}_{\mathcal{D}^c(s_T,\theta,\theta_{\text{old}})}\mathbb{P}_\theta(s_T\mid s_0)\|\nabla\log\mathbb{P}_\theta(s_T|s_0)\|\,\middle|\mathscr{F}_{\theta_{\text{old}}}\right]$$

$$\overset{\text{Markov's inequality}}{\le}\frac{4R}{\log(1+\epsilon_{\text{high}})}\underbrace{\sum_{s_T\in\mathcal{S}_T}\mathbb{E}\left[\|\theta-\theta_{\text{old}}\|\mathbb{P}_\theta(s_T|s_0)\|\nabla\log\mathbb{P}_\theta(s_T|s_0)\|^2|\mathscr{F}_{\theta_{\text{old}}}\right]}_{R_1}$$

$$+\frac{4RLT}{\log(1+\epsilon_{\text{high}})}\underbrace{\sum_{s_T\in\mathcal{S}_T}\mathbb{E}\left[\|\theta-\theta_{\text{old}}\|^2\mathbb{P}_\theta(s_T|s_0)\|\nabla\log\mathbb{P}_\theta(s_T|s_0)\|\,|\mathscr{F}_{\theta_{\text{old}}}\right]}_{R_2}. \tag{33}$$

We divide the expression into two parts. For the first part $R_1$, we have:

$$R_1 \overset{\text{Cauchy–Schwarz inequality}}{\leq} \sqrt{\mathbb{E}\left[\sum_{s_T \in \mathcal{S}_T} \|\theta - \theta_{\text{old}}\|^2 \, \mathbb{P}_\theta(s_T|s_0)\bigg| \mathscr{F}_{\theta_{\text{old}}}\right]}$$

$$\times \sqrt{\mathbb{E}\left[\sum_{s_T \in \mathcal{S}_T} \mathbb{P}_\theta(s_T|s_0)\|\nabla \log \mathbb{P}_\theta(s_T|s_0)\|^4 \bigg| \mathscr{F}_{\theta_{\text{old}}}\right]}$$

$$\overset{\text{Lemma C.13}}{\leq} 2L\sqrt{\mathbb{E}\left[\|\theta - \theta_{\text{old}}\|^2|\mathscr{F}_{\theta_{\text{old}}}\right]}\sqrt{\mathbb{E}\left[\sum_{s_T \in \mathcal{S}_T} \mathbb{P}_\theta(s_T|s_0)|\log \mathbb{P}_\theta(s_T|s_0)|^2 \bigg| \mathscr{F}_{\theta_{\text{old}}}\right]}$$

$$\overset{\text{Lemma C.12}}{\leq} 2T^2 L \log |\mathcal{V}|\sqrt{\mathbb{E}\left[\|\theta - \theta_{\text{old}}\|^2|\mathscr{F}_{\theta_{\text{old}}}\right]}.$$

For $O_2$, by a similar argument we obtain

$$R_2 \leq 2\sqrt{2L}T\sqrt{\log |\mathcal{V}|}\sqrt{\mathbb{E}\left[\|\theta - \theta_{\text{old}}\|^4|\mathscr{F}_{\theta_{\text{old}}}\right]}.$$

Substituting the above estimates for $R_1$ and $R_2$ into Eq. 33, we finally obtain

$$\mathbb{E}\left[\|\Delta_c(\theta, \theta_{\text{old}})\||\mathscr{F}_{\theta_{\text{old}}}\right]$$
$$\leq \frac{4\sqrt{2}RLT^2}{\log(1 + \epsilon_{\text{high}})}\left(\sqrt{2}\log |\mathcal{V}|\sqrt{\mathbb{E}\left[\|\theta - \theta_{\text{old}}\|^2|\mathscr{F}_{\theta_{\text{old}}}\right]} + \sqrt{L}\log |\mathcal{V}|\sqrt{\mathbb{E}\left[\|\theta - \theta_{\text{old}}\|^4|\mathscr{F}_{\theta_{\text{old}}}\right]}\right).$$

With this, we complete the proof. $\qquad\square$

### C.4.14 THE PROOF OF LEMMA C.10

*Proof.* First, we know that

$$B'_G(s_T) := A'_G(s_T) - \frac{1}{\sigma'_{\theta_{\text{old}}}}\left(\frac{r(s_T)}{|s_T|} - \mu'_{\theta_{\text{old}}}\right).$$

Then, we can compute the following difference:

$$B'_G(s_T) \leq \frac{|r(s_T)|}{|s_T|}\left|\frac{1}{\sigma_G + \delta} - \frac{1}{\sigma'_{\theta_{\text{old}}}}\right| + \left|\frac{\mu'_G}{\sigma_G + \delta} - \frac{\mu'_{\theta_{\text{old}}}}{\sigma'_{\theta_{\text{old}}}}\right|$$

$$\leq R\left|\frac{1}{\sigma_G + \delta} - \frac{1}{\sigma'_{\theta_{\text{old}}}}\right| + R\left|\frac{1}{\sigma_G + \delta} - \frac{1}{\sigma'_{\theta_{\text{old}}}}\right| + \frac{1}{\delta}\left|\mu'_G - \mu'_{\theta_{\text{old}}}\right|$$

$$= 2R\left|\frac{1}{\sigma_G + \delta} - \frac{1}{\sigma'_{\theta_{\text{old}}}}\right| + \frac{1}{\delta}\left|\mu'_G - \mu'_{\theta_{\text{old}}}\right|$$

$$\leq \frac{2R}{\delta^2}\left|\sigma_G - \sigma'_{\theta_{\text{old}}}\right| + \frac{1}{\delta}\left|\mu'_G - \mu'_{\theta_{\text{old}}}\right|.$$

By a straightforward calculation, we obtain

$$\mathbb{E}\left[\sum_{s_T \in \mathcal{S}_T} \xi_G(s_T)B'^2_G(s_T)\bigg|\mathscr{F}_{\theta_{\text{old}}}\right]$$

$$\leq \frac{4R^2}{\delta^4}\mathbb{E}\left[\sum_{s_T \in \mathcal{S}_T} \xi_G(s_T)\left(\sigma_G - \mathbb{E}\left[\sigma_G|\mathscr{F}_{\theta_{\text{old}}}\right]\right)^2\bigg|\mathscr{F}_{\theta_{\text{old}}}\right] + \frac{1}{\delta^2}\mathbb{E}\left[\sum_{s_T \in \mathcal{S}_T} \xi_G(s_T)\left(\mu'_G - \mu'_{\theta_{\text{old}}}\right)^2\bigg|\mathscr{F}_{\theta_{\text{old}}}\right]$$

$$\leq \frac{8R^2}{\delta^4|G|} + \frac{1}{\delta^2}\frac{1}{|G|}. \tag{34}$$

Then we can estimate $\mathbb{E}\left[\|\Delta_s(\theta, \theta_{\text{old}})\||\mathscr{F}_{\theta_{\text{old}}}\right]$. Specifically, we have

$$\mathbb{E}\left[\|\Delta_{s,1}(\theta, \theta_{\text{old}})\||\mathscr{F}_{\theta_{\text{old}}}\right]$$

$$\leq \sqrt{2TL}(1 + \epsilon_{\text{high}}) \sqrt{\sum_{s_T \in \mathcal{S}_T} \mathbb{E}\left[\xi_G(s_T)\|\log \mathbb{P}_\theta(s_T|s_0)\|^2\Big|\mathscr{F}_{\theta_{\text{old}}}\right]} \sqrt{\mathbb{E}\left[\sum_{s_T \in \mathcal{S}_T} \xi_G(s_T)B_G'^2(s_T)\Big|\mathscr{F}_{\theta_{\text{old}}}\right]}$$

$$= \mathcal{O}\left(\frac{1}{\sqrt{|G|}}\right).$$

With this, we complete the proof. $\qquad\square$

### C.4.15 THE PROOF OF LEMMA C.11

*Proof.* For any $p \in \{1, 2, 4\}$, we can compute the following expression:

$$\|\theta_{n,s+1} - \theta_{n,s}\|^p \leq \eta^p \left\|\sum_{s_T \in \mathcal{S}_T} \xi_G(s_T)\mathbf{1}_{\mathcal{D}(s_T,\theta_{n,s},\theta_{n,0})}\frac{1}{|s_T|}\nabla\left(\overline{\text{ClipMin}}\left(s_T, \theta_{n,s}, \theta_{n,0}\right)\right) A_G'(s_T)\right\|^p$$

$$\stackrel{\text{AM-GM inequality}}{\leq} \eta^p(2R)^p \underbrace{\sum_{s_T \in \mathcal{S}_T} \xi_G(s_T)\mathbf{1}_{\mathcal{D}(s_T,\theta_{n,s},\theta_{n,0})}\left\|\frac{1}{|s_T|}\nabla\left(\overline{\text{ClipMin}}\left(s_T, \theta_{n,s}, \theta_{n,0}\right)\right)\right\|^p}_{\Theta_{n,s}}. \quad (35)$$

Next, we derive bounds for $\Theta_s$. As a consequence, we obtain

$$\Theta_{n,s} \leq (1 + \epsilon_{\text{high}})^p \sum_{s_T \in \mathcal{S}_T} \xi_G(s_T)\left\|\frac{1}{|s_T|}\sum_{t=1}^T \nabla \log \mathbb{P}_{\theta_{n,s}}(s_t|s_{t-1})\right\|^p$$

$$\leq (1 + \epsilon_{\text{high}})^p \sum_{s_T \in \mathcal{S}_T} \xi_G(s_T)\frac{1}{|s_T|}\sum_{t=1}^T \left\|\nabla \log \mathbb{P}_{\theta_{n,s}}(s_t|s_{t-1})\right\|^p$$

$$\leq (1 + \epsilon_{\text{high}})^p 2^{p-1} \sum_{s_T \in \mathcal{S}_T} \xi_G(s_T)\frac{1}{|s_T|}\sum_{t=1}^T \|\nabla \log \mathbb{P}_{\theta_{n,0}}(s_t|s_{t-1})\|^p$$

$$+ (1 + \epsilon_{\text{high}})^p 2^{p-1} \sum_{s_T \in \mathcal{S}_T} \xi_G(s_T)\frac{1}{|s_T|}\sum_{t=1}^T \|\nabla \log \mathbb{P}_{\theta_{n,s}}(s_t|s_{t-1}) - \nabla \log \mathbb{P}_{\theta_{n,0}}(s_t|s_{t-1})\|^p$$

$$\stackrel{\text{Lemma C.14}}{\leq} (1 + \epsilon_{\text{high}})^p 2^{p-1} \sum_{s_T \in \mathcal{S}_T} \xi_G(s_T)\frac{1}{|s_T|}\sum_{t=1}^T \|\nabla \log \mathbb{P}_{\theta_{n,0}}(s_t|s_{t-1})\|^p$$

$$+ (1 + \epsilon_{\text{high}})^p 2^{p-1} L^p \sum_{s_T \in \mathcal{S}_T} \xi_G(s_T)\|\theta_{n,s} - \theta_{n,0}\|^p$$

$$= (1 + \epsilon_{\text{high}})^p 2^{p-1} \sum_{s_T \in \mathcal{S}_T} \xi_G(s_T)\frac{1}{|s_T|}\sum_{t=1}^T \|\nabla \log \mathbb{P}_{\theta_{n,0}}(s_t|s_{t-1})\|^p$$

$$+ (1 + \epsilon_{\text{high}})^p 2^{p-1} L^p \|\theta_{n,s} - \theta_{n,0}\|^p. \quad (36)$$

Taking the conditional expectation with respect to $\mathscr{F}_{n-1}$ on both sides of the above inequality, we get:

$$\mathbb{E}\left[\Theta_{n,s}|\mathscr{F}_{n-1}\right]$$

$$\leq (1 + \epsilon_{\text{high}})^p 2^{p-1} \left(\sum_{s_T \in \mathcal{S}_T} \mathbb{P}_{\theta_{n,0}}(s_T|s_0)\frac{1}{|s_T|}\sum_{t=1}^T \|\nabla \log \mathbb{P}_{\theta_{n,0}}(s_t|s_{t-1})\|^p + L^p\|\theta_{n,s} - \theta_{n,0}\|^p\right)$$

$$\stackrel{\text{Lemma C.13}}{\leq} (1 + \epsilon_{\text{high}})^p 2^{p-1} \left((2L)^{p/2}\sum_{s_T \in \mathcal{S}_T} \mathbb{P}_{\theta_{n,0}}(s_T|s_0)\frac{1}{|s_T|}\sum_{t=1}^T |-\log \mathbb{P}_{\theta_{n,0}}(s_t|s_{t-1})|^{p/2} + L^p\|\theta_{n,s} - \theta_{n,0}\|^p\right)$$

$$\stackrel{\text{Lemma C.12}}{\leq} (1 + \epsilon_{\text{high}})^p 2^{p-1} \left((2L)^{p/2}\log^{p/2}|\mathcal{V}| + L^p\mathbb{E}\left[\|\theta_{n,s} - \theta_{n,0}\|^p|\mathscr{F}_{n-1}\right]\right).$$

Substituting the above result into Eq. 35, we obtain

$$
\mathbb{E}\left[\|\theta_{n,s+1} - \theta_{n,s}\|^p | \mathscr{F}_{n-1}\right]
$$
$$
\leq \eta^p (2R)^p (1 + \epsilon_{\text{high}})^p 2^{p-1} \left((2L)^{p/2} \log^{p/2} |\mathcal{V}| + L^{p/2} \mathbb{E}\left[\|\theta_{n,s} - \theta_{n,0}\|^p | \mathscr{F}_{n-1}\right]\right). \quad (37)
$$

We now consider

$$
\sum_{s=0}^{K} \mathbb{E}\left[\|\theta_{n,s+1} - \theta_{n,0}\|^p | \mathscr{F}_{n-1}\right]
$$

and obtain

$$
\sum_{s=1}^{K} \mathbb{E}\left[\|\theta_{n,s} - \theta_{n,0}\|^p | \mathscr{F}_{n-1}\right]
$$
$$
\overset{\text{AM-GM inequality}}{\leq} \sum_{s=1}^{K} s^{p-1} \sum_{k=1}^{s} \mathbb{E}\left[\|\theta_{n,k} - \theta_{n,k-1}\|^p | \mathscr{F}_{n-1}\right]
$$
$$
\overset{\text{Eq. 37}}{\leq} \sum_{s=1}^{K} s^{p-1} \sum_{k=1}^{s} \eta^p (2R)^p (1 + \epsilon_{\text{high}})^p 2^{p-1} \left((2L)^{p/2} \log^{p/2} |\mathcal{V}| + L^{p/2} \mathbb{E}\left[\|\theta_{n,k-1} - \theta_{n,0}\|^p | \mathscr{F}_{n-1}\right]\right)
$$
$$
\leq K^{p+1} \eta^p (2R)^p (1 + \epsilon_{\text{high}})^p 2^{p-1} (2L)^{p/2} \log^{p/2} |\mathcal{V}|
$$
$$
+ K^p \eta^p (2R)^p (1 + \epsilon_{\text{high}})^p 2^{p-1} L^p \sum_{s=1}^{K} \mathbb{E}\left[\|\theta_{n,s} - \theta_{n,0}\|^p | \mathscr{F}_{n-1}\right].
$$

Since we have the following condition on the learning rate $\eta$:

$$
\eta \leq \frac{1}{4K(1 + \epsilon_{\text{high}})RL}.
$$

Hence we can further obtain

$$
\sum_{s=1}^{K} \mathbb{E}\left[\|\theta_{n,s} - \theta_{n,0}\|^p | \mathscr{F}_{n-1}\right]
$$
$$
\leq K^{p+1} \eta^p (2R)^p (1 + \epsilon_{\text{high}})^p 2^{p-1} (2L)^{p/2} \log^{p/2} |\mathcal{V}|
$$
$$
+ K^p \eta^p (2R)^p (1 + \epsilon_{\text{high}})^p 2^{p-1} L^p \sum_{s=1}^{K} \mathbb{E}\left[\|\theta_{n,s} - \theta_{n,0}\|^p | \mathscr{F}_{n-1}\right]
$$
$$
\leq K^{p+1} \eta^p (2R)^p (1 + \epsilon_{\text{high}})^p 2^{p-1} (2L)^{p/2} \log^{p/2} |\mathcal{V}|
$$
$$
+ \frac{1}{2} \sum_{s=1}^{K} \mathbb{E}\left[\|\theta_{n,s} - \theta_{n,0}\|^p | \mathscr{F}_{n-1}\right],
$$

which means,

$$
\sum_{s=1}^{K} \mathbb{E}\left[\|\theta_{n,s} - \theta_{n,0}\|^p | \mathscr{F}_{n-1}\right] \leq K^{p+1} \eta^p (2R)^p (1 + \epsilon_{\text{high}})^p 2^p (2L)^{p/2} \log^{p/2} |\mathcal{V}|.
$$

With this, we complete the proof.

$\square$

### C.4.16 THE PROOF OF THEOREM 5.2

*Proof.* First we focus on

$$
(J^* - J(\theta_{n+1,0})) - (J^* - J(\theta_{n,0})).
$$

To handle this, we invoke Lemma C.1. In particular, we have

$$(J^* - J(\theta_{n+1,0})) - (J^* - J(\theta_{n,0}))$$

$$\overset{\text{Lemma C.1}}{\leq} -\nabla J(\theta_{n,0})^\top (\theta_{n+1,0} - \theta_{n,0}) + \frac{RL}{2} (2T \log |\mathcal{V}| + 1) \|\theta_{n+1,0} - \theta_{n,0}\|^2$$

$$= \underbrace{-\eta \sum_{s=0}^{K-1} \nabla J(\theta_{n,0})^\top \nabla \mathcal{L}_{\text{TIC-GRPO}}(\theta_{n,s}, \theta_{n,0}) + \frac{RL}{2} (2T \log |\mathcal{V}| + 1) \|\theta_{n+1,0} - \theta_{n,0}\|^2.}_{X_n} \quad (38)$$

Then we get that

$$X_n \overset{\text{Eq. 15}}{=} -\frac{\eta}{\sigma_{\theta_{n,0}}} \sum_{s=0}^{K-1} \nabla J(\theta_{n,0})^\top \nabla J(\theta_{n,s}) \underbrace{- \frac{\eta}{\sigma_{\theta_{n,0}}} \sum_{s=0}^{K-1} \nabla J(\theta_{n,0})^\top M_{\theta_{n,0},1}}_{M_n}$$

$$- \frac{\eta}{\sigma_{\theta_{n,0}}} \sum_{s=0}^{K-1} \nabla J(\theta_{n,0})^\top (\Delta_{g,1}(\theta_{n,s}, \theta_{n,0}) + \Delta_{g,2}(\theta_{n,s}, \theta_{n,0}) + \Delta_c(\theta_{n,s}, \theta_{n,0}) + \Delta_s(\theta_{n,s}, \theta_{n,0}))$$

$$= -\frac{\eta}{2R} \sum_{s=0}^{K-1} \|\nabla J(\theta_{n,s})\|^2 + \frac{\eta}{\sigma_{\theta_{n,0}}} \sum_{s=0}^{K-1} \nabla J(\theta_{n,s})^\top (\nabla J(\theta_{n,0}) - \nabla J(\theta_{n,s}))$$

$$- \frac{\eta}{\sigma_{\theta_{n,0}}} \sum_{s=0}^{K-1} \nabla J(\theta_{n,0})^\top (\Delta_{g,1}(\theta_{n,s}, \theta_{n,0}) + \Delta_{g,2}(\theta_{n,s}, \theta_{n,0}) + \Delta_c(\theta_{n,s}, \theta_{n,0}) + \Delta_s(\theta_{n,s}, \theta_{n,0}))$$

$$+ M_n$$

$$\overset{\text{Lemma C.2}}{\leq} -\frac{\eta}{2R} \sum_{s=0}^{K-1} \|\nabla J(\theta_{n,s})\|^2 + \frac{\eta}{\delta} \sqrt{2L} R^2 \sqrt{\log |\mathcal{V}|} (2 \log |\mathcal{V}| + 1) L \sum_{s=0}^{K-1} \|\theta_{n,s} - \theta_{n,0}\|$$

$$+ \frac{\eta}{\delta} \sqrt{2L} R \sqrt{\log |\mathcal{V}|} \sum_{s=0}^{K-1} (\|\Delta_{g,1}(\theta_{n,s}, \theta_{n,0})\| + \|\Delta_{g,2}(\theta_{n,s}, \theta_{n,0})\| + \|\Delta_c(\theta_{n,s}, \theta_{n,0})\| + \|\Delta_s(\theta_{n,s}, \theta_{n,0})\|)$$

$$+ M_n.$$

It can be observed that the sequence $\{M_n, \mathscr{F}_n\}_{n \geq 1}$ constitutes a martingale difference sequence. Taking the conditional expectation with respect to $\mathscr{F}_{n-1}$ on both sides of the above inequality, we obtain

$$\mathbb{E}[X_n | \mathscr{F}_{n-1}]$$

$$\leq -\frac{\eta}{2R} \sum_{s=0}^{K-1} \mathbb{E}[\|\nabla J(\theta_{n,s})\|^2 | \mathscr{F}_{n-1}] + \frac{3\sqrt{2}\eta}{\delta} L^{3/2} \log^{3/2} |\mathcal{V}| \sum_{s=0}^{K-1} \mathbb{E}[\|\theta_{n,s} - \theta_{n,0}\| \big| \mathscr{F}_{n-1}]$$

$$+ \frac{\eta}{\delta} \sqrt{2L} R \sqrt{\log |\mathcal{V}|} \sum_{s=0}^{K-1} (\mathbb{E}[\|\Delta_{g,1}(\theta_{n,s}, \theta_{n,0})\| \big| \mathscr{F}_{n-1}] + \mathbb{E}[\|\Delta_{g,2}(\theta_{n,s}, \theta_{n,0})\| \big| \mathscr{F}_{n-1}]$$

$$+ \mathbb{E}[\|\Delta_c(\theta_{n,s}, \theta_{n,0})\| \big| \mathscr{F}_{n-1}] + \mathbb{E}[\|\Delta_s(\theta_{n,s}, \theta_{n,0})\| \big| \mathscr{F}_{n-1}]).$$

Substituting into the above inequality the results on $\mathbb{E}[\|\Delta_{g,1}(\theta_{n,s}, \theta_{n,0})\| \big| \mathscr{F}_{n-1}]$, $\mathbb{E}[\|\Delta_{g,2}(\theta_{n,s}, \theta_{n,0})\| \big| \mathscr{F}_{n-1}]$, $\mathbb{E}[\|\Delta_c(\theta, \theta_{\text{old}})\| \big| \mathscr{F}_{n-1}]$, and $\mathbb{E}[\|\Delta_s(\theta, \theta_{\text{old}})\| \big| \mathscr{F}_{n-1}]$ from Lemmas

C.15, C.8, C.9, and C.10, respectively, we obtain

$$\mathbb{E}\left[X_n|\mathscr{F}_{n-1}\right]$$

$$\leq -\frac{\eta}{2R}\sum_{s=0}^{K-1}\mathbb{E}\left[\|\nabla J(\theta_{n,s})\|^2|\mathscr{F}_{n-1}\right] + \mathcal{O}(\log^{3/2}|\mathcal{V}|)\eta\sum_{s=0}^{K-1}\mathbb{E}\left[\|\theta_{n,s}-\theta_{n,0}\|\big|\mathscr{F}_{n-1}\right]$$

$$+ \mathcal{O}(\sqrt{\log|\mathcal{V}|})\sum_{s=0}^{K-1}\mathbb{E}\left[\|\theta_{n,s}-\theta_{n,0}\|^2|\mathscr{F}_{n-1}\right] + \mathcal{O}(\log|\mathcal{V}|)\eta\sqrt{\sum_{s=0}^{K-1}\mathbb{E}\left[\|\theta_{n,s}-\theta_{n,0}\|^2\big|\mathscr{F}_{n-1}\right]}$$

$$+ \mathcal{O}(\log|\mathcal{V}|)\sqrt{\sum_{s=0}^{K-1}\mathbb{E}\left[\|\theta_{n,s}-\theta_{n,0}\|^4\big|\mathscr{F}_{n-1}\right]} + \mathcal{O}\left(\frac{1}{\sqrt{|G|}}\right)\eta\bigg).$$

Note that the quantities hidden in the $\mathcal{O}$ notation are constants depending only on other parameters of the problem and are independent of the iteration number $n$.

Then, substituting the estimate for

$$\sum_{s=1}^{K}\mathbb{E}\left[\|\theta_{n,s}-\theta_{n,0}\|^p|\mathscr{F}_{n-1}\right], \ \ p\in\{1,2,4\}$$

from Lemma C.11 into the above expression, we finally obtain

$$\mathbb{E}\left[X_n|\mathscr{F}_{n-1}\right] \leq -\frac{\eta}{2R}\sum_{s=0}^{K-1}\mathbb{E}\left[\|\nabla J(\theta_{n,s})\|^2|\mathscr{F}_{n-1}\right] + \mathcal{O}\left(\log^2|\mathcal{V}|\eta^2\right) + \mathcal{O}\left(\frac{1}{\sqrt{|G|}}\right)\eta.$$

Substituting the above expression into Eq. 38, we finally obtain

$$\mathbb{E}\left[(J^*-J(\theta_{n+1,0})) - (J^*-J(\theta_{n,0}))\,|\mathscr{F}_{n-1}\right]$$

$$\leq -\frac{\eta}{2R}\sum_{s=0}^{K-1}\mathbb{E}\left[\|\nabla J(\theta_{n,s})\|^2|\mathscr{F}_{n-1}\right] + \mathcal{O}\left(\log^2|\mathcal{V}|\eta^2\right)$$

$$+ \mathcal{O}\left(\frac{\log|\mathcal{V}|}{|G|^{1/4}}\right)\eta + \frac{1}{2}\left(2TL\log|\mathcal{V}| + L\right)\mathbb{E}\left[\|\theta_{n+1,0}-\theta_{n,0}\|^2|\mathscr{F}_{n-1}\right]$$

$$\leq -\frac{\eta}{2R}\sum_{s=0}^{K-1}\mathbb{E}\left[\|\nabla J(\theta_{n,s})\|^2|\mathscr{F}_{n-1}\right] + \mathcal{O}\left(\log^2|\mathcal{V}|\eta^2\right) + \mathcal{O}\left(\frac{1}{\sqrt{|G|}}\right)\eta$$

$$+ \frac{1}{2}\left(2TL\log|\mathcal{V}| + L\right)\sum_{s=1}^{K}\mathbb{E}\left[\|\theta_{n,s}-\theta_{n,0}\|^2|\mathscr{F}_{n-1}\right]$$

$$\overset{\text{Lemma C.11}}{\leq} -\frac{\eta}{2R}\sum_{s=0}^{K-1}\mathbb{E}\left[\|\nabla J(\theta_{n,s})\|^2|\mathscr{F}_{n-1}\right] + \mathcal{O}\left(\log^2|\mathcal{V}|\eta^2\right) + \mathcal{O}\left(\frac{1}{\sqrt{|G|}}\right)\eta.$$

Taking expectation on both sides of the above inequality, we obtain

$$\mathbb{E}\left[J^*-J(\theta_{n+1,0})\right] - \mathbb{E}\left[J^*-J(\theta_{n,0})\right]$$

$$\leq -\frac{\eta}{2R}\sum_{s=0}^{K-1}\mathbb{E}\left[\|\nabla J(\theta_{n,s})\|^2\right] + \mathcal{O}\left(\log^2|\mathcal{V}|\eta^2\right) + \mathcal{O}\left(\frac{1}{\sqrt{|G|}}\right)\eta.$$

Summing the above inequality over the index $n$ from 1 to $N$, we finally obtain

$$\frac{1}{N}\sum_{n=1}^{N}\sum_{s=0}^{K-1}\mathbb{E}\left[\|\nabla J(\theta_{n,s})\|^2\right] = \mathcal{O}\left(\frac{1}{N\eta}\right) + \mathcal{O}(\log^2|\mathcal{V}|\eta) + \mathcal{O}\left(\frac{1}{\sqrt{|G|}}\right).$$

Therefore, we conclude that when $\eta = \frac{1}{\log|\mathcal{V}|\sqrt{N}}$, we achieve the optimal convergence rate:

$$\frac{1}{N}\sum_{n=1}^{N}\sum_{s=0}^{K-1}\mathbb{E}\left[\|\nabla J(\theta_{n,s})\|^2\right] = \mathcal{O}\left(\frac{\log|\mathcal{V}|}{\sqrt{N}}\right) + \mathcal{O}\left(\frac{1}{\sqrt{|G|}}\right).$$

With this, we complete the proof. $\qquad\square$

