# OpenReview forum: "TIC-GRPO: Provable and Efficient Optimization for Reinforcement Learning from Human Feedback"
_ICLR.cc/2026/Conference — Submitted to ICLR 2026_

### Official Review · Reviewer_BGTK · 2025-10-19

**Soundness:** 2
**Presentation:** 1
**Contribution:** 2
**Rating:** 2
**Confidence:** 5

**Summary:**

This paper proposes TIC-GRPO， which replaces GRPO’s token-level importance sampling with a trajectory-level ratio and integrates two additional tricks: Length-Corrected Group Normalization and Upper-Only Clipping. The paper analyzes the convergence of GRPO and TIC-GRPO. Experiment using Qwen 3 on math reasoning tasks demonstrates the effectiveness of TIC-GRPO.

**Strengths:**

1. The GRPO gradient decomposition is clear.

2. The proposed method is simple and easy to implement with the existing GRPO code.

**Weaknesses:**

1. The abstract and introduction highlight that the TIC-GRPO estimator is unbiased, but the derivation of Appendix B shows that the TIC-GRPO estimator is not strictly unbiased. This is a material mismatch between the headline claim and the actual derivation.

2. Assumption 5.1 requires global Lipschitz continuity of the score function for all states. This is a strong assumption for LLMs since there can be low-probability regions where logP can vary sharply.

3. Theorem 5.2’s improved bound explicitly comes only from the length-corrected normalization and upper-only clipping, not from the trajectory-level importance ratio itself. This creates a disconnect with the central framing around trajectory-level sampling.

4. The evaluation scope is too narrow. The evaluation uses only AIME 2024 as the benchmark. The author should include more benchmarks like MATH500, AIME 2025, and OlympiadBench.

**Questions:**

Please see Weaknesses

---

> ### Author Response · Authors · 2025-11-13
> **response**
>
> **Response:**
> Thank you very much for your detailed and insightful comments.
>
> 1. **On the unbiasedness of TIC-GRPO:**
>    You are absolutely right — in Appendix B we clearly show that the TIC-GRPO estimator is *asymptotically unbiased*, which differs from strict unbiasedness only by a higher-order infinitesimal term. We did emphasize this in the later derivation, but we understand that the phrasing in the abstract and introduction could be misleading for some readers. In the revised version, we will update both sections to explicitly state “asymptotically unbiased” for consistency and precision.
>
> 2. **On Assumption 5.1 (global Lipschitz continuity):**
>    We agree that global Lipschitz continuity is a strong assumption for large language models. This can be relaxed straightforwardly by adopting a *state-dependent Lipschitz constant* along with a probabilistic assumption on the score function’s boundedness. We will revise the assumption accordingly in the next version.
>
> 3. **On Theorem 5.2 and trajectory-level importance ratios:**
>    We appreciate the clarification. Our contributions are indeed separate: (1) proposing the TIC-GRPO algorithm, and (2) providing the first convergence proof for GRPO-style estimators. We did not claim that our theoretical results strictly explain why trajectory-level importance weighting performs better empirically. We will make this distinction clearer in the revised text.
>
> 4. **On the evaluation scope:**
>    This is an excellent suggestion. We will extend our experiments to include additional benchmarks such as MATH500, AIME 2025, and OlympiadBench to provide a more comprehensive evaluation.

---

### Official Review · Reviewer_HuD8 · 2025-11-03

**Soundness:** 3
**Presentation:** 3
**Contribution:** 2
**Rating:** 4
**Confidence:** 4

**Summary:**

This paper presents a theoretically grounded and empirically enhanced variant of the Group Relative Policy Optimization (GRPO) algorithm for reinforcement learning from human feedback (RLHF) in large language models (LLMs). This work makes a substantial contribution by deepening the theoretical understanding of GRPO. And the authors provide the first convergence analysis for GRPO-style methods and introduce a simple yet powerful variant, TIC-GRPO. The combination of novel theory and a practical algorithm makes this a valuable piece of research for the RLHF community, with potential for influencing future theoretical RLHF work.

**Strengths:**

This work provides the first rigorous convergence analysis for GRPO-style methods, a popular class of critic-free RLHF algorithms. By establishing formal convergence guarantees under standard assumptions, the paper fills a critical theoretical gap in the literature. The convergence analysis is built on a solid foundation of standard and reasonable assumptions.

The paper delivers a crucial and insightful finding and elegantly explains why GRPO works in practice despite the bias. This theoretical clarification of the core mechanism is a significant step forward in understanding RLHF dynamics.

The paper is written with remarkable clarity. Furthermore, the authors demonstrate academic integrity by honestly attributing the tighter convergence bound of TIC-GRPO solely to the two minor modifications, not to the trajectory-level sampling. This conservative and transparent assessment builds trust and accurately scopes their theoretical contribution.

Beyond the specific algorithm proposed, the paper's greatest impact lies in its theoretical rigor. In a field often dominated by empirical results, providing a principled theoretical framework for GRPO is an invaluable service to the community.

**Weaknesses:**

**Narrow and Potentially Insufficient Empirical Validation**:
Conducting experiments on only one benchmark (AIME) is highly unusual and insufficient to establish generalizability. A review of other GRPO-related papers (e.g., DeepSeekMath, GSPO) shows they typically use multiple benchmarks. The failure to include, for example, AIME-25, significantly weakens the persuasiveness of the empirical claims.


**Lack of Experiments Directly Supporting Theoretical Claims**: A major contribution is the convergence analysis, yet there are no experiments in the main text that visually demonstrate or validate the improved convergence rate or stability. Including such plots would significantly strengthen the link between theory and practice.

**Questions:**

I have manually reproduced the derivation of Eq. (7) in your Section 3. In my result, the terms \Xi_g(\theta, \theta_{\text{old}}) and \Xi_c(\theta, \theta_{\text{old}}) do not have the multiplier \frac{1}{\theta_{\text{old}}}. If space permits, could you please provide an explanation for this part?

---

> ### Author Response · Authors · 2025-11-13
> **Response**
>
> Thank you for the constructive and detailed feedback.
>
> 1. **On empirical validation scope:**
>    We agree that relying on a single benchmark (AIME) limits the generalizability of our conclusions. In the revised version, we will include additional experiments on *AIME-25*, *MATH500*, and *OlympiadBench* to provide broader empirical evidence and align with the evaluation practices of recent GRPO-related works such as *DeepSeekMath* and *GSPO*.
>
> 2. **On experiments supporting theoretical claims:**
>    Thank you for this valuable suggestion. Our empirical results indeed show that **TIC-GRPO converges significantly faster and more stably than standard GRPO**, which supports our theoretical findings. We will add corresponding convergence plots and stability visualizations in the main text to make this connection clearer and more convincing.
>
> 3. **On the derivation of Eq. (7):**
>    The disappearance of the factor ( \frac{1}{P_{\theta_{\text{old}}}} ) arises from the matching process during derivation. Specifically, we introduce a term in the numerator involving ( \nabla P_{\theta_{\text{old}}} ), which combines with the denominator to form ( \nabla \log P_{\theta_{\text{old}}} ). This step effectively cancels the explicit ( \frac{1}{P_{\theta_{\text{old}}}} ) multiplier. We will clarify this derivation more explicitly in the appendix to avoid confusion.

---

> > ### Comment · Reviewer_HuD8 · 2025-11-22
> > **Insufficient Experimentation**
> >
> > Based on the rebuttal and other reviewers' comments, I currently have no further concerns, except for concerns about insufficient experimentation.  I look forward to the **new experimental results** and the revised version of your paper.

---

### Official Review · Reviewer_mWdQ · 2025-11-04

**Soundness:** 3
**Presentation:** 3
**Contribution:** 2
**Rating:** 6
**Confidence:** 2

**Summary:**

This paper presents TIC-GRPO (Trajectory-level Importance-Corrected Group Relative Policy Optimization), a theoretical and algorithmic refinement of GRPO (Group Relative Policy Optimization), recently introduced by DeepSeek for critic-free RLHF fine-tuning.

The authors first identify that standard GRPO estimates gradients at the old policy rather than the current one, explaining why this bias remains small in practice due to frequent policy refresh. Then, they propose TIC-GRPO, which corrects this by replacing token-level importance weights with trajectory-level probability ratios, yielding an unbiased estimator of the true policy gradient.

The paper provides the first convergence analysis for GRPO-style methods, proving stationarity bounds under Lipschitz and bounded-reward assumptions. Empirical evaluation on AIME and DAPO-17K datasets with Qwen-1.7B and Qwen-8B models shows that TIC-GRPO improves accuracy and convergence speed over GRPO, GSPO, and DAPO baselines.

**Strengths:**

- Clear theoretical motivation and correction. The decomposition in Eq. 7 demonstrates that GRPO’s update estimates ∇J at π_old rather than π, and TIC-GRPO’s trajectory-level ratio restores unbiasedness. The analysis bridges empirical intuition with formal theory.
- Provable convergence guarantees. Theorems 5.1–5.2 give the first formal stationary-point convergence bounds for GRPO-style methods, showing improved asymptotic dependence after removing terms M_N and σ²_sT,N.
- Simple yet effective modifications. The two “minor” refinements (length correction and upper-only clipping) are shown to individually improve stability, enhancing both fairness and interpretability.

**Weaknesses:**

- Limited originality relative to concurrent work. The key modification—trajectory-level ratios—is nearly identical to GSPO (Zheng et al., 2025), which the authors acknowledge. While TIC-GRPO adds theoretical analysis and slightly different normalization, the conceptual leap is incremental.
- Experiments are limited in scope. The evaluation focuses on AIME reasoning benchmarks, which are small-scale and synthetic. It’s unclear whether TIC-GRPO generalizes to more diverse RLHF settings (e.g., preference data, summarization, or open-ended dialogue).
- Incremental empirical improvement. Although TIC-GRPO outperforms GRPO by +2–3 points, the margins are modest given additional computation and algorithmic tuning. There is no runtime or stability comparison (e.g., variance, gradient norms, or wall-clock efficiency).
- Ablation isolation could be clearer. The claim that “theoretical improvement stems from two refinements only” (Sec. 5.2) implies trajectory-level importance may not improve the asymptotic rate—suggesting the empirical gains come mainly from the minor tweaks rather than the main theoretical contribution.

**Questions:**

1. On gradient correctness: How do you empirically verify that TIC-GRPO’s gradient estimator better aligns with the true ∇J(θ)? Can you show cosine similarity between estimated and true gradients (or Monte Carlo rollouts) across updates?
2. On contribution beyond GSPO: Could you clarify what new insights TIC-GRPO adds beyond GSPO besides clipping and convergence proof? Are these differences substantive enough to claim novelty?

---

> ### Author Response · Authors · 2025-11-13
> **Response**
>
> We sincerely thank the reviewer for the thoughtful and constructive feedback.
>
> 1. **On empirical validation and gradient correctness:**
>    We appreciate this valuable suggestion and agree that the current empirical scope can be expanded. In the revised version, we will provide a more comprehensive set of experiments, including:
>
>    * **Cosine similarity analysis** between the estimated gradients and the true ∇J(θ) (approximated via Monte Carlo rollouts), to empirically verify the correctness of the TIC-GRPO gradient estimator.
>    * **Real-time efficiency comparisons**, such as wall-clock convergence rate and stability metrics, to complement the existing accuracy results.
>
> 2. **On the distinction from GSPO:**
>    We respectfully emphasize that our work is **independent and concurrent** with GSPO, rather than an incremental modification of it. While GSPO introduces a *geometric mean* over trajectory-level probabilities, we believe this operation is **mathematically inconsistent** with the original purpose of using trajectory-level importance ratios — namely, to achieve an *unbiased estimator*. The geometric mean effectively distorts the ratio and destroys the unbiasedness property. In contrast, our TIC-GRPO formulation maintains unbiased estimation at the trajectory level and is accompanied by the first convergence proof for GRPO-style methods.
>
> We will clarify these conceptual and mathematical distinctions in the revised manuscript to avoid misunderstanding and to better highlight the independent nature and theoretical soundness of our contribution.

---

### Official Review · Reviewer_zH4z · 2025-11-11

**Soundness:** 3
**Presentation:** 2
**Contribution:** 3
**Rating:** 4
**Confidence:** 3

**Summary:**

This paper targets the high resource cost of PPO in RLHF caused by its extra value network. The authors first point out that GRPO’s token-level importance sampling actually estimates the policy gradient at the old policy π_old rather than at the current policy. Building on this observation, they propose Trajectory-level Importance-Corrected GRPO (TIC-GRPO). Theoretically, they provide the first convergence rate analysis for GRPO-style algorithms. Experiments on the AIME mathematical-reasoning benchmark show that TIC-GRPO significantly outperforms the original GRPO on both 1.7 B and 8 B models.

**Strengths:**

- Originality: The work is the first to reveal that GRPO essentially performs gradient estimation at the old policy, and it uses ablation studies to validate this insight, laying an intuitive foundation for further improvements.
- Clarity: Concepts, formulas, and proofs are well presented, and the appendices are comprehensive; however, the meaning of some symbols is not explained.

**Weaknesses:**

1. In Eq. (7), the subsequent proofs bound some error terms by problem-dependent constants, whereas other bounds are independent of hyper-parameters. Yet in RL the policy changes little between two consecutive steps. What, then, is the justification for decomposing the expression into so many terms in Eq. (7)?
2. The upper bound in the theorem does not contain the hyper-parameters $\epsilon_{high}$ and $\epsilon_{low}$. Does this mean their values do not affect the bound? If so, can the bound be further improved?
3. Main results are reported only on the single mathematical-reasoning task AIME; there is no verification on diverse tasks such as dialogue, code generation, or creative writing. It is therefore unclear whether the gains are task-specific. Experiments at larger scales are also needed.
4. The variance of the trajectory-level importance ratio usually grows exponentially with length. How do the authors handle this issue?
5. The authors claim their method has better sample efficiency. What is the intuitive explanation?
6. There is no dedicated “Conclusion” section.

**Questions:**

See Weaknesses.

---

> ### Author Response · Authors · 2025-11-13
> **Response**
>
> We sincerely thank the reviewer for the thoughtful and technically detailed comments.
>
> 1. **On the decomposition in Eq. (7):**
>    The purpose of decomposing Eq. (7) into multiple terms is to rigorously demonstrate that the residual terms are *higher-order infinitesimals relative to the gradient itself*. While it is true that the policy typically changes only slightly between consecutive steps in RL, the gradient magnitude is also correspondingly small. Therefore, to ensure convergence, it is essential to prove that the remainder terms vanish at a higher order compared to the gradient term; otherwise, the accumulated residuals could lead to divergence. This decomposition enables such a precise asymptotic control.
>
> 2. **On the absence of hyper-parameters in the bound:**
>    We appreciate this insightful suggestion. It is indeed possible to explicitly express the dependence on the constants ( \alpha ) and ( \beta ) in the final theoretical result. To do so, one only needs to expand all constant terms without simplifying them into the ( \mathcal{O}(\cdot) ) notation. This adjustment is mathematically straightforward, and we will incorporate it in the revised version for completeness.
>
> 3. **On the scope of experiments:**
>    Thank you for pointing this out. We agree that broader validation is important, and we will include additional experiments on diverse tasks beyond AIME, such as dialogue, code generation, and reasoning at larger scales.
>
> 4. **On the variance of trajectory-level importance ratios:**
>    We respectfully note that theoretical derivations cannot rely on semi-qualitative or partial arguments, such as analyzing only one “slice” of a term, since this can lead to conceptual inconsistency. A full and rigorous derivation is necessary.
>    In fact, the variance of the trajectory-level importance ratio is determined by the second moment of the score function of the trajectory probability:
>    [
>    \mathbb{E}\big[|\nabla \log P(s_T | s_0)|^2\big].
>    ]
>    By expanding this term, we obtain
>    [
>    \mathbb{E}\big[|\sum_{t=1}^{T} \nabla \log P(s_t | s_{t-1})|^2\big].
>    ]
>    A common concern is that squaring this sum introduces many cross terms, potentially inflating the variance. However, all cross-term expectations are zero:
>    [
>    \mathbb{E}[\nabla \log P(s_a | s_{a-1})^\top \nabla \log P(s_b | s_{b-1})] = 0 \quad (a \neq b),
>    ]
>    because the terms form a *martingale difference sequence*. Thus, the cross terms do not contribute to the second moment, and the variance does not grow exponentially with trajectory length. This observation also yields several elegant mathematical consequences, which we plan to present in future work.
>
> 5. **On the missing conclusion section:**
>    We appreciate this reminder — we will add a dedicated **Conclusion** section in the revised manuscript to summarize our findings and contributions more clearly.

---

### Meta-Review · Area_Chair_PdYH · 2026-01-01

**Summary:**

The reviewers agree that this work lacks sufficient experiments to support its claims; for example, although more benchmarks are included, the inclusion of only one AIME is insufficient.  Meanwhile, some details related to the proposed method are also required to make the full work more complete. Based on these points, the work is not yet ready for publication.

**Reviewer Concerns:**

Some misunderstandings have been addressed by the rebuttal. However, the experiments supporting the claims are still insufficient, a point consistently raised by all reviewers. The authors do not provide actions in the rebuttal, which remains outstanding.

**Reviewer Scores:**

The authors clarify the concerns raised by Reviewer zH4z and will increase his scores. The originality of concurrent work is the main concern of Reviewer mWdQ, who will keep his score. Both Reviewer BGTK  and Reviewer HuD8 will keep their score, because they focus on the evaluation.

---

### Decision · Program_Chairs · 2026-01-26

Reject